# The promotion effect of nitrous acid on aerosol formation in wintertime Beijing: possible contribution of traffic-related emission

Yongchun Liu[1*], Yusheng Zhang[1], Chaofan Lian[2,6], Chao Yan[3], Zeming Feng[1], Feixue Zheng[1], Xiaolong Fan[1], Yan Chen[2,6], Weigang Wang[2,6*], Biwu Chu[3,4], Yonghong Wang[3], Jing Cai[3], Wei Du[3], Kaspar R. Daellenbach[3], Juha Kangasluoma[1,3], Federico Bianchi[1,3], Joni Kujansuu[1,3], Tuukka Petäjä[3], Xuefei Wang[6], Bo Hu[5], Yuesi Wang[5], Maofa Ge[2], Hong He[4] and Markku Kulmala[1,3*]

1. Aerosol and Haze Laboratory, Advanced Innovation Center for Soft Matter Science and Engineering, Beijing University of Chemical Technology, Beijing, 100029, China

2. State Key Laboratory for Structural Chemistry of Unstable and Stable Species, Beijing National Laboratory for Molecular Sciences, Institute of Chemistry, Chinese Academy of Sciences, Beijing 100190, China

3. Institute for Atmospheric and Earth System Research/Physics, Faculty of Science, University of Helsinki, P.O. Box 64, FI-00014, Finland

4. State Key Joint Laboratory of Environment Simulation and Pollution Control, Research Center for Eco-Environmental Sciences, Chinese Academy of Sciences, Beijing, 100085, China

5. State Key Laboratory of Atmospheric Boundary Layer Physics and Atmospheric Chemistry, Institute of Atmospheric Physics, Chinese Academy of Sciences, Beijing, 100029, China

6. University of Chinese Academy of Sciences, Beijing 100049, PR China

*Correspondence to:* liuyc@buct.edu.cn, wangwg@iccas.ac.cn or markku.kulmala@helsinki.fi

## **Abstract**

Secondary aerosol is a major component of $PM_{2.5}$, yet its formation mechanism in the ambient atmosphere is still an open question. Based on field measurements in downtown Beijing, we show that the photolysis of nitrous acid (HONO) could promote the formation of organic and nitrate aerosol in wintertime Beijing as evidenced by the growth of the mass concentration of organic and nitrate aerosols linearly increasing as a function of consumed HONO from early morning to noon. The increased nitrate also lead to the formation of particulate matter ammonium by enhancing the neutralization of nitric acid by ammonia. We further illustrate that over 50 % of the ambient HONO during pollution events in wintertime Beijing might be related to traffic-related emission including direct emission and formation via the reaction between OH and vehicle-emitted NO. Overall, our results highlight that the traffic-related HONO plays an important role in the oxidative capacity and in turn, contribute to the haze formation in winter Beijing. Mitigation of HONO and $NO_x$ emission from the vehicles might be an effective way to reduce secondary aerosol mass formation and severe haze events in wintertime Beijing.

# 1.    Introduction

China is one of the most suffering countries from the pollution of fine particulate matter with diameter less than or equal to 2.5 μm ($PM_{2.5}$) (Lelieveld et al., 2015). Although the regional air quality has been continuously improving since the central government of China issued the Clean Air Act in 2013 (Vu et al., 2019), $PM_{2.5}$ concentration is still significantly higher than that in developed countries (Fu et al., 2014;An et al., 2019). Nowadays, a consensus has been reached that haze events are driven by local emissions (An et al., 2019), regional transport (Zheng et al., 2015b) and secondary formation (Huang et al., 2014;He et al., 2018) of pollutants under unfavorable meteorological conditions (stagnant atmosphere and high relative humidity) (Zhu et al., 2018;Liu et al., 2017c). A feedback loop between meteorological parameters and haze formation has also been found playing an important role in the evolution of haze events (Zhang et al., 2018).

Secondary aerosol can contribute up to ~70 % to the aerosol mass concentration on polluted days (Huang et al., 2014). Several reaction pathways have been proposed in the atmospheric chemistry community, such as sulfate formation via heterogeneous oxidation of $SO_2$ promoted by $H_2O_2$ and/or $NO_2$ on mineral dust (Huang et al., 2015;He et al., 2014), aqueous oxidation of $SO_2$ promoted by $NO_2$ in the presence or absence of $NH_3$ in particle-bound water film (He et al., 2014;Wang et al., 2016), catalytic conversion of $SO_2$ to sulfate by black carbon (Zhang et al., 2020), nitrate formation via efficient hydrolysis of $N_2O_5$ on aerosol surfaces (Wang et al., 2017c;Wang et al., 2019;Kulmala, 2018;Li et al., 2017), and the haze formation initiated by new particle

formation and growth (Guo et al., 2014;Guo et al., 2020). During the past years, strict
control of coal combustion has successfully reduced the $SO_2$ concentration, resulting in
a reduction of sulfate ($SO_4^{2-}$) component in $PM_{2.5}$; in stark contrast, the contributions
from organic and nitrate become increasingly more significant in China (Lang et al.,

2017).

The formation of secondary organic aerosol (SOA) starts from the gas-phase
oxidation of volatile organic compounds (VOCs) leading to various oxidized low-
volatility and semi-volatile products (Bianchi et al., 2019), followed by their
partitioning into the particle phase (Hallquist et al., 2009). Similarly, the formation of
nitrate aerosol in the daytime is largely due to the partitioning of gaseous nitric acid,
which is formed via the oxidation of $NO_2$ by OH (Seinfeld and Pandis, 2006;Wang et
al., 2019). It is traditionally believed that the wintertime atmospheric oxidation capacity
is weak due to the weak solar radiation, which limits the formation of SOA and nitrate
(Sun et al., 2013). However, it is very recently shown that the peak OH concentration
on polluted days in winter Beijing varies from $2\times10^6$ to $6\times10^6$ molecules $cm^{-3}$, which
is 6-10 times higher than what is predicted by the global model (Tan et al., 2018). This
discrepancy can be largely reduced after accounting for other OH production processes
in model simulations, which shows that the photolysis of nitrous acid (HONO)
dominates the initiation of $HO_x$ (OH and $HO_2$) and $RO_x$ (RO and $RO_2$) radical chain in
wintertime Beijing (Tan et al., 2018), and some other cities (Ren et al., 2006;Stutz et
al., 2013).
The HONO concentration has been measured with a wide rang from 0.18 to 9.71

ppbv at different locations, such as Beijing (Zhang et al., 2019d;Hu et al., 2002;Hendrick et al., 2014;Wang et al., 2017b), Shanghai (Wang et al., 2013;Zhang et al., 2019b), Guangdong (Hu et al., 2002;Su et al., 2008a), Hongkong (Xu et al., 2015), Shandong (Li et al., 2018), Xi'an (Huang et al., 2017b) and so on in China since 2000. More recently, modelling studies have suggested that nitrous acid (HONO) could enhance secondary aerosols formation in Beijing-Tianjin-Hebei (BTH) region (Zhang et al., 2019c), Pearl-River-Delta (PRD) region of China (Zhang et al., 2019a;Xing et al., 2019) and Houston (Czader et al., 2015). These results imply that the role of HONO in haze chemistry might be crucial in wintertime Beijing, while the direct evidence from observation has not been reported, yet. On the other hand, the HONO budget has been investigated via modelling studies (Liu et al., 2019c;Zhang et al., 2019c) and photostationary state calculations (Wang et al., 2017b;Li et al., 2018;Huang et al., 2017b;Lee et al., 2016;Oswald et al., 2015;Zhang et al., 2019d) at different locations. At the present time, the study of the HONO budget is still far from closed, which would require a significant effort on both the accurate measurement of HONO and the determination of related kinetic parameters for its production pathways (Liu et al., 2019c). For example, photo-enhanced conversion of $NO_2$ (Su et al., 2008b) and photolysis of particulate nitrate were found to be the two major mechanisms with large potential of HONO formation during noontime, but the associated uncertainty may reduce their importance (Liu et al., 2019c). The heterogeneous reactions of $NO_2$ on ground/aerosol surfaces were proposed to be an important HONO source during nighttime (Wang et al., 2017b;Zhang et al., 2019c) and daytime in Beijing-Tianjin-

Hebei (BTH) (Zhang et al., 2019c), but it was unimportant compared with the unknown
sources and the homogeneous reaction between NO and OH in Ji'an (Li et al., 2018) or
compared with the traffic emission on haze days in Beijing (Zhang et al., 2019d). The
traffic emission was found to be an important HONO source during nighttime and a
minor daytime HONO source in BTH (Zhang et al., 2019c). However, it was proposed
that direct emission of HONO from vehicles should contribute about 51.1 % (Meng et
al., 2019) and 52 % of HONO source on haze days in Beijing (Zhang et al., 2019d).
These results mean that more studies are still required on the HONO budget. In
particular, it is meaningful to analyze the HONO budget in polluted events for
understanding the possible influence of HONO sources on secondary pollutants
formation.
In this work, we carried out comprehensive measurements at a newly constructed
observation station (Aerosol and Haze Laboratory, Beijing University of Chemical
Technology, AHL/BUCT Station) located in the western campus of Beijing University
of Chemical Technology in downtown Beijing. We show observational evidence that
HONO has a prominent promotion effect on the secondary aerosol mass formation in
winter. Traffic-related emission seems to be a vital contributor to ambient HONO
during the pollution events in winter in Beijing.
**2. Materials and methods**
**2.1 Field measurements.** Field measurements were performed at AHL/BUCT Station
(Lat. 39º56′31″ and Lon. 116º17′52″) from February 1 to June 30, 2018. The
observation station is on a rooftop of the main building, which is 550 m from the 3$^{rd}$
ring road in the East, 130 m from the Zizhuyuan road in the North and 565 m from the
Nandianchang road in the West (Figure S1). The station is surrounded by both traffic
and residential emissions, thus, is a typical urban observation site.

Ambient air was sampled from the roof of the main building with five floors (~18

m above the surface). A $PM_{2.5}$ inlet (URG) was used to cut off the particles with
diameter larger than 2.5 μm before going to a Nafion dryer (MD-700-24, Perma Pure).
Then a Time-of-Flight Aerosol Chemical Speciation Monitor equipped a $PM_{2.5}$
aerodynamic lens (ToF-ACSM, Aerodyne) and an Aethalometer (AE33, Magee
Scientific) were connected to the manifold of aerosol sampling tube. The Reynolds
number in the aerosol sampling tube was 800 with the total flow rate of 16.7 lpm and
the residence time of 6.5 s. The details about ToF-ACSM measurement was described
in the Supplement information. Ambient air was drawn from the roof using a Teflon
sampling tube (BMET-S, Beijing Saak-Mar Environmental Instrument Ltd.) with the
residence time <10 s for gas phase pollutants measurements. Trace gases including NOx,
$SO_2$, CO and $O_3$ were measured with the corresponding analyzer (Thermo Scientific,
42i, 43i, 48i and 49i). Volatile organic compounds (VOCs) was measured using an
online Single Photon Ionization Time-of-flight Mass Spectrometer (SPI-ToF-MS
3000R, Hexin Mass Spectrometry) with unit mass resolution (UMR). The principle and
the configuration of the instrument has been described in detail elsewhere (Gao et al.,
2013) and the Supplement information. HONO concentration was measured using a
home-made Long Path Absorption Photometer (LOPAP) (Tong et al., 2016). The details
are described in the Supplement information. Particle size and number concentration
from 1 nm to 10 μm were measured with Scanning Mobility Particle Sizer (SMPS 3936,
TSI), particle size magnifier (PSM, Airmodus) and Neutral Cluster and Air Ion
Spectrometer (NIAS, Airel Ltd.). Meteorological parameters including temperature,
pressure, relative humidity (RH), wind speed and direction were measured using a
weather station (AWS310, Vaisala). Visibility and planetary boundary layer (PBL)
height were measured using a visibility sensor (PWD22, Vaisala) and a ceilometer
(CL51,Vaisala), respectively
**2.2 HONO budget calculation.** Multiple sources of ambient HONO have been
identified, such as emission from soil ($E_{soil}$) (Oswald et al., 2015;Meusel et al., 2018)
and vehicle exhaust ($E_{vehicle}$) (Trinh et al., 2017), production through homogeneous
reaction between NO and OH ($P_{NO\text{-}OH}$) in the atmosphere, photolysis of nitrate ($P_{nitrate}$)
(Bao et al., 2018), nitrous acid ($P_{HNO3}$) and nitrophenenol ($P_{nitrophenol}$) (Sangwan and
Zhu, 2018), heterogeneous reaction of $NO_2$ on aerosol surface ($P_{aerosol}$) (Liu et al., 2015)
and ground surface ($P_{ground}$) (Liu et al., 2019c;Li et al., 2018;Wang et al., 2017b).
However, the photolysis of $HNO_3$ and nitrophenol were excluded in this work because
they were believed as minor sources (Lee et al., 2016) and their concentrations were
unavailable during our observation. The removal pathways of HONO including
photolysis ($L_{photolysis}$), the homogeneous reaction with OH radical ($L_{HONO\text{-}OH}$) and dry
deposition ($L_{deposition}$) (Liu et al., 2019c) were considered.
The HONO budget could be calculated by,
$$\frac{dc_{HONO}}{dt} = E_{HONO} + P_{HONO} - L_{HONO} + T_{vertical} + T_{horizontal} \qquad (1)$$
where $\frac{dc_{HONO}}{dt}$ is the observed change rate of HONO mixing ratios (ppbv h$^{-1}$); $E_{HONO}$
represents the emission rate of HONO from different sources (ppbv h$^{-1}$); $P_{HONO}$ is the
in-*situ* production rate of HONO in the troposphere (ppbv h$^{-1}$); $L_{HONO}$ is the loss rate of
HONO (ppbv h$^{-1}$) (Li et al., 2018); $T_{vertical}$ and $T_{horizontal}$ are the vertical and horizontal
transport (Soergel et al., 2011), which can mimic source or sink terms depending on the
HONO mixing ratios of the advected air relative to that of the measurement site and
height (Soergel et al., 2011).
The emission rate ($E_{HONO}$, ppbv h$^{-1}$) was calculated based on the emission flux
($F_{HONO}=EI_{HONO}/A$, g m$^{-2}$ s$^{-1}$) and PBL height ($H$, m) according to the following equation,
$E_{HONO} = \frac{a \cdot F_{HONO}}{H}$  (2)
where, $EI_{HONO}$, is the emission inventory of HONO (g s$^{-1}$), $A$ is the urban area of Beijing
(m$^2$), α is the conversion factor ($\alpha = \frac{1 \times 10^9 \cdot 3600 \cdot R \cdot T}{M \cdot P} = \frac{2.99 \times 10^{13} \cdot T}{M \cdot P}$), $M$ is the molecular
weight (g mol$^{-1}$), $T$ is the temperature (K) and $P$ is the atmospheric pressure (Pa).
The production rates of HONO ($P_{HONO}$, ppbv h$^{-1}$) in the troposphere was calculated
by,
$P_{HONO} = 3600 \cdot k_1 \cdot c_{precursor}$   (3)
where, $k_1$ is the quasi first-order reaction rate constant (s$^{-1}$), $c_{precursor}$ is the concentration
of precursor (ppbv). For homogeneous reaction between NO and OH,
$k_1 = k_2 \cdot c_{OH}$   (4)
where, $k_2$ is the second-order reaction rate constant ($7.2 \times 10^{-12}$ cm$^3$ molecule$^{-1}$ s$^{-1}$) (Li et
al., 2012), $c_{OH}$ is the OH concentration (molecules cm$^{-3}$). For heterogeneous reaction,
$k_1 = \frac{\gamma \cdot A_s \cdot \omega}{4} \cdot Y_{HONO}$  (5)
where, $A_s$ is the surface area concentration of the reactive surface (m$^2$ m$^{-3}$), $\omega$ is the
molecular mean speed (m s$^{-1}$), $\gamma$ is the uptake coefficient of the precursor, $Y_{HONO}$ is the
yield of HONO. For ground surface, the surface area concentration is
$A_s = \frac{\delta}{H}$ (6)
where $\delta$ is the surface roughness, which is calculated according to the mean project area,
perimeter and height of the buildings in Beijing.
$\delta = \frac{f_{building}*(A_{projected}+h*P_{building})}{A_{projected}} + f_{blank}$ (7)
where $f_{building}$ (0.31) and $f_{blank}$ (0.69) are the fraction of the projected area ($A_{projected}$) of
buildings and blank space, respectively; $P_{building}$ and $h$ are the perimeter and the height
of the building, respectively. The $f_{building}$ and $P_{building}$ are measured from ~1000 buildings
randomly selected on the Google Map using ImageJ software. The mean height (44.5
m) of the building in Beijing is linearly extrapolated from the literature data based on
remote measurement using Light Detection and Ranging (LiDAR) sensor from 2004 to
2008 (Cheng et al., 2011). The $\delta$ in Beijing is calculated to be 3.85, which is slightly
higher than the value (2.2) used by Li et al. (2018).
As for photolysis reaction, the first-order reaction rate was
$k_1 = J$ (8)
where, $J$ is the photolysis rate to produce HONO (s$^{-1}$).
The loss rates of HONO by photolysis ($L_{photolysis}$), homogeneous reaction with
OH radicals ($L_{HONO\text{-}OH}$) and dry deposition ($L_{deposition}$) (Liu et al., 2019c) were calculated
according to the following equations.
$L_{photolysis} = 3600 \cdot J_{HONO} \cdot c_{HONO}$ (9)
$L_{HONO-OH} = 3600 \cdot k_{HONO-OH} \cdot c_{OH} \cdot c_{HONO}$ (10)
$$L_{deposition} = \frac{3600 \cdot v_d \cdot c_{HONO}}{H} \quad (11)$$
where, $J_{HONO}$ is the photolysis rate of HONO (s[-1]), $k_{HONO-OH}$ is the second-order reaction
rate constant between HONO and OH ($6 \times 10^{-12}$ cm$^3$ molecule$^{-1}$ s$^{-1}$) (Atkinson et al.,
2004), and $v_d$ is the dry deposition rate of HONO (0.001 m s$^{-1}$) (Han et al., 2017).

Vertical transport by advection ($T_{vertical}$), which is an important sink of HONO in

the night (Gall et al., 2016;Meng et al., 2019), can be calculated according to equation

(12).

$$T_{vertical} = -K_h(z,t)\frac{\partial c(z,t)}{\partial z}\frac{1}{h} \quad (12)$$
where $K_h(z,t)$ is the eddy diffusivity of heat (m$^2$ s$^{-1}$) at height $z$ (m) and time $t$, $h$ is the
height of the second layer (18 m in this study) (Gall et al., 2016). On the other hand,
both the vertical and horizontal transport can be estimate according to Eq. (13),
$$T_{vertical} = k_{dilution}(c_{HONO} - c_{HONO,background}) \quad (13)$$
where $k_{dilution}$ is a dilution rate (0.23 h$^{-1}$, including both vertical and horizontal transport)
(Dillon et al., 2002), $c_{HONO}$ and $c_{HONO,background}$ is the HONO concentration at the
observation site and background site, respectively (Dillon et al., 2002).

In addition, even though all the current known sources had been considered in

models, the modelled daytime HONO concentrations were still lower than the observed
concentration (Tang et al., 2015;Michoud et al., 2014). Therefore, the HONO
concentration could be described in equation (14).
$$\frac{dc_{HONO}}{dt} = E_{soil} + E_{vehicle} + P_{NO-OH} + P_{nitrate} + P_{aerosol} + P_{ground} + P_{unknown} -$$
$$L_{photolysis} - L_{HONO-OH} - L_{deposition} + T_{vertical} + L_{horizontal} \quad (14)$$
**3. Results and discussion**
**3.1 Overview of the air pollution.** The mass concentration of non-refractory $PM_{2.5}$
($NR$-$PM_{2.5}$) and HONO along with metrological parameters are shown in Fig. 1. The
time series of other pollutants ($SO_2$, CO, $O_3$, benzene, toluene and black carbon) are
shown in Fig. S2 in the Supplement information.

Similar to previous measurements (Guo et al., 2014;Wang et al., 2016), the air

pollution events showed a periodic cycle of 3-5 days during the observation, as
indicated by the concentration of $NR$-$PM_{2.5}$ (Fig. 1A), gaseous pollutants and the
visibility. During the observation period, 20-60% of hourly $PM_{2.5}$ concentration was
higher than 75 μg m$^{-3}$ (the criterion for pollution according to the national air quality
standards) in each month (Fig. S3A). Both the frequency of severe polluted episodes
and the mean mass concentration of $PM_{2.5}$ and $NR$-$PM_{2.5}$ were obviously higher in
March than that in the rest months (Fig. 1 and S3). This can be explained by both the
intensive emission during the heating season as evidenced by the high concentration of
primary pollutants including CO, $SO_2$ and BC (Table S1) and the stagnant
meteorological conditions supported by the low wind speed ($<2$ m s$^{-1}$) and the low
planetary boundary layer (PBL) height, in particular, in March (Fig. S4A).

OA and nitrate dominated the $NR$-$PM_{2.5}$, while their relative contribution varied

significantly during the observation (Fig. 1B and Table S1). This is similar to the
previously reported $NR$-$PM_{1.0}$ composition (Sun et al., 2015). The monthly mean
fraction of OA varied from 45.9±10.2 % to 52.6±18.7 %, which was accompanied by a
slight increase of sulfate from 16.0±9.1 % to 18.2±8.0 % (Fig. S4D). At the same time,
the monthly mean fraction of nitrate and chloride decreased from 26.7±8.8 % to
16.7±12.8 % and from 7.7±6.1 % to 0.3±0.2 %, respectively. Ammonium showed a
peak value (14.2±2.8 %) in March, then slightly decreased to 12.2±5.2 %. The intensive
emission of chloride from coal combustion during heating season (Cho et al., 2008) and
firework burning (Zhang et al., 2017), which was transported from Tangshan during
Chinese New Year (Fig. S5A and B), led to high fraction of chloride in February and
March. The decrease in nitrate and ammonium fractions from February to June should
be related to the increase in temperature (Fig. S2) which was in favor of $NH_4NO_3$
decomposition (Wang et al., 2015). Besides the reduction of the contribution from other
components, secondary formation due to increased UV light (Fig. S4C) might also
favor the increased OA fraction (Huang et al., 2014). This means that chemical
transformation in March should still be vigorous although the UV light intensity in
March is lower than in summer (Fig. S4C). It also implies other factors may compensate
the weak UV light intensity in March.
HONO, which has been recognized as the important precursor of primary OH
radical (Ren et al., 2006;Alicke et al., 2003), ranged from 0.05 to 10.32 ppbv from
February 1 to June 30, 2018 (Fig. 1C) with the mean value of 1.26±1.06 ppbv. In winter
(February and March), HONO concentration was 1.15±1.10 ppbv and comparable to
the previous results (1.05±0.89 ppbv) measured in the winter of Beijing (Wang et al.,
2017b;Hou et al., 2016), while it was slightly lower than that from April to June
(1.35±1.11 ppbv) in this work and those measured in the summer of Shanghai (2.31
ppbv, in May) (Cui et al., 2018) and Guangzhou (2.8 ppbv, in July) (Qin et al., 2009).
The mean HONO concentration in March (1.53±1.25 ppbv) was higher than that in
February and April (Fig. S3D), while was slightly higher or close to that in May and
June. Chamber studies have found that HONO is responsible for the initiation of
photosmog reactions (Rohrer et al., 2005). It is reasonable to postulate that HONO
probably play an important role in the secondary chemistry of particle formation in
March.
**3.2 Promotion effect of HONO photolysis on aerosol formation in winter.** Oxidation
of precursors by OH radicals is the main mechanism regarding to secondary aerosol
formation in the troposphere. After partially ruling out the possible influence of PBL
variation by normalizing the concentrations of all pollutants to CO (Cheng et al., 2016)
or BC (Liggio et al., 2016), we found all secondary species including sulfate, nitrate
and ammonium show obvious daytime peaks from 7:00 am to 6:00 pm (Figure S5C)
(Cheng et al., 2016). The similar trends were observed after the concentrations of
pollutants were normalized to BC (not shown). This suggests they might connect with
photochemistry.
Photolysis of $H_2O_2$, HCHO, $O_3$ and HONO, and the reaction between NO and $HO_2$
are known as sources of OH radical in the atmosphere (Alicke et al., 2003;Volkamer et
al., 2010;Tan et al., 2018;Tang et al., 2015). In this work, the concentration of $H_2O_2$,
HCHO and $HO_2$ are unavailable. Thus, their contributions to OH production were not
discussed here. However, it has been well recognized that the photolysis of HONO is
the dominant source of OH in the dawn and dusk period (Holland et al., 2003), even
contributes up to 60% of daytime OH source in winter (Spataro et al., 2013;Rohrer et
al., 2005). In addition, it has been confirmed that HONO dominates the primary OH
source at various locations (Tan et al., 2018;Liu et al., 2019c;Tan et al., 2017;Aumont
et al., 2003). Therefore, it is meaningful to discuss the contribution of HONO to
secondary aerosol formation through OH production. We simply compared the OH
production via photolysis of HONO ($P_{OH\text{-}HONO}=J_{HONO}\times c_{HONO}$) and $O_3$ ($P_{OH\text{-}}$
$_{O3}=J_{O1D}\times c_{O3}$) in Fig. 2 when the $PM_{2.5}$ concentration was larger than 50 μg m$^{-3}$ and the
RH was less than 90 % to understand the chemistry in pollution events. Under these
conditions, local chemistry should be more important as 75 % of the wind speed was
less than 1.0 m s$^{-1}$ (Fig. S6). The details about the $J_{HONO}$ and $J_{O1D}$ calculation were
shown in the Supplement information and their time series were shown in Fig. S7. On
polluted days in winter, the daytime $P_{OH\text{-}HONO}$ was always significantly higher than the
$P_{OH\text{-}O3}$ in winter and the maximal $P_{OH\text{-}HONO}$ and $P_{OH\text{-}O3}$ were $1.73\pm0.86\times10^7$ molecules
cm$^{-3}$ s$^{-1}$ ($2.43\pm1.21$ ppb h$^{-1}$) and $1.03\pm1.06\times10^7$ molecules cm$^{-3}$ s$^{-1}$ ($1.45\pm1.49$ ppb h$^{-}$
$^1$), respectively (Fig. 2A). Owing to the high HONO concentration accumulated
throughout the night, the maximal $P_{OH\text{-}HONO}$ in winter was as about 2-6 times of that
was observed in the wintertime of Colorado, USA (~0.59 ppb h$^{-1}$) (Kim et al., 2014),
New York, USA (~0.40 ppb h$^{-1}$) (Kanaya et al., 2007) and Nanjing, China ($0.90\pm0.27$
ppb h$^{-1}$) (Liu et al., 2019b). In the period from April to June, the daily maxima of $P_{OH\text{-}}$
$_{HONO}$ and $P_{OH\text{-}O3}$ were $2.48\pm1.42\times10^7$ molecules cm$^{-3}$ s$^{-1}$ ($3.48\pm1.99$ ppb h$^{-1}$) and
$6.51\pm4.17\times10^7$ molecules cm$^{-3}$ s$^{-1}$ ($9.15\pm5.86$ ppb h$^{-1}$), respectively. These results
mean that the photolysis of HONO should play an important role in the initiation of the
daytime $HO_x$ and $RO_x$ chemistry on polluted days in winter, while photolysis of $O_3$
becomes more important from April to June. This is consistent with the previous
findings that HONO photolysis dominants the primary OH source in winter of BTH
(Xing et al., 2019;Tan et al., 2018), Colorado and New York City (Ren et al., 2006;Kim
et al., 2014), while photolysis of $O_3$ and HCHO related reactions usually dominated
primary OH production in summer (Alicke et al., 2003).
Oxidation of trace gas pollutants, in particular VOCs, by OH is their main removal
pathway in the troposphere (Atkinson and Arey, 2003), subsequently, contribute to
secondary aerosol formation (Kroll and Seinfeld, 2008). A very recent work has found
that oxidation of VOCs from local traffic emission is still efficient even under pollution
conditions (Guo et al., 2020). We partially ascribe this to the high HONO concentration
in winter Beijing. To confirm this assumption, 12 episodes in winter were chosen (Fig.
1) to uncover the connection between aerosol formation and HONO photolysis. The 1st,
3rd and 5th episodes were clean days and the other 9 episodes were typical haze events
with duration above 2 days. The features of these episodes were summarized in Table
S2. Fig. 2C shows the CO-normalized daytime profiles of OA and HONO in the 7th and
12th episodes as two examples. In all selected cases, HONO exhibited quick reduction
due to the photolysis after sunrise, and simultaneously, OA concentration started to
increase. This is similar to the evolution of the concentration of pollutants in a typical
smog chamber experiment. We further show the formation of OA ($\Delta C_{OA}/C_{CO}$) as a
function of the consumed HONO ($-\Delta C_{HONO}/C_{CO}$) in Fig. 2D. Except for the 4th episode
that was highly affected by firework emission during the Spring Festival, $\Delta C_{OA}/C_{CO}$
showed a linear dependence on $-\Delta C_{HONO}/C_{CO}$ in winter, and the correlation coefficient
was 0.75. As the meteorological condition was stagnant during these cases as indicated

by the low wind speed (< 1.0 m s$^{-1}$, Fig. S5D), it was reasonable to ascribe the increase

of OA concentration to local secondary formation initiated by OH radical and

photolysis of HONO should play an important role in initiation the $HO_x$ and $RO_x$

chemistry. This kind of correlation could not be seen for the pollution events from April

to June because the primary OH production was no longer dominated by HONO

photolysis as indicated by Fig. 2D. It should be noted that oxidation of biogenic alkenes

by $O_3$ might also contribute to OA formation. However, anthropogenic VOCs instead

of biogenic VOCs dominated the wintertime VOCs in Beijing (Liu et al., 2017a).

Although vehicles can emit isoprene (Zou et al., 2019), the contribution of isoprene to

the observed increase of OA concentration should be unimportant due to the low

concentration of isoprene in winter (Zou et al., 2019). Therefore, it is reasonable to

conclude that the increase of OA concentration in daytime might be mainly resulted

from oxidation of VOCs by OH.

Similar to OA, $\Delta C_{nitrate}/C_{CO}$ in winter also showed good linear correlation with -

$\Delta C_{HONO}/C_{CO}$ (R=0.67, Fig. S5E), suggesting that the increase of particle-phase nitrate

in the daytime should also be promoted by OH radical from HONO photolysis.

Interestingly, $\Delta C_{ammonium}/CO$ also showed a good correlation with $-\Delta C_{HONO}/C_{CO}$

(R=0.61, Fig. S5E), although particle-phase ammonium should not be directly related

to oxidation of $NH_3$ by OH. We explained the increased ammonium as the result of

enhanced neutralization of $HNO_3$ by $NH_3$ (Wang et al., 2018;Wen et al., 2018;Sun et

al., 2018) because $NH_4^+$ was adequate to neutralize both sulfate and nitrate as shown in

Fig.S8. This was consistent with the recent work which observed the important role of

photochemical reactions in daytime nitrate formation, while hydrolysis of $N_2O_5$ mainly
contributed to nighttime nitrate (Tian et al., 2019). Although a recent work has found
that daytime hydrolysis of $N_2O_5$ on hygroscopic aerosols is also an important source of
daytime nitrate in winter Beijing (Wang et al., 2017a), the linearly correlation between
$\Delta C_{nitrate}/C_{CO}$ and $\Delta C_{HONO}/C_{CO}$ at least implies that the promotion effect of HONO on
nitrate formation could not be excluded. On the other hand, the correlation between
$\Delta C_{sulfate}/C_{CO}$ and $-\Delta C_{HONO}/C_{CO}$ was much weaker (R=0.26), suggesting a weak
connection between particle-phase sulfate and gas-phase $H_2SO_4$. This was also
consistent with the previous understanding that heterogeneous reactions of $SO_2$ were
the dominant pathway for sulfate formation (Zheng et al., 2015a;He et al., 2018;Zhang
et al., 2020). Overall, this work well supported the recent modeling results that HONO
could obviously promote the aerosol production in winter (Zhang et al., 2019a;Zhang
et al., 2019c;Xing et al., 2019;An et al., 2013) from the point of view of observation.
**3.3 HONO budget in polluted events.** To understand the possible sources of HONO
in polluted events in winter, the HONO budget was calculated for the events when the
$PM_{2.5}$ concentration was larger than 50 µg m$^{-3}$ and the RH was less than 90 % according
to the method described in Section 2.2.
**Vehicle emission.** The $E_{vehicle}$ was calculated according to Eq. (2) using the relative
emission rate of HONO to $NO_x$ and the emission inventory of $NO_x$ from vehicles. Firstly,
the ratio of HONO/$NO_x$ was calculated according to the method reported by Xu et al.
(Xu et al., 2015) and Li et al. (Li et al., 2018) from the fresh nighttime plumes which
were strictly satisfy the following criteria: 1) $NO_x > 45$ ppb (highest 25% of $NO_x$ data);
2) $\Delta NO/\Delta NO_x > 0.8$, with good correlation between NO and $NO_x$ (R > 0.9, P < 0.05);
3) Good correlation between HONO and $NO_x$ ($R^2$>0.65, P < 0.05); and 4) Dataset from
5:00 am to 8:00 am. The mean emission ratio of HONO to $NO_x$ was 1.8±0.5% based
on 5 fresh vehicle exhaust plumes during our observation (Table S3). This value is
higher than that in Hongkong (1.2±0.4%) (Xu et al., 2015), Beijing (1.3%) (Zhang et
al., 2019d) and Jinan (0.53±0.20%) (Li et al., 2018) using the same method, while is
comparable with the result measured in tunnel experiments (2.1%) carried out in
Beijing (Yang et al., 2014). Secondly, low HONO concentration should be companied
with high $NO_x$ and high ratio of $\Delta NO/\Delta NO_x$ if direct emission from vehicles was the
major source of HONO and the source from secondary formation was negligible in the
urban atmosphere. Therefore, we further estimated the HONO/$NO_x$ ratio using a low
limit correlation method (Li et al., 2012). In the 2D space of HONO verse $NO_x$ (Fig.
S8), the lowest marge with $\Delta NO/\Delta NOx$ larger than 0.8 were chosen for linear
correlation. The ratio of $\Delta HONO/\Delta NOx$ is 1.17±0.05%. This value is lower than that
estimated through empirical method discussed above, while is very close to that
measured in Hongkong (1.2±0.4%) (Xu et al., 2015) and (1.23±0.35%) (Liang et al.,
2017), Guangzhou (1.0%) (Li et al., 2012) and Beijing (1.3% and 1.41 %) (Zhang et al.,
2019d;Meng et al., 2019). Finally, several studies have measured the direct emission of
HONO from vehicle exhaust. The HONO/$NO_x$ was 0.18% from gasoline cars through
chassis dynamometer tests in China (Liu et al., 2017d), while it was 0-0.95% for
gasoline vehicles and 0.16-1.0 % for diesel vehicles measured under real-world driving
test cycles in Japan (Trinh et al., 2017). Thus, three levels of vehicle emission factor
were considered. 1.17±0.05% was taken as the middle value which was very close to
the mean emission ratio (1.21) for all of these reported values in China (Li et al.,
2018;Xu et al., 2015;Yang et al., 2014;Liu et al., 2017d;Gall et al., 2016;Meng et al.,
2019), while 0.18% (Liu et al., 2017d) and 1.8 % were the lower limit and the upper
limit, respectively.

The $E_{vehicle}$ was calculated using the hourly $NO_x$ emission inventory from vehicles

in Beijing (Yang et al., 2019) after converted to emission flux of HONO ($F_{HONO}=F_{NOx}\times$
$HONO/NO_x$) and the PBL height as described in Section 2.2. Thus, the calculated
emission rate reflected the diurnal variation of both the emission inventory and the PBL
height. The calculated hourly middle value of $E_{vehicle}$ using the $HONO/NO_x$ of 1.17%
was from 0.085±0.038 to 0.34±0.15 ppbv h$^{-1}$, which was slightly higher than the
daytime emission rate of HONO in Xi'an (Huang et al., 2017b). This is reasonable
when the vehicle population in Beijing is taken into consideration. The lower limit of
$E_{vehicle}$ was 0.013±0.006-0.053±0.023 ppbv h$^{-1}$, which was close to the estimated
emission rate of HONO in Jinan (Li et al., 2018). The upper limit was in the range of
0.13±0.06-0.53±0.23 ppbv h$^{-1}$.
**Soil emission.** The emission flux of HONO from soil depends on the water content, the
nitrogen nutrient content and the temperature of soil (Oswald et al., 2013). Oswald et
al. (2013) measured the emission flux of HONO from 17 soil samples, including
eucalyptus forest, tropical rain forest, coniferous forest, pasture, woody savannah,
grassland, stone desert, maize field, wheat field, jujube field an cotton field etc. Tropical
rain forest, coniferous forest and grassland are the typical plants in downtown Beijing
(Huang et al., 2017a). At the same time, their emission fluxes of HONO are comparable
(Oswald et al., 2013). Thus, we used the emission flux from grassland to calculate the
emission rate of HONO from soil in Beijing because the temperature and water holding
content dependent emission flux of HONO was available for grassland soil. Three
levels of water content including 25-35%, 35-45% and 45-55% were considered. The
temperature dependence of $F_{HONO}$ was calculated using the mean value of the $F_{HONO}$
with different water content, while the low limit and upper limit of $F_{HONO}$ were
calculated using the emission flux from 45-55% of water content and 25-35% of water
content, respectively. The lower limit, the middle value and the upper limit of the $E_{soil}$
are 0.0032±0.0027-0.013±0.014, 0.0046±0.0039-0.020±0.20 and 0.0057±0.0047-
0.025±0.024 ppbv h$^{-1}$, respectively, calculated according to Eq. (2).
**Homogeneous reaction between NO and OH.** Direct measurement of OH
concentration was unavailable in this work, while several methods were used to
estimate the ambient OH concentration. In winter in Beijing, it has been found that the
OH concentration is linearly correlated with $J_{O1D}$, that's, $c_{OH}=J_{O1D}\times2\times10^{11}$ molecules
cm$^{-3}$ (Tan et al., 2019). However, Tan et al. (2018) reported a larger conversion factor
($4.33\times10^{11}$ molecules cm$^{-3}$). Li et al. (2018) estimated the OH radical concentration
considering both photolysis rate and NO$_2$ concentration, namely,
$$c_{OH} = \frac{4.1\times10^9\times(J_{O1D})^{0.83}\times(J_{NO_2})^{0.19}\times(140c_{NO_2}+1)}{0.41c_{NO_2}^2+1.7c_{NO_2}+1} \quad (15)$$
Overall, the estimated OH concentrations according to Eq. (15) were comparable with
that estimated by Tan et al. (2019) (Fig. S10C). The method for the photolysis rates
calculation were shown in the SI and the time series of the photolysis rates were shown

in Fig. S7. On polluted days, high concentration of $NO_2$ resulted into lower OH

concentrations estimated using the Eq. (15). Therefore, the corresponding $P_{NO-OH}$ was

taken as the low limit for homogeneous reaction between NO and HONO because

polluted events were discussed in this work, while $P_{NO-OH}$ calculated using the OH

concentration ($J_{O1D} \times 4.33 \times 10^{11}$ molecules cm$^{-3}$) (Tan et al., 2018) was taken as the upper

limit and $P_{NO-OH}$ calculated using the OH concentration ($J_{O1D} \times 2 \times 10^{11}$ molecules cm$^{-3}$)

(Tan et al., 2019) was the middle value. In the night, OH concentration usually varied

from $1.0 \times 10^5$ molecules cm$^{-3}$ (Li et al., 2012;Tan et al., 2018) in winter to $5 \times 10^5$

molecules cm$^{-3}$ in summer (Tan et al., 2017). The nighttime OH concentration was

estimated linearly correlated with the product of nighttime $O_3$ concentration and alkenes

concentration, namely,

$$c_{OH,night} = 1 \times 10^5 + 4 \times 10^5 \times \frac{(c_{O_3} \times c_{alkenes})_{night} - (c_{O_3} \times c_{alkenes})_{night,min}}{(c_{O_3} \times c_{alkenes})_{night,max} - (c_{O_3} \times c_{alkenes})_{night,min}} \quad (16)$$

The time series of OH concentration calculated using different methods was shown in

Fig. S11. Thus, the lower limit, the middle value and the upper limit of $P_{NO-OH}$ were

0.007±0.019-0.43±0.26, 0.026±0.053-0.99±0.79 and 0.028±0.053-2.14±1.71 ppbv h$^{-1}$,

respectively, calculated according to Eqs. (3) and (4). The calculated middle value of

$P_{NO-OH}$ (with mean daytime value of 0.49±0.35 ppb h$^{-1}$) was comparable with these

estimated values by Li et al. (2018) (0.4 ppb h$^{-1}$) and Huang et al. (2017b) (0.28 ppb h$^{-1}$). It should be noted that measured NO concentration was used to calculate the $P_{NO-OH}$.

Besides vehicle emission, power plant and industries also contribute NO emission. 40 %

of NOx was from vehicle emission according to the emission inventory of $NO_x$ in

Beijing (He et al., 2002).

It should be noted that OH concentration was estimated based on $J_{O1D}$ (Tan et al.,

2019;Tan et al., 2018) or $J_{O1D}$ and $J_{NO2}$ (Li et al., 2018). As discussed in Section 3.2,
HONO was an important primary OH source in the daytime. Unfortunately, it could not
be parameterized for calculating OH concentration because the measured or modelled
OH concentration was unavailable in this work. This might underestimate the early
daytime OH concentration, subsequently, the contribution of homogeneous reaction of
NO with OH to HONO source. This need to be further investigated in the future.
**Photolysis of nitrate.** A recent work reported the photolysis rate of nitrate ($J_{nitrate}$) in
ambient PM$_{2.5}$ at a solar zenith angle of 0° (Bao et al., 2018). The $J_{nitrate}$ varied from
$1.22 \times 10^{-5}$ to $4.84 \times 10^{-4}$ s$^{-1}$ with the mean value of $8.24 \times 10^{-5}$ s$^{-1}$. These values were
further normalized according to the zenith angle and UV light at our observation station
to calculate the low limit, the upper limit and the middle $J_{nitrate}$. The time series of the
measured nitrate concentration and the middle value of $J_{nitrate}$ were shown in Fig. 1 and
Fig. S7, respectively. Therefore, the corresponding daytime lower limit, the middle
value and the upper limit of HONO from photolysis of nitrate were 0.0011±0.0021-
0.096±0.092, 0.0072±0.0021-0.66±0.092 and 0.042±0.082-3.86±0.008 ppbv h$^{-1}$,
respectively, calculated in the light of Eqs. (3) and (8).
**Heterogeneous reactions of NO$_2$ on aerosol and ground surface.** The production of
HONO from heterogeneous reactions of NO$_2$ on aerosol surface was calculated
according to Eqs. (3) and (5). The aerosol surface concentration was measured with a
SMPS. The uptake coefficient ($\gamma$) of NO$_2$ on different particles varied from $5 \times 10^{-9}$ to
$9.6 \times 10^{-6}$ (Ndour et al., 2009;Underwood et al., 2001;Underwood et al., 1999), while it
was recommended to be $1.2 \times 10^{-8}$ (Crowley et al., 2010), which was used to calculate
the $P_{aerosol}$ in the base case. It has been found that the γ highly depends on the relative
humidity (RH). The low limit bound of $P_{aerosol}$ was calculated based on the RH
dependent uptake coefficient of $NO_2$ on kaolinite ($\gamma_{NO2}=4.47 \times 10^{39}/(1.75 \times 10^{46} + 1.93$
$\times 10^{45}RH)$, while the upper limit of $P_{aerosol}$ was calculated according to the RH
dependent γ on hematite ($\gamma_{NO2}=4.46 \times 10^{39}/(6.73 \times 10^{44} + 3.48 \times 10^{44} RH)$ (Liu et al.,
2015). Heterogeneous reaction of $NO_2$ on black carbon (BC) was also considered in the
night. The surface area concentration of BC was calculated according to its specific
area ($87 \, m^2 \, g^{-1}$) (Su et al., 2018) and the measured mass concentration. The $\gamma_{NO2}$ on BC
is $1.17 \times 10^{-5}$, with a HONO yield of 0.8 (Han et al., 2013). The light enhanced uptake γ
of $NO_2$ ($1.9 \times 10^{-6}$) on mineral dust was further parameterized (Ndour et al., 2008) after
normalized to the solar radiation intensity in Beijing.
The contribution of heterogeneous reaction of $NO_2$ on ground surface was
calculated similar to that on mineral dust. The same kinetics for heterogeneous reaction
of $NO_2$ on aerosol surface were used to calculate the nighttime contribution of ground
surface. A recent work observed a significant enhancement of $NO_2$ and HONO
formation by UV light on the real urban grime (Liu et al., 2019a). Thus, RH dependent
kinetic data measured on urban grime ($\gamma_{NO2}=7.4 \times 10^{-7}+5.5 \times 10^{-8} RH$) was used to
calculate the daytime upper limit for heterogeneous uptake of $NO_2$ on the ground
surface. The $A_s$ of aerosols varied from $1 \times 10^{-4}$ to $4.8 \times 10^{-3} \, m^{-1}$ with a mean value of
$1.4 \pm 0.5 \times 10^{-3} \, m^{-1}$ during pollution events. This value is comparable with that used in
modeling studies (Zhang et al., 2016;Aumont et al., 2003). The $A_s$ of ground surface
which was calculated according to Eq. (6) and (7) varied from $1.5 \times 10^{-3}$ to $3.85 \times 10^{-2}$ m$^{-}$
$^1$ with a mean value of $1.3 \pm 0.9 \times 10^{-2}$ m$^{-1}$ during pollution events. The surface roughness
was 3.85 calculated according to Eq. (7). The $Y_{HONO}$ was set to 0.5 because of the
hydrolysis reaction of $NO_2$ (Liu et al., 2015), while it was 0.8 for light enhanced
reaction (Liu et al., 2019a;Ndour et al., 2008) and on BC (Han et al., 2013).
The lower limit, the middle value and the upper limit of $P_{aerosol}$ were $0.038 \pm 0.030$-
$0.087 \pm 0.072$, $0.038 \pm 0.030$-$0.088 \pm 0.072$ and $0.041 \pm 0.032$ - $0.092 \pm 0.073$ ppbv h$^{-1}$,
respectively. The corresponding values were $0.00027 \pm 0.00017$-$0.0020 \pm 0.0012$,
$0.0014 \pm 0.00095$-$0.0089 \pm 0.006$ and $0.0025 \pm 0.0023$-$0.060 \pm 0.032$ ppbv h$^{-1}$ for $P_{ground}$.
Although the $A_s$ of ground surface was higher than that of aerosol, the larger $\gamma_{NO2}$
($1.17 \times 10^{-5}$) on soot particles than that on other aerosols and ground surface led to a
larger production rate of HONO in this work. The $P_{aerosol}$ calculated in this work was
on the same orders as soil emission, while it was lower than the $P_{aerosol}$ estimated by
Huang et al. (Huang et al., 2017b) because different calculation methods have been
used. In their work, the production rate of HONO was estimated based on the
conversion rate (Huang et al., 2017b), whilst it was calculated based on the measured
aerosol surface area concentration and uptake coefficient of $NO_2$ on different particles
in this work.
It should be pointed out that HONO production from heterogeneous reaction of
$NO_2$ on both aerosol and ground surface greatly depend on the $\gamma_{NO2, BET}$ and $A_s$. The $A_s$
of aerosols was comparable with the modeling input. However, the small nighttime $\gamma_{NO2,}$
$_{BET}$ ($10^{-8}$ - $10^{-7}$) on dust were used in this work rather than the $\gamma_{NO2, BET}$ ($1 \times 10^{-6}$) used in
modelling studies (Zhang et al., 2016;Aumont et al., 2003;Gall et al., 2016). This leads
to a lower production rate of HONO from heterogeneous reaction of $NO_2$ on aerosols.
As for heterogeneous reaction of $NO_2$ on ground surface, besides the small $\gamma_{NO2, BET}$
used in this work, the $A_s$ of ground surface (0.0015 to 0.0385 $m^{-1}$) calculated using the
surface roughness and PBL height was also significantly lower than the fixed value of
0.3 $m^{-1}$ in modeling studies that might overestimate the contribution of HONO
production from heterogeneous reaction of $NO_2$ on ground surface. It should be noted
that the initial uptake coefficient ($\gamma_{ini}$) was parameterized in this work. This will
overestimate the contribution of heterogeneous reaction of $NO_2$ to HONO source
because the steady-state uptake coefficient is usually one order of magnitude lower than
$\gamma_{ini}$ (Han et al., 2013;Liu et al., 2015). These results mean that heterogeneous reaction
might not be a major HONO source. This is consistent with a recent work that found
heterogeneous reaction being unimportant when compared with traffic emission during
haze events in winter in Beijing (Zhang et al., 2019d).
**Sinks of HONO.** The loss rates of HONO by photolysis ($L_{photolysis}$), homogeneous
reaction with OH radicals ($L_{HONO-OH}$) and dry deposition were calculated according to
Eqs. (9)-(11). The daytime $J_{HONO}$ varied from $1.71\times10^{-5}$ to $1.13\times10^{-3}$ $s^{-1}$ on polluted
days in winter, while it was in the range of $5.89\times10^{-5}$ to $1.53\times10^{-3}$ $s^{-1}$ from April to June.
These values are comparable to modelling results ($3.9\times10^{-5}$-$1.8\times10^{-3}$ $s^{-1}$) (Gall et al.,
2016). The daytime $L_{photolysis}$ were in the range of 0.03-5.23 ppb $h^{-1}$ and 0.25-7.10 ppb
$h^{-1}$ in winter and the rest months, respectively. It was the major sink of HONO in the
daytime. The $L_{HONO-OH}$ varied from 0.0049 to 0.069 ppbv $h^{-1}$ in winter using the $k_{HONO-}$
$_{OH}$ of $6 \times 10^{-12}$ cm$^3$ molecule$^{-1}$ s$^{-1}$ (Atkinson et al., 2004) and the middle value of OH
concentrations. It was from 0.0050 to 0.085 ppbv h$^{-1}$ from April to June. The $L_{deposition}$
was in the range of 0.004-0.056 ppbv h$^{-1}$ in winter and 0.004-0.030 ppbv h$^{-1}$ from April
to June, calculated according to Eq. (11).

As pointed in Section 2.2, vertical transport by advection is an important nocturnal

sink of HONO (Gall et al., 2016). In this work, the vertical distribution of HONO
concentration is unavailable. Recently, Meng et al. (2019) measured the vertical
distribution of HONO in Beijing in December, 2016. The concentration of HONO
showed nearly flat profiles from ground level to 240 m in pollution events after sunset,
while negative profiles of HONO were observed in pollution events during night (Meng
et al., 2019). The nighttime concentration gradient was 0.0047±0.0025 ppb m$^{-1}$ derived
from the nighttime dataset (Meng et al., 2019). In the daytime, we assume a zero
concentration gradient. On the other hand, the eddy diffusivity of heat in urban
environment was measured in New Delhi, Indian (Yadav et al., 2003). Using their
dataset with the wind speed lower than 2.0 m s$^{-1}$, we derived the relationship between
the $K_h$ and the wind speed (WS) ($K_h$=0.9389×WS-0.3374 m$^2$ s$^{-1}$). The nighttime $T_{vertical}$
changed from 0.15 to 0.37 ppbv h$^{-1}$ in winter, while it was from 0.12 to 0.68 ppbv h$^{-1}$
according to Eq. (12) from April to June. Because the wind speed was usually lower
than 1.0 m s$^{-1}$ in pollution events (Fig. S6), horizontal transport should have little
influence on the daytime sources or sinks of HONO because of the short lifetime of
HONO (Spataro and Ianniello, 2014). In the night, 79 % of the wind speed was lower
than 1.0 m s$^{-1}$ in winter, thus the air masses from suburban areas should have influence
on the sources and sinks of HONO in Beijing. If the HONO concentration at
background was zero, the vertical and horizontal transport rate of HONO varied from
0.17 to 0.61 ppbv h$^{-1}$ which is calculated in the light of Eq. (13) on haze days in winter
and from 0.15 to 0.74 ppbv h$^{-1}$ in pollution events from April to June. These values
were higher than that calculated according to Eq. (12). Because the background HONO
concentration was unavailable, we only considered the nighttime transport calculated
according to Eq. (12) in the following section.
**Comparison among different HONO sources.** Fig. 3 summarizes the diurnal patterns
of each sources with different parameterizations during the pollution events from
February to March. The black dots and lines mean the middle values, while the shadow
indicates the corresponding lower bound and upper bound. In the nighttime, vehicle
and soil emission, and homogeneous reaction between NO and OH were the important
sources of HONO. In the daytime, however, photolysis of nitrate and homogeneous
reaction between NO and OH dominated the sources of HONO. Heterogeneous
reactions of NO$_2$ on aerosol surface and ground surfaces were not the major HONO
source during night unlike the modelled results (Zhang et al., 2016;Aumont et al., 2003).

Fig. 4A-F shows the HONO budget estimated using the middle values among these

parameters during the polluted events. The mean production rate of HONO varied in
the range of 0.25 - 1.81 ppbv h$^{-1}$ from these identified sources, while the corresponding
loss rate was from 0.21 to 2.34 ppbv h$^{-1}$ during the polluted events in winter. The main
loss of HONO was the photolysis during the daytime (1.74± 0.44 ppbv h$^{-1}$), whereas it
was vertical transport in the nighttime (0.28±0.08 ppbv h$^{-1}$). Direct emission from
vehicles exhaust was the largest nighttime source of HONO ($0.23\pm0.06$ ppbv h$^{-1}$),
followed by heterogeneous reactions of $NO_2$ on the ground surface ($0.07\pm0.01$ ppbv h$^{-}$
$^1$), homogeneous reaction between NO and OH ($0.04\pm0.01$ ppbv h$^{-1}$), emission from
soil ($0.014\pm0.005$ ppbv h$^{-1}$), and heterogeneous reactions of $NO_2$ on aerosol surface
($0.006\pm0.002$ ppbv h$^{-1}$). $P_{NO-OH}$ and $P_{nitrate}$ dominated the daytime HONO production,
with daytime mean values of $0.49\pm0.35$ ppbv h$^{-1}$ and $0.34\pm0.23$ ppbv h$^{-1}$, respectively.
As shown in Fig. 4, these six sources still underestimated the daytime sources of HONO.
The $P_{unknown}$ was $0.20\pm0.24$ ppbv h$^{-1}$ in February and March, while it was $0.50\pm0.24$
ppbv h$^{-1}$ from April to June.

The $E_{vehicle}$ contributed $57.0\pm10.0\%$ and $51.5\pm20.1\%$ to the nighttime HONO

sources from February to March and the rest months, respectively, even when the
$P_{unknown}$ was taken into consideration. The relative contribution of daytime $E_{vehicle}$
decreased to $15.2\pm15.4\%$ in winter and $9.7\pm7.8\%$ from April to June. Thus, the daily
mean fraction of the $E_{vehicle}$ was $39.6\pm24.3\%$ and $34.0\pm24.3\%$ from February to March
and from April to June, respectively. This means that the $E_{vehicle}$ dominates the nighttime
HONO source during the polluted events in Beijing, which is consistent with the
previous result that vehicle emission was the major nighttime or haze day HONO source
(51.1 % -52 %) in Beijing (Zhang et al., 2019c;Meng et al., 2019). As pointed out in
Section 3.3, $E_{vechicle}$ was calculated based on the $NO_x$ inventory from vehicle sector. On
the other hand, NO is prone to be quickly converted to $NO_2$ and $NO_z$ (including HONO,
$HNO_3$, $N_2O_5$, PAN and organonitrate etc) by $O_3$, $HO_2$, $RO_2$ and OH in the atmosphere.
It is reasonable to assume that local traffic emission dominates the ambient NO source
in the urban environment. Thus, homogeneous reaction between NO and OH in the
atmosphere could also be related to vehicle exhaust. As shown in Fig.3, although the
diurnal curve of $P_{NO-OH}$ coincided well with that of OH concentration (Fig. S10), which
means the $P_{NO-OH}$ should be mainly determined by OH concentration, the $P_{NO-OH}$ should
still reflect the indirect contribution of traffic related emission to HONO source because
the ambient NO concentration was used to calculate the $P_{NO-OH}$. Traffic-related HONO
sources ($E_{vehicles} + P_{NO-OH}$) might contribute 57.8±15 .8% and 48.6±15.9 % to the daily
HONO source in winter and the rest months, respectively. Even if 40 % of $NO_x$ was
from vehicle exhaust in Beijing (He et al., 2002), traffic-related source ($E_{vehicles} +$
$0.4P_{NO-OH}$) might still contribute 46.9±20.5 % in winter and 39.9±20.5 % from April to
June to the corresponding daily HONO source. The contribution of traffic-related
source was still an important daytime source of HONO (43.9±10.6 % for $E_{vehicles} + P_{NO-}$
$_{OH}$, and 26.7±12.4 % for $E_{vehicles} + 0.4P_{NO-OH}$) on polluted days in winter.

As shown in Fig. 3, uncertainties existed when calculating each HONO source. To

further understand the role of traffic emission, we also estimated the lower limit of the
traffic-related contribution as follows: 1) the lower limit of $E_{vehicle}$ was obtained by
using the lowest reported emission ratio of $HONO/NO_x$ from vehicles (0.18%) (Liu et
al., 2017d) rather than 1.17%, which was the empirical value calculated based on the
field measurement in Fig. S7; 2) the lower limit for homogeneous reaction between NO
and OH radical was calculated according to the method by Li et al. (2018); 3) the upper
limit of the emission rate from soil was estimated using the emission flux of HONO
with low water content (Oswald et al., 2013); 4) the upper limit of HONO production
rate from heterogeneous reaction of $NO_2$ on the aerosol was calculated using the large
RH-dependent uptake coefficient of $NO_2$ on hematite (Liu et al., 2015) rather the value
recommended by Crowley et al. (Crowley et al., 2010); 5) the upper limit for
heterogeneous reaction on ground surface was calculated using the RH-dependent
kinetic data measured on urban grime (Liu et al., 2019a). As shown in Fig. 5, traffic-
related source ($E_{vehicles}$ + $P_{NO-OH}$) contributed 25.7±15.8 % to the daily HONO sources
in winter if all NO was assumed to be dominated by local traffic emission, while it was
14.5±15.8 % when 40 % of NO was considered as local traffic emission (He et al.,
2002). Under this circumstance, the daytime $P_{unknown}$ of HONO in winter increased to
0.83±0.36 ppbv h$^{-1}$, which was corresponding to 58.1±8.6 % of the HONO source. This
means these assumptions might underestimate the contribution of the HONO sources.
In addition, $P_{ground}$, $P_{aerosol}$ and $P_{nitrate}$ could be also partially related to traffic emission
of $NO_x$ (Lee et al., 2016;Tan et al., 2017). These results mean that the contribution of
traffic-related emission might be larger than our estimation in this work. Therefore, our
work at least suggests that traffic related emission should be a very important HONO
source in winter Beijing although more work is required based on comprehensive
modelling studies.
**4. Conclusions and atmospheric implications**.
In this work, the promotion effect of HONO on aerosol mass formation in polluted
events was observed based on the good correlation between the growth of OA and
nitrate mass concentration and the consumed HONO from early morning to noon during
the polluted days in winter. This promotion effect could be related to OH production

from photolysis of HONO on aerosol formation followed by oxidation process of the

corresponding precursors. Our observation supports well the recent modelling studies

that HONO may significantly promote secondary aerosol mass formation (Zhang et al.,

2019a;Zhang et al., 2019c;Xing et al., 2019;An et al., 2013). Based on budget analysis

calculations, traffic-related sources (direction emission and conversion of NO from

vehicle emission) was found to be an important contributor to HONO source during

polluted days in winter in Beijing. This means that HONO from the traffic-related

sources can have an important role in aerosol mass formation in the atmosphere.

Vehicle population in China is increasing very quickly (Liu et al., 2017b;Wang et

al., 2011). Thus, the negative influences of the vehicle emission on air quality will

increase especially in populous metropolitan areas (Yang et al., 2019;Guo et al., 2020),

such as Beijing and Shanghai, if targeted pollution control technologies are not applied.

It has been estimated that the vehicles emission accounted for over 40% of total urban

$NO_x$ emissions in Beijing (He et al., 2002). In the atmosphere, $NO_x$ involves very

complicated reaction network, from which finally leads to aerosol mass formation and

production of ozone in VOC limited environment. At the same time, reactions of $NO_x$

also leads to some reactive $NO_z$ species (Seinfeld and Pandis, 2006). In particular,

HONO is an important precursor of OH, which governs the conversion of primary

pollutants to secondary pollutants in the atmosphere. Besides indirect production of

HONO from NO, the vehicles also directly emits HONO as discussed in this work.

Even if the low limit of emission factor was used to calculating the HONO source from

the vehicles, the traffic-related emission can still be an important source of HONO in

winter Beijing. Therefore, this work implies that mitigation of HONO and $NO_x$
emission from vehicles might be an effective way to reduce secondary aerosol mass
formation and can have a positive effect on severe haze events in wintertime Beijing.
It should be pointed out that we only considered $O_3$ and HONO when discussing
the sources of OH. Other sources such as $HO_2$ (and $RO_2$) with NO, ozonolysis of
alkenes and photolysis of OVOCs might also contribute to OH radicals in the
atmosphere (Tan et al., 2018). In the future it will be vital to comprehensively analyze
OH sources and to quantify the role of HONO in secondary aerosol mass formation
although photolysis of HONO is the major OH source in winter. On the other hand, as
discussed in Section 3.3, uncertainties about the HONO budget might originate from
the emission factors, OH concentration, and reaction kinetics and so on. The source of
HONO from vehicles was calculated based on the emission inventories, which should
have a significant bias (Squires et al., 2020). For example, the emission flux of $NO_x$
calculated using the emission inventory from Yang et al. (2019) is as 2.4±0.5 times as
the reported emission flux reported by Squires et al. (2020). To take the next step, it is
required to measure the emission factors from vehicle exhaust under real road
conditions in the future. When calculating the OH concentration, the factor between
OH concentration and $J_{O1D}$ might vary over locations and seasons due to different
$NO_x$/VOCs ratio (Holland et al., 2003). Direct measurements of OH concentration
would be helpful for decreasing the uncertainty of both OH sources and HONO budget
analysis. Finally, it is necessary to quantify the contribution of traffic-related source of
HONO on secondary aerosol formation based on modelling studies in the future.

*Data availability*. The experimental data are available upon request to the corresponding authors.


*Supplement*. The supplement related to this article is available online at:


*Competing interests*. The authors declare that they have no conflict of interest.


## Author information

*Author contributions*. YL, WW and MK designed the experiments. YL wrote the paper and performed HONO budget analysis. YZ, CL, WW, YC, MG and XW carried out HONO measurement. ZF, FZ, JC, WD and KD did aerosol composition measurements. BC and JK did particle size measurements. YW, BH and YW analyzed meteorological data analysis. CY, FB, JK, TP, HH, MG and MK revised the manuscript.

## Acknowledgements:

This research was financially supported by the National Natural Science Foundation of China (41877306), the Ministry of Science and Technology of the People's Republic of China (2019YFC0214701), Academy of Finland via Center of Excellence in Atmospheric Sciences (272041, 316114, and 315203) and European Research Council vShandong Universityia ATM-GTP 266 (742206), the Strategic Priority Research

Program of Chinese Academy of Sciences and Beijing University of Chemical
Technology.

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

**Figure captions**

Fig. 1. An overviewed measurement of non-refractory-PM$_{2.5}$ (NR-PM$_{2.5}$), HONO, NOx, PM$_{2.5}$ and meteorological parameters from Feb. 1 to July 1, 2018. (A) the mass concentration of different components of PM$_{2.5}$, (B) the mass fraction of individual component, (C) HONO and NOx concentration, (D) temperature and RH, (E) wind speed and wind direction, (F) UVB and PBL height and (G) visibility and PM$_{2.5}$ concentration during observation. We consider the period before Apr. 1 as winter. During the winter period, 12 cases are selected and numbered, including three clean cases (1, 3, and 5, marked in yellow) and the rest 9 pollution episodes (marked in blue).

**Fig. 2.** Contribution of HONO to OH production and correlation between OA and HONO concentration. Diurnal production rates of OH from photolysis of HONO and O$_3$ on polluted days with PM$_{2.5}$ concentration larger than 50 μg m$^{-3}$ and RH less than 90 % (A) from Feb 1 to Mar 31, (B) from Apr 1 to Jun 30; (C) Daytime variation of OA/CO and HONO/CO concentration for the 7$^{th}$ and 12$^{th}$ episodes and (D) correlation of the daytime OA/CO increased and consumed HONO/CO.

**Fig. 3.** Diurnal pattern of HONO sources calculated with different parameterizations. The low bound, the middle value, and upper bound of (A) soil emission calculated based on 45-55%, 35-45% and 25-35% of water content, (B) vehicle emission with relative emission factor to NOx of 0.18%, 1.17±0.05% and 1.8 %, (C) production from reaction between NO and OH, whose concentration estimated using Xu (Xu et al., 2015), (Tan et al., 2019)

**Fig. 4.** The budget of HONO (A) and (B) Diurnal production rates of HONO, (C) and

(D) loss rates of HONO, (E) and (F) relative contribution of each source on polluted
days with PM$_{2.5}$ concentrations higher than 50 μg m$^{-3}$ and RH less than 90 %. The left
column shows the data from February 1 to March 31) and the right one shows the data
from April 1 to June 30.
**Fig. 5.** (A)-(B) Diurnal production rates and (C)-(D) diurnal loss rates of HONO; (E)-
(F) relative contribution of HONO sources on polluted days with PM$_{2.5}$ concentrations
higher than 50 μg m$^{-3}$ and RH less than 90 %. The $E_{vehicle}$ is calculated using the low
limit of HONO/NOx from vehicles (0.18%) (Liu et al., 2017d) and the $P_{NO-OH}$ is
calculated using the low limit of OH concentration, while the upper limit of $E_{soil}$, $P_{aerosol}$
and $P_{ground}$ are used as described in the text.

## Figures

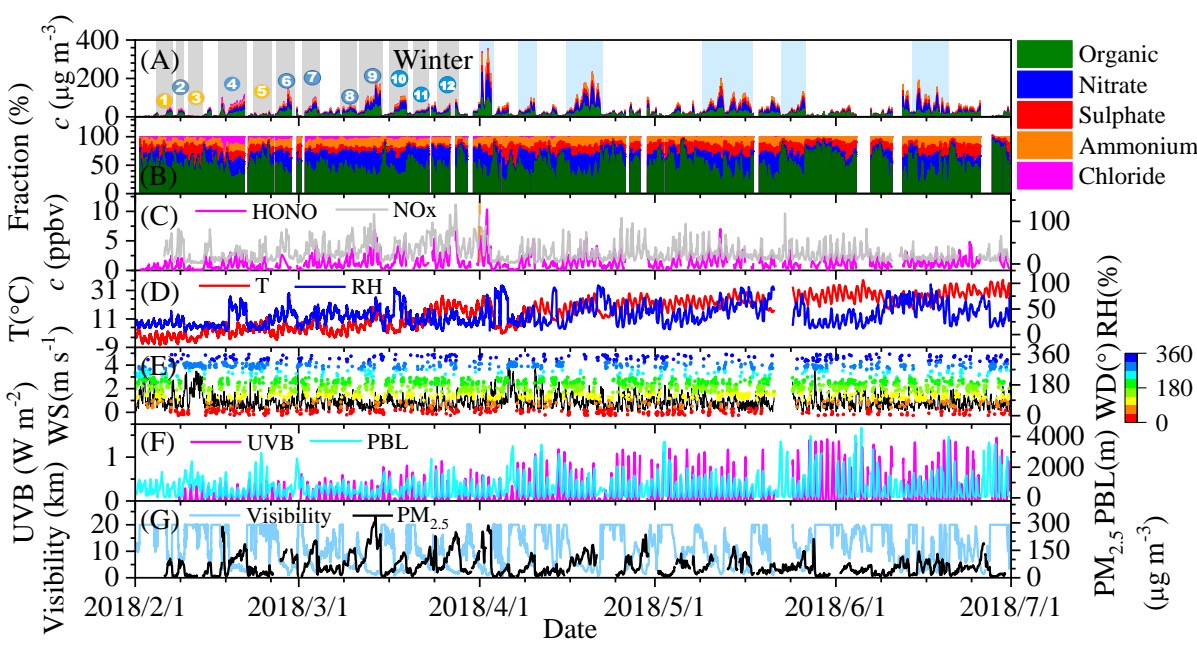

**Fig. 1**.

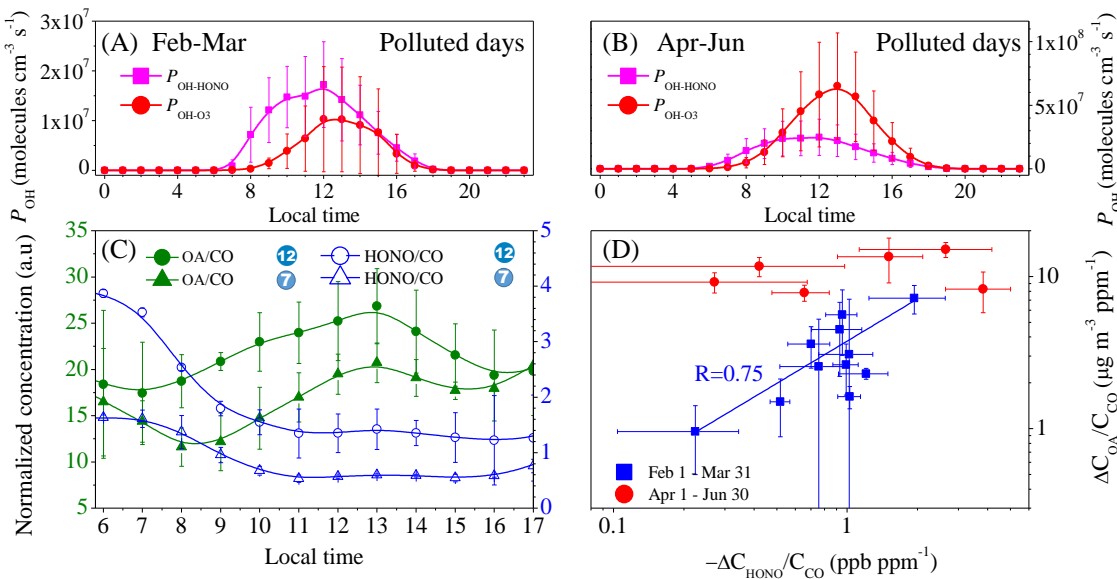


**Fig. 2**.

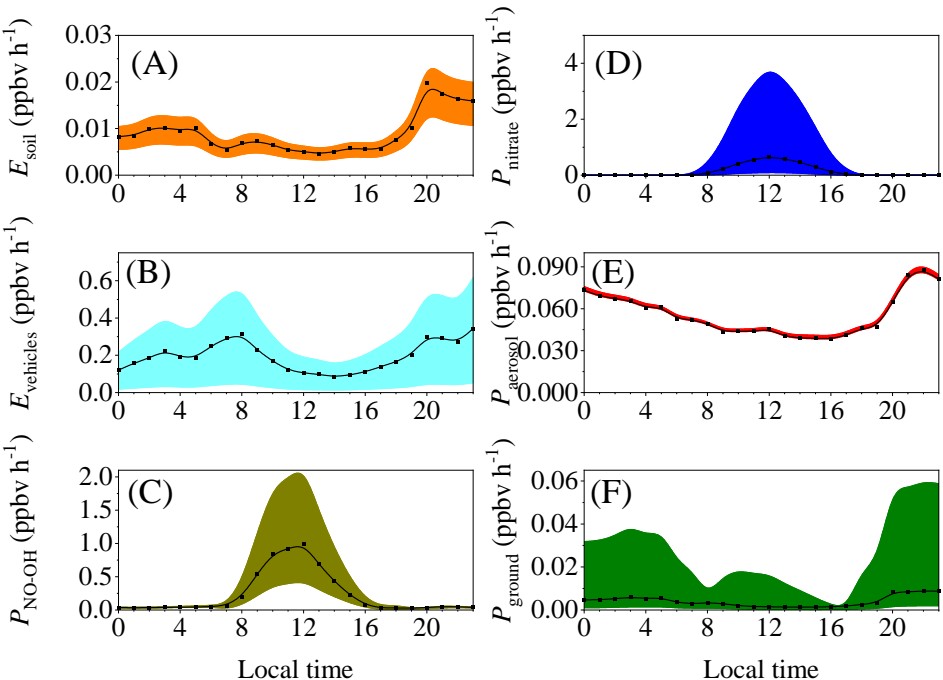


**Fig. 3**


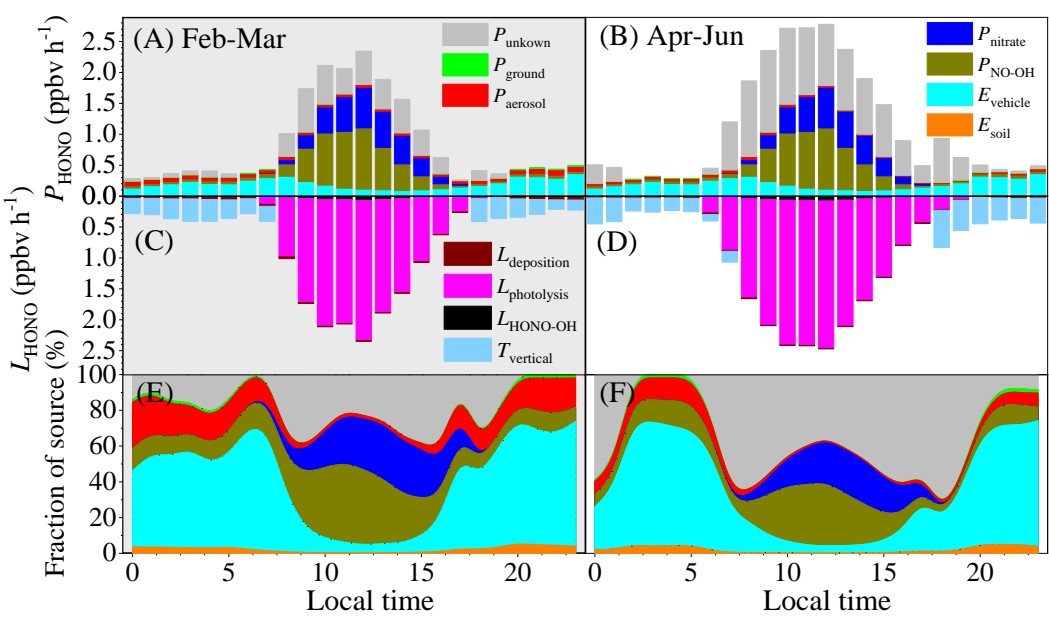


**Fig. 4**.


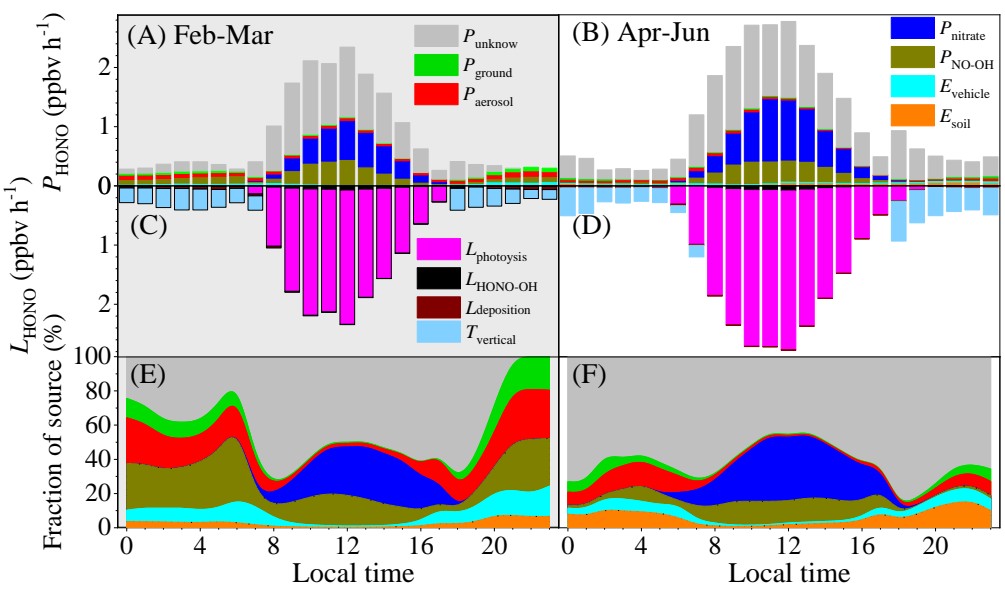


**Fig. 5**.