# Peer review of "The promotion effect of nitrous acid on aerosol formation in 1 wintertime Beijing: possible contribution of traffic-related 2 emission 3 4 Yongchun Liu1\*, Yusheng Zhang1, Chaofan Lian2,6, Chao Yan3, Zeming Feng1, Feixue 5 Zheng1"

_Atmospheric Chemistry and Physics, 2020_

## Referee Comment (RC1) · Anonymous Referee #1 · 6 Apr 2020

The work by Liu et al present field measurements of HONO, a key source of radicals in the boundary layer, along with supplementary gas and particle-phase measurements in over a 5-month period in Beijing. The authors used this dataset to probe the sources and their contributions to ambient HONO, with a focus on pollution events. Using a steady state approach, the authors calculated the contributions of different sources to the HONO budget and concluded that traffic emission (via direct emission and conversion of NO by homogenous reactions) was the key source of HONO during winter pollution events. Liu et al present a comprehensive and interesting long-term dataset that enables the authors to perform a budget analysis to investigate the main sources of HONO. The main sources of HONO in Beijing and urban area in general is an open

research topic, and consequently this work would be of interest to the community, particularly their HONO budget analysis of the haze events.

There are, however, a few issues that in my opinion should be addressed prior to publication. While the manuscript is mostly well written, it is long. For example, Section 3.1 and 3.2 could be much shorter. In my opinion much of the text in these sections is unnecessary and could be reduced, without losing the key points.

The most interesting work is presented in Section 3.3, where a detailed and comprehensive budget analysis is presented. However, throughout section 3.3 some of the calculations and equations need more explanation, as it was not always clear from section 2.2 how they were performed. Some examples are given below in the minor comments. It would also help the reader if the equation used to calculate the rates of emission for each source (I.e. the eqn numbers) were referenced throughout section 3.3.

My major comment is from Section 3.3, the way the OH concentration was estimated is problematic in my opinion. As the authors rightly point out, to measure OH is difficult and requires highly specialized kit and therefore as they did not have access to these instruments, OH concentration needs to be estimated for this study. I am not sure about the way the OH concentration was calculated, as the equations they use (e.g. 13), use the levels of ozone and NO2. The problem is that during winter, the main source of OH in Beijing is HONO photolysis, as the authors themselves state earlier in the manuscript (Section 3.2, line 288, with references), and is in facto one of their conclusions from Fig 2. Therefore, without considering OH produced from HONO photolysis, how can they be sure that their estimated OH concentration is reasonable, especially in winter? I think that for an atmosphere as chemically complex as Beijing, to estimate the OH concentration will affect the budget analysis, both sources and loss terms, and therefore the conclusions drawn from it. If this is not possible, then the uncertainties with their approach to estimating OH concentrations should at least be

discussed/quantified.

Minor comments Line 82: I assume you mean nitrous acid not nitric acid?

Line 112: it would be good to specify that your ACSM was configured for PM2.5, as many of these instruments measure  $\mathsf{PM1}$

Line 207: Why does it matter if PM2.5 is above 75 ug m-3? I assume you are referring to the regulatory limits, but it is good to be clear on this.

Line 211: I am a bit confused by your explanation for there being more pollution episodes and higher concentrations of BC, CO and PM2.5 in March, as it was the heating season. But isn't February just as cold? So why would there more heating in March?

Line 222: Are the percentages listed for nitrate, chloride and ammonium also monthly means?

Line 226: I am bit surprised that fireworks is regionally transported from Tangshan, are there no fireworks in Beijing?

Line 232-240: As example to one of my main comments above, I found that this paragraph was repeating much of the information presented in the preceding one. Perhaps these two paragraphs could be edited and combined.

Line 272: why have you chosen to subset the data based on 'when the PM2.5 concentration was larger than 50  $\mu$ g m-3 and the RH was less than 90 %'. Furthermore, as you state 'Under these conditions, local chemistry should be more important as 75 % of the wind speed was less than 1.0 m s-1'. Why not then subset the data based solely on wind speed if local sources are of interest?

Line 275: How where these maximal POH-HONO and POH-O3 values calculated? I could not find the equation in the methods or reference.

Line 304: Is it not the production of the OH that changes in winter relative to summer,

rather than the rate of oxidation of VOC by OH? Please clarify

Line 318-20 and Fig 2: I am not so sure that is 'reasonable to mainly ascribe the increase of OA concentration to local secondary formation initiated by OH radical from HONO photolysis'. This is because if only OH from HONO photolysis was driving secondary formation, then shouldn't the OA/CO peak earlier, as the ambient HONO is essentially run out by 10am?

Line 324: But cant there also be anthropogenic sources of alkenes? For example, isoprene can also be from vehicle emissions (See e.g. Zou et al 2019).

Line 379: at the start of the sentence, you state that hourly NOx EI were available, yet than go to give a yearly emission factor? Why wasn't hourly EI used, and did you consider your measured NOx concentrations, as the diurnal variation in NOx would be important? Please clarify in more detail how the Evehicle was calculated Especially as emission inventories can have significant bias (See for example very recent work for Beijing by Squires et al, 2020).

Line 381: This may be related to the above comment, but how did you report a range for calculated middle value of Evehicle, when the NOx EI rate is constant and the HONO/NOx is constant? Furthermore, what does the middle value refer to? Daily avg? hourly avg? please specify. This applies throughout this section

Line 386: the reported range for the upper limit is the same as reported for the lower limit, I'm guessing a typo?

Line 389: Please give the reference for the emission flux you used, the value and also why grassland was the most appropriate.

Line 397: why does only the middle values for Esoil have uncertainty calculated? Also how did you estimate the uncertainty for Esoil? And why did you use a range of soil water content for lower, middle and upper, why not just use a single value?

Line 416: Please provide more information on the night time temperature dependence

of OH concentration, and the equations used in this calculation.

Line 419: Please give the reported OH concentrations by Li et al (2018) and Huang et al (2017) and if they were calculated or measured OH levels.

Line 433: How was the HONO form nitrate photolysis calculated? Which equation (give number)? What do these ranges represent?

Line 447: by the end of this paragraph, it was not at all clear to me which uptake co-efficient you actually used. Please clarify.

Line 477-9: the authors state that 'Heterogeneous reactions of NO2 on aerosol surface and ground surfaces were unimportant compared with other sources because of the very low uptake coefficient'. What do you mean by the very low uptake coefficient, low compared to what? Is the issue more that you used the wrong co-efficient?

Fig 2D: if you take the bottom and top points in Feb/Mar (blue), I am not sure there this a correlation. It would be good to check if you get a similar slope and r2 without these 2 points.

References Squires, F. A., Nemitz, E., Langford, B., Wild, O., Drysdale, W. S., Acton, W. J. F., Fu, P., Grimmond, C. S. B., Hamilton, J. F., Hewitt, C. N., Hollaway, M., Kotthaus, S., Lee, J., Metzger, S., Pingintha-Durden, N., Shaw, M., Vaughan, A. R., Wang, X., Wu, R., Zhang, Q., and Zhang, Y.: Measurements of traffic dominated pollutant emissions in a Chinese megacity, Atmos. Chem. Phys. Discuss., https://doi.org/10.5194/acp-2019-1105, in review, 2020. Zou, Yu, Xue Jiao Deng, Tao Deng, Chang Qin Yin, and Fei Li. "One-Year Characterization and Reactivity of Isoprene and Its Impact on Surface Ozone Formation at A Suburban Site in Guangzhou, China." Atmosphere 10, no. 4 (2019): 201.

---

## Referee Comment (RC2) · Anonymous Referee #2 · 7 Apr 2020

This paper reports the possible promotion effect of nitrous acid (HONO) on the formation of secondary organic aerosol (SOA) and nitrate in winter based on a five-month comprehensive observation in urban Beijing. Evidence for the relationship between secondary aerosol formation and the consumed HONO was obtained from the observations. The detailed budget of ambient HONO was explored, and vehicle emission was proposed as a significant source of HONO in Beijing. Overall, the manuscript is logically organized and well written, and the measurement data are much valuable. I would like to recommend that it can be considered for publication after the following major and specific comments being properly addressed.

Major Comments:

Section 2.2. HONO budget calculation: the description of the budget calculation is not clear enough. Firstly, the method used in this study was the budget analysis, other than the stationary state analysis. For a thorough budget analysis, physical terms (such as vertical and horizontal transport) should be considered for equations (1) and (12). At least, the authors need to evaluate if the physical processes were negligible for the analysis presented here. Secondly, a number of parameters (e.g., Fhono, Coh, gamma, Yhono, Jnitrate, Jhono, etc.) are required for the analysis. It is not clear how these parameters were obtained or approximated in this study. Although they were described more or less in the following Section 3.3, the authors may need clearly state the data source and uncertainties of these key parameters at their first appearance. This may help the readers better understand the overall methodology of this study.

Section 3.3 and Figures 3&4: following the first comment, the description and discussion of the HONO budget are also not clear and need clarification. The contributions of heterogeneous reactions of NO2 on aerosol and ground surfaces were too low, and they were even lower than the OH+NO reactions during the nighttime. This is unusual. Is it reasonable to approximate the nighttime OH concentrations linearly with the temperature? Furthermore, the heterogeneous reactions of NO2 on aerosol and ground surfaces were highly dependent on the NO2 concentrations, the uptake coefficients, and surface density, some of which are highly uncertain. It is not clear what values were actually adopted for the uptake coefficients of NO2 on aerosol and ground surfaces, and how much were the ambient NO2 levels and surface density? More details about the calculation of HONO budget are needed.

The authors attempted to quantify the contributions of vehicle-emitted NO to ambient HONO via NO+OH reaction based on the source apportionment of NO emissions. This is not convincing because the homogeneous HONO formation is generally limited by OH other than NO. This means that the produced HONO should be not linearly dependent to the NO emissions. It can be concluded that vehicle emission should

contribute significantly to not only direct HONO emission but also HONO formation through reactions of NO and NO2. However, the current quantification analysis needs be more careful.

Specific Comments:

Line 44: fine particulate matter with diameter less than or equal to 2.5. . .

Line 78: on polluted days

Lines 84-94: to my knowledge, there have been a number of observational studies of HONO in recent years in China, and similar HONO budget analyses were performed. I suggest the authors to comprehensively review the existing results about the HONO sources in China and compare them against the source analysis results obtained in the present study.

Line 138: replace "nitrous acid" by "nitric acid"

Line 146: Equation (1) describes the budget analysis other than the stationary state analysis. Both methods are different. Transport terms need be considered here.

Lines 198-202 and Fig. 1: it would be much better if the authors could also plot the other related parameters, such as NOx and meteorological parameters, in Fig. 1. It is difficult for the readers to look at the same measurements separately from main text and supplement.

Line 228: the increase in temperature. . .

Line 262: what does "in RO2 chemistry" mean? Rephrase this sentence.

Line 266: replace "dominating" by "dominant"

Lines 270-282: it is not clear how the POH-HONO and POH-O3 were calculated from the current discussion. The calculated POH-HONO and POH-O3 levels in winter and April-June seem to be too high. Detailed calculation methods should be given here.

Usually, the OH+NO reactions should be subtracted from the photolysis of HONO to denote the real contribution of HONO to the OH source.

Line 291: photolysis of HCHO is actually the primary source of HO2.

Line 330: delete "and".

Lines 333-336: Several recent papers about the nitrate aerosol trend and formation mechanisms in China are highly relevant to this study, and should be acknowledged. Wen et al., Summertime fine particulate nitrate pollution in the North China Plain: increasing trends, formation mechanisms, and implications for control policy, Atmospheric Chemistry and Physics, 18, 11261-11275, 2018.

Sun et al., Two years of online measurement of fine particulate nitrate in the western Yangtze River Delta: influences of thermodynamics and N2O5 hydrolysis, Atmospheric Chemistry and Physics, 18, 17177-17190, 2018.

Line 363: high or low HONO concentration?

Line 408: Tan et al., (2019)

Line 407-408: it would be better if the authors could provide the estimated OH levels here.

Lines 430-431: it is not clear how the photolysis frequency of nitrate was corrected? Details are needed here. What are the J values used finally?

Line 553-554: again, it is budget analysis other than stationary state calculation.

Line 568: indirect production

---

## Referee Comment (RC3) · Anonymous Referee #3 · 9 Apr 2020

The title of this paper is very intriguing that (1) wintertime HONO promotes aerosol formation and (2) >50% of observed HONO is traffic related in Beijing. After reviewing this paper, I think it will be a grave mistake if the editor decides to publish this paper with these two conclusions in any form. The conclusions are pure speculations. I find no evidence to support either of the two claims in this paper.

The discussion for conclusion (1) is in section 3.2. One of the many mistakes in this section is that the authors do not understand that the largest source of OH is from the reaction of HO2+NO. Even when OH production from HONO photolysis is larger than from O3 photolysis, the effect on OH is much smaller than the photolysis rate

comparison. Line 301-304 is based on another paper; the data in this paper do not either support or dispute that oxidation by OH promotes aerosol formation. Figure 2D is used at the observation evidence supporting conclusion (1). There are many reasons that HONO/CO correlates with OA/CO. For example, CO is primary in winter in Beijing. If HONO and OA variations are from secondary sources, there will be high correlations as shown. Line 318 states ". . . it was reasonable to mainly ascribe the increase of OA concentration to local secondary formation initiated by OH radical from HONO photolysis." It is a pure speculation. The observation data in this paper do not support this statement. It is the same with Line 328. The vague statement cannot be supported by the data in this paper. Line 332 is again a speculation. Ammonia is mostly neutralized by sulfate in Beijing. Line 338-400 is another speculative and ambiguous statement. Line 345-345 cites other people's work but is not supported by the data in this work.

Conclusion (2) is based on some calculation that was not described in the paper. Line 376-378 states that the mean emission factor is 1.17% with a lower limit of 0.18% and an upper limit of 1.8%. (Why is the mean so close to the upper limit and 6.6 times larger than the lower limit?) The mean value is similar to previous studies and is not the reason for conclusion (2). Line 381 gives a vehicle HONO emission rate of 0.085 to 0.34 ppbv/h. The unit implies some volume was used in the calculation. No discussion was given on what volume was used and how it varied in a day. Another important factor not considered in this study is the outflow of vehicle HONO and NOx by advection at night. It is the largest sink at night but is not included in the budget discussion. The nighttime source of NO2 from the ground is 38 times less than vehicle emissions. However, no other paper I know of found that HONO concentrations at night cannot be explained mostly by a ground source. It led me to conclude that the vehicle HONO emission source in this paper is overestimated by 10-100 times. The authors should look at previous modeling papers that included vehicle HONO emissions. What they found is that the effect of vehicle HONO emissions is small.

In summary, I think that the calculation and reasoning of this paper are either incorrect or ambiguous. It does seem likely that any revision can correct the flaws in the two main conclusions. I suggest that the authors scratch the conclusions and redo the analysis of their observation data.

---

## Author Comment (AC1) · 18 May 2020

**Reviewer 1#**

The work by Liu et al present field measurements of HONO, a key source of radicals in the boundary layer, along with supplementary gas and particle-phase measurements in over a 5-month period in Beijing. The authors used this dataset to probe the sources and their contributions to ambient HONO, with a focus on pollution events. Using a steady state approach, the authors calculated the contributions of different sources to the HONO budget and concluded that traffic emission (via direct emission and conversion of NO by homogenous reactions) was the key source of HONO during winter pollution events. Liu et al present a comprehensive and interesting long-term dataset that enables the authors to perform a budget analysis to investigate the main sources of HONO. The main sources of HONO in Beijing and urban area in general is an open research topic, and consequently this work would be of interest to the community, particularly their HONO budget analysis of the haze events.

**Response:** Thank you so much for your positive comments.

There are, however, a few issues that in my opinion should be addressed prior to publication. While the manuscript is mostly well written, it is long. For example, Section 3.1 and 3.2 could be much shorter. In my opinion much of the text in these sections is unnecessary and could be reduced, without losing the key points.

**Response:** Thank you so much for your instructive suggestions. We have revised the redundant descriptions in Section 3.1 and 3.2.

In Section 3.1, the paragraph "In particular, the frequency of severe polluted episodes in March was obviously higher than that in the rest months (Fig. 1 and S3), resulting in the highest monthly mean concentration of $PM_{2.5}$ (88.5±60.0 μg m$^{-3}$) and NR-$PM_{2.5}$ (67.0±56.8 μg m$^{-3}$). This can be explained by both intensive emission during the heating season, which is supported by the high concentration of primary pollutants including CO, $SO_2$ and BC (Table S1), and the stagnant meteorological conditions that physically and chemically promote the accumulation of pollutants. For example, the low wind speed (<2 m s$^{-1}$) mainly from south-based directions accompanied with the low planetary boundary layer (PBL) height frequently occurred in March compared with other months (Fig. S4A)." has been shortened as "Both the frequency of severe polluted episodes and the mean mass concentration of $PM_{2.5}$ and NR-$PM_{2.5}$ were obviously higher in March than that in the rest months (Fig. 1 and S3). This can be explained by both the intensive emission during the heating season as evidenced by the high concentration of primary pollutants including CO, $SO_2$ and BC (Table S1) and the stagnant meteorological conditions supported by the low wind speed ($<2$ m s$^{-1}$) and the low planetary boundary layer (PBL) height in March (Fig. S4A)" from lines 249 to 255 in the revised manuscript. In addition, we also deleted the sentences "It should be noted that the median mass concentrations of nitrate and OA also were higher in March than that in other months (Fig. S4C). The median mass concentrations of nitrate were 1.42, 8.76, 6.30, 3.15, and 3.23 μg m$^{-3}$ from February to June, respectively. And the corresponding OA concentrations were 4.78, 14.04, 11.64, 13.89, and 14.08 μg m$^{-3}$. Secondary formation is the major source of OA and nitrate in the atmosphere" from line 232 to 237 in the original version of the manuscript.

In Section 3.2, we deleted the following sentences "The hourly averaged $P_{\text{OH-HONO}}/P_{\text{OH-O3}}$ ratio varied in the range of 1-25.4 during the daytime, while it varied from 0.3 to 2.8 from April to June" (lines 284-288, in the original manuscript), "Although the high loading of fine particles in polluted days could reduce the surface solar radiation (Li et al., 2017), subsequently, the OH concentration, the noon-time OH radical concentrations observed in polluted wintertime of Beijing were $2.4 \times 10^6$ cm$^{-3}$ compared with $3.6 \times 10^6$ cm$^{-3}$ in clean days (Tan et al., 2018). It was around 2 times compared to places such as Tokyo (Kanaya et al., 2007) and New York City (Ren et al., 2006)" (lines 295-300, in the original manuscript), and " This implies that oxidation of atmospheric trace gases by OH may still be highly effective even in wintertime, thereby facilitating the vigorous formation of secondary pollutants in Beijing" (lines 301-304, in the original manuscript).

The most interesting work is presented in Section 3.3, where a detailed and comprehensive budget analysis is presented. However, throughout section 3.3 some of the calculations and equations need more explanation, as it was not always clear from section 2.2 how they were performed. Some examples are given below in the minor comments. It would also help the reader if the equation used to calculate the rates of emission for each source (I.e. the eqn numbers) were referenced throughout section 3.3.

**Response:** Thank you for your good suggestions. We have referenced all the equation numbers throughout the revised manuscript. For example, in line 389 in the revised manuscript, "The $E_{vehicle}$ was calculated according to Eq. (2) using the relative emission rate of HONO to $NO_x$ and the emission inventory of $NO_x$ from vehicles"; In lines 446-448 in the revised manuscript, "The lower limit, the middle value and the upper limit of the $E_{soil}$ are 0.0032±0.0027-0.013±0.014, 0.0046±0.0039-0.020±0.20 and 0.0057±0.0047-0.025±0.024 ppbv h$^{-1}$, respectively, calculated according to Eq. (2)"; In lines 473-475 in the revised manuscript, "Thus, the lower limit, the middle value and the upper limit of $P_{NO-OH}$ were 0.007±0.019-0.43±0.26, 0.026±0.053-0.99±0.79 and 0.028±0.053-2.14±1.71 ppbv h$^{-1}$, respectively, calculated according to Eqs. (3) and (4)"; In lines 495-498 in the revised manuscript, "Therefore, the corresponding daytime lower limit, the middle value and the upper limit of HONO from photolysis of nitrate were 0.0011±0.0021-0.096±0.092, 0.0072±0.0021-0.66±0.092 and 0.042±0.082-3.86±0.008 ppbv h$^{-1}$, respectively, calculated in the light of Eqs. (3) and (8)"; And in lines 499-501 in the revised manuscript, "The production of HONO from heterogeneous reactions of $NO_2$ on aerosol surface was calculated according to Eqs. (3) and (5)".

We also added the more details about budget calculations. For example, we added the following paragraphs "Oswald et al. (2013) measured the emission flux of HONO from 17 soil samples, including eucalyptus forest, tropical rain forest, coniferous forest, pasture, woody savannah, grassland, stone desert, maize field, wheat field, jujube field an cotton field etc. Tropical rain forest, coniferous forest and grassland are the typical plants in downtown Beijing (Huang et al., 2017a). At the same time, their emission fluxes of HONO are comparable (Oswald et al., 2013). Thus, we used the emission flux from grassland to calculate the emission rate of HONO from soil in Beijing because the temperature and water holding content dependent emission flux of HONO was available for grassland soil" lines 433-441; "The method for the photolysis rates calculation were shown in the SI and the time series of the photolysis rates were shown in Fig. S7" in lines 458-460; "The time series of the measured nitrate concentration and the middle value of $J_{nitrate}$ were shown in Fig. 1 and Fig. S7, respectively" in lines 493-495; "The $A_s$ of aerosols varied from $1\times10^{-4}$ to $4.8\times10^{-3}$ m$^{-1}$ with a mean value of $1.4\pm0.5\times10^{-3}$ m$^{-1}$ during pollution events. This value is comparable with that used in modeling studies (Zhang et al., 2016;Aumont et al., 2003). The $A_s$ of ground surface which was calculated according to Eq. (6) and (7) varied from $1.5 \times 10^{-3}$ to $3.85 \times 10^{-2}$ $m^{-1}$ with a mean value of $1.3 \pm 0.9 \times 10^{-2}$ $m^{-1}$ during pollution events. The surface roughness was 3.85 calculated according to Eq. (7). The $Y_{HONO}$ was set to 0.5 because of the hydrolysis reaction of $NO_2$ (Liu et al., 2015), while it was 0.8 for light enhanced reaction (Liu et al., 2019;Ndour et al., 2008) and on BC (Han et al., 2013)" in lines 523-530 in the revised manuscript.

My major comment is from Section 3.3, the way the OH concentration was estimated is problematic in my opinion. As the authors rightly point out, to measure OH is difficult and requires highly specialized kit and therefore as they did not have access to these instruments, OH concentration needs to be estimated for this study. I am not sure about the way the OH concentration was calculated, as the equations they use (e.g. 13), use the levels of ozone and $NO_2$. The problem is that during winter, the main source of OH in Beijing is HONO photolysis, as the authors themselves state earlier in the manuscript (Section 3.2, line 288, with references), and is in facto one of their conclusions from Fig 2. Therefore, without considering OH produced from HONO photolysis, how can they be sure that their estimated OH concentration is reasonable, especially in winter? I think that for an atmosphere as chemically complex as Beijing, to estimate the OH concentration requires a box-model approach. It is important as the estimated OH concentration will affect the budget analysis, both sources and loss terms, and therefore the conclusions drawn from it. If this is not possible, then the uncertainties with their approach to estimating OH concentrations should at least be discussed/quantified.

**Response:** Thank you for your instructive suggestion. We agree with you that OH concentration is an important parameter when calculating the HONO source from homogenous reaction between NO and OH. Some uncertainties of OH concentration should be resulted from the estimation approaches used in this work. In strictly speaking, it is better to directly measure OH concentration using a LIF or a CIMS. However, these instruments are unavailable at the present time. A box-model simulation is an other choice as you suggested because both the source and sink terms can be considered. A comprehensive sheet of VOCs including different isomers is required when doing box-model simulation because the reactivity varies greatly among different isomers even with the same mass to charge ratio (m/z). Unfortunately, the

VOCs concentration was measured using a SPI-MS in this work. Similar to a PTR-MS, this instrument cannot separate the isomers with the same m/z. Therefore, we did not simulate OH concentration using a box-model. If we had the measured or modeled OH concentration, we could parameterize the $J_{HONO}$ and $c_{HONO}$ into the Eq. (15).

We indirectly verified the estimated OH concentration by comparison with diurnal curves of OH concentration in this work and that reported in literatures (Tan et al., 2019). Overall, the estimated OH concentration was comparable with that measured in the literature (Fig. S10). In addition, we further compared it with the OH concentration derived from the measured $H_2SO_4$ concentration using a nitrate-CIMS from December 2019 to February 2020. In a box model, we considered the daytime source of $H_2SO_4$ from oxidation of $SO_2$ by OH and the sinks of $H_2SO_4$ using the measured concentration of the monomer and the dimer of $H_2SO_4$, the condensation sink (CS, from 1 nm to 10 µm of aerosol) and the meteorological parameters. Then, OH concentration was calculated using a steady state method. As shown in Fig. R1, the predicted OH concentrations was generally comparable between these two different methods. This means the estimated OH concentration in this work is overall credible.

[Figure]

Fig. R1. Comparison of OH concentration between UV proxy method and $H_2SO_4$ proxy method.

In the revised manuscript (lines 482-488), we added the discussion about the uncertainties from OH concentration estimation as you suggested. "It should be noted that OH concentration was estimated based on $J_{O1D}$ (Tan et al., 2019;Tan et al., 2018) or $J_{O1D}$ and $J_{NO2}$ (Li et al., 2018). As discussed in Section 3.2, HONO was an important primary OH source in the daytime. Unfortunately, it could not be parameterized for calculating OH concentration because the measured or modelled OH concentration was unavailable in this work. This might underestimate the early daytime OH concentration, subsequently, the contribution of homogeneous reaction of NO with OH to HONO source. This need to be further investigated in the future".

Minor comments Line 82: I assume you mean nitrous acid not nitric acid?

**Response:** Thank you. It is nitrous aid. We have corrected this error in line 81 in the revised manuscript..

Line 112: it would be good to specify that your ACSM was configured for $PM_{2.5}$, as many of these instruments measure $PM_1$

**Response:** Thank you. It has been pointed out in the revived manuscript in lines 136-137 as "Then a Time-of-Flight Aerosol Chemical Speciation Monitor equipped a $PM_{2.5}$ aerodynamic lens (ToF-ACSM, Aerodyne)".

Line 207: Why does it matter if $PM_{2.5}$ is above 75 ug m$^{-3}$? I assume you are referring to the regulatory limits, but it is good to be clear on this.

**Response:** Thank you. Yes, it the regulatory limits. The air quality is in pollution level if the $PM_{2.5}$ concentration is above 75 ug m$^{-3}$ according to the national air quality standards. In the revised manuscript (lines 247-249), we revised the sentences "During the observation period, 20-60% of hourly $PM_{2.5}$ concentration was higher than 75 μg m$^{-3}$ (the criterion for pollution according to the national air quality standards) in each month (Fig. S3A)"

Line 211: I am a bit confused by your explanation for there being more pollution episodes and higher concentrations of BC, CO and $PM_{2.5}$ in March, as it was the heating season. But isn't February just as cold? So why would there more heating in March?

**Response:** Thank you for your comments. In Beijing, the air quality is always determined by both the emission and meteorological conditions. In both February and March are in heating season. However, in March the meteorological condition is more favorable for haze formation due to low wind speed and low PBL height. This is a long sentence in the original manuscript. We revised it as "This can be explained by both the intensive emission during the heating season as evidenced by the high concentration of primary pollutants including CO, $SO_2$ and BC (Table S1) and the stagnant meteorological conditions supported by the low wind speed (<2 m s$^{-1}$) and the low planetary boundary layer (PBL) height, in particular, in March (Fig. S4A)" in lines 251-255 in the revised manuscript.

Line 222: Are the percentages listed for nitrate, chloride and ammonium also monthly means?
**Response:** Yes, they are monthly means. It has been pointed out in the revised manuscript. "At the same time, the monthly mean fraction of nitrate and chloride decreased from 26.7±8.8 % to 16.7±12.8 % and from 7.7±6.1 % to 0.3±0.2 %, respectively" in lines 260-262 in the revised manuscript.

Line 226: I am bit surprised that fireworks is regionally transported from Tangshan, are there no fireworks in Beijing?
**Response:** Thanks. According to the regulation on fireworks of Beijing government, firework burning is totally forbidden within the fifth ring road of Beijing. Based on back trajectory analysis of air masses, we found that firework burning in Tangshan should also contribute to the high mass concentration of chloride in Beijing during Chinese New Year.

Line 232-240: As example to one of my main comments above, I found that this paragraph was repeating much of the information presented in the preceding one. Perhaps these two paragraphs could be edited and combined.
**Response:** Thank you for your suggestion. We deleted the paragraph "It should be noted that the median mass concentrations of nitrate and OA also were higher in March than that in other months (Fig. S4C). The median mass concentrations of nitrate were 1.42, 8.76, 6.30, 3.15, and 3.23 μg m$^{-3}$ from February to June, respectively. And the corresponding OA concentrations were 4.78, 14.04, 11.64, 13.89, and 14.08 μg m$^{-3}$. Secondary formation is the major source of OA and nitrate in the atmosphere" in lines 232-237 in the original manuscript.

Line 272: why have you chosen to subset the data based on 'when the PM$_{2.5}$ concentration was larger than 50 μg m$^{-3}$ and the RH was less than 90 %'. Furthermore, as you state 'Under these conditions, local chemistry should be more important as 75 % of the wind speed was less than 1.0 m s$^{-1}$'. Why not then subset the data based solely on wind speed if local sources are of interest?

**Response:** Thank you for your comment. Because the aim of this paper is to understand the influence of HONO on secondary aerosol formation and the possible HONO source during pollution events. The dataset based solely on wind speed less than 1.0 m$^{-1}$ could be also meaningful to discuss local chemistry. However, there are around one third data with low wind speed (<1.0 m$^{-1}$) and low PM$_{2.5}$ concentration (< 50 μg m$^{-3}$). Therefore, the concentration of PM$_{2.5}$ was considered as one of the standards in this work. In the revised manuscript (lines 307-311), we revised it as "We simply compared the OH production via photolysis of HONO ($P_{OH\text{-}HONO}=J_{HONO}\times c_{HONO}$) and O$_3$ ($P_{OH\text{-}O3}=J_{O1D}\times c_{O3}$) in Fig. 2 when the PM$_{2.5}$ concentration was larger than 50 μg m$^{-3}$ and the RH was less than 90 % to understand the chemistry in pollution events".

Line 275: How where these maximal $P_{OH\text{-}HONO}$ and $P_{OH\text{-}O3}$ values calculated? I could not find the equation in the methods or reference.

**Response:** Thank you for your comment. They were calculated according to $P_{OH\text{-}HONO}=J_{HONO}*c_{HONO}$ and $P_{OH\text{-}O3}=J_{O1D}*c_{O3}$. In lines 307-311 in the revised manuscript, we defined them. "We simply compared the OH production via photolysis of HONO ($P_{OH\text{-}HONO}=J_{HONO}\times c_{HONO}$) and O$_3$ ($P_{OH\text{-}O3}=J_{O1D}\times c_{O3}$) in Fig. 2 when the PM$_{2.5}$ concentration was larger than 50 μg m$^{-3}$ and the RH was less than 90 % to understand the chemistry in pollution events". In addition, the $J$ values were also added in Fig. S7 and Fig. R2.

[Figure]

Fig. R2. The photolysis rate of $NO_2$, HONO, $O_3$ (O1D) and nitrate (middle value) from 8:00 am to 6:00 pm.

Line 304: Is it not the production of the OH that changes in winter relative to summer, rather than the rate of oxidation of VOC by OH? Please clarify

**Response:** Thank you. We mean OH production from photolysis of HONO will compensate the relative weak sunlight in winter. As you suggested above, we removed this redundant sentences (This implies that oxidation of atmospheric trace gases by OH may still be highly effective even in winter, thereby facilitating the vigorous formation of secondary pollutants in Beijing) in lines 301-304 in the original manuscript.

Line 318-20 and Fig 2: I am not so sure that is 'reasonable to mainly ascribe the increase of OA concentration to local secondary formation initiated by OH radical from HONO photolysis'. This is because if only OH from HONO photolysis was driving secondary formation, then shouldn't the OA/CO peak earlier, as the ambient HONO is essentially run out by 10am?

**Response:** Thank you for your instructive comment. During pollution events in winter in Beijing, the absolution HONO concentration was still above 0.6 ppb at noon (Fig. R3). The instrument automatically carried out zero point calibration twice per day during our observation. The measured HONO concentration at noon was much higher than the detection limit (10 ppt). Therefore, it reflected the real HONO concentration. Thus, it means that a photochemical steady state should be achieved between the daytime HONO sources and sinks and OH from photolysis of HONO should play an important role in initiation the $HO_x$ and $RO_x$ chemistry even after 10:00 am. In lines 350-353 in the revised manuscript, we revised this sentence to "…it was reasonable to ascribe the increase of OA concentration to local secondary formation initiated by OH radical and photolysis of HONO should play an important role in initiation the $HO_x$ and $RO_x$ chemistry".

[Figure]

Fig. R3. The diurnal curve of HONO concentration during pollution events in winter.

Line 324: But can't there also be anthropogenic sources of alkenes? For example, isoprene can also be from vehicle emissions (See e.g. Zou et al 2019).

**Response:** Thank you. We agree with you that vehicles can also emit isoprene. In August, 2019, we measured the isoprene using a GC-FID, the mean isoprene concentration was 0.35 ppb. And the mean concentration was 0.5±0.3 ppb at noon and 0.11±0.11 ppb in night, respectively. The nighttime isoprene concentration was comparable with that in winter (0.13 ppb) in Guangzhou (Zou et al., 2019). If we assume the isoprene concentration in winter being equivalent to the nighttime concentration in August, the SOA formation potential of isoprene is 0.015±0.015 μg m$^{-3}$. This contributes less than 0.5 % to the typical increase of OA concentration observed in this work. Therefore, we think the contribution of isoprene to the observed increase of OA concentration should be unimportant. In lines 358-360 in the revised manuscript, we added a sentence as "Although vehicles can emit isoprene (Zou et al., 2019), the contribution of isoprene to the observed increase of OA concentration should be unimportant due to the low concentration of isoprene in winter (Zou et al., 2019)".

Line 379: at the start of the sentence, you state that hourly NOx EI were available, yet than go to give a yearly emission factor? Why wasn't hourly EI used, and did you consider your measured NOx concentrations, as the diurnal variation in NOx would be important? Please clarify in more detail how the Vehicle was calculated Especially as emission inventories can have significant bias (See for example very recent work for Beijing by Squires et al, 2020).

**Response:** Thank you for your good suggestion. In this work, we used the hourly emission inventory of NO$_x$ (Fig. R4A) (Yang et al., 2019) to calculate the emission rate of HONO from vehicles. When calculating the emission rate of HONO, we converted the the hourly emission inventory (*EI*) to hourly emission flux (*F=EI/A*, where *A* is the area of Beijing). Then, the emission flux was normalized to the hourly mean PBL height according to Eq. (2). This resulted into the diurnal variation of NO$_x$. If the annual emission inventory was used to calculate the emission rate of HONO, the daily emission inventory was required to normalize to the measured NO$_x$ concentration to catch the diurnal variation. As shown in Fig. R4B, the emission rate of HONO using these two methods are overall comparable, but the daytime emission rate of HONO based on the hourly emission inventory is higher than that calculated using the annual emission inventory. We used the hourly emission inventory because it contained the traffic details on road such as emission factor of NO$_x$ for vehicle category, speed, traffic volume and congestion map and so on (Yang et al., 2019).

[Figure]

Fig. R4. (A) Hourly emission inventory of NOx from vehicles in Beijing (Yang et al., 2019) and calculated emission rate of HONO from vehicles using different methods.

To make it clearer, in lines 180-185 the revised manuscript, we added more details as "The emission rate ($E_{HONO}$, ppbv h$^{-1}$) was calculated based on the emission flux ($F_{HONO}$=$EI_{HONO}$/$A$, g m$^{-2}$ s$^{-1}$) and PBL height ($H$, m) according to the following equation,

$$E_{HONO} = \frac{a \cdot F_{HONO}}{H} \quad (2)$$

where, $EI_{HONO}$, is the emission inventory of HONO (g s$^{-1}$), $A$ is the urban area of Beijing (m$^2$), α is the conversion factor (α = $\frac{1 \times 10^9 \cdot 3600 \cdot R \cdot T}{M \cdot P} = \frac{2.99 \times 10^{13} \cdot T}{M \cdot P}$), $M$ is the molecular weight (g mol$^{-1}$), $T$ is the temperature (K) and $P$ is the atmospheric pressure (Pa)".

In lines 421- 423 in the revised manuscript, we revised the sentence "The $E_{vehicle}$ was calculated using the hourly NO$_x$ emission inventory from vehicles in Beijing (Yang et al., 2019) after converted to emission flux of HONO ($F_{HONO}$=$F_{NOx}$× HONO/NO$_x$) and the PBL height as described in Section 2.2"

We agree with you that a bias should exist for the emission inventory. According to the most recent work, the MEIC 2013 emissions inventory might significantly overestimate the emission of $NO_x$ in Beijing (Squires et al., 2020). In this work, we used the newest emission inventory (Yang et al., 2019) but not the MEIC 2013. We compared the wintertime emission flux of $NO_x$ calculated using the emission inventory from Yang et al. (2019) with the emission flux reported by Squires et al. (2020) (Fig. R5). The former emission flux is as 2.4±0.5 times as the later one. This will introduce an additional uncertainty to our estimation. In lines 712-716 in the revised manuscript, we added a paragraph "The source of HONO from vehicles was calculated based on the emission inventories, which should have a significant bias (Squires et al., 2020). For example, the emission flux of $NO_x$ calculated using the emission inventory from Yang et al. (2019) is as 2.4±0.5 times as the reported emission flux reported by Squires et al. (2020).".

[Figure]

Fig. R5. Comparison of the wintertime emission flux of $NO_x$ based on different emission inventory.

Line 381: This may be related to the above comment, but how did you report a range for calculated middle value of vehicle, when the NOx EI rate is constant and the HONO/NOx is constant? Furthermore, what does the middle value refer to? Daily avg? hourly avg? please specify. This applies throughout this section

**Response:** Thank you. As discussed above, the hourly emission of NOx reflected the diurnal variation vehicle emission. In addiction, the variation of PBL height was considered when we calculating the emission rate. Although the HONO/NOx was constant, the emission rate should show a diurnal variation. Three different emission rates of HONO were calculated because we chose three levels of HONO/NOx, i.e., the lower limit is 0.18% measured using a Chassis dynamometer test (Liu et al., 2017), 1.17% calculated using the low limit correlation of field data and 1.8 % using the empirical analysis of field data in this work. We called "middle value" for the emission rate calculated using HONO/NOx=1.17%. In lines 423-425 in the revised manuscript, we added a sentence "Thus the calculated emission rate reflected the diurnal variation of both the emission inventory and the PBL height". In lines 425-427 in the revised manuscript, the sentence was revised "The calculated hourly middle value of $E_{vehicle}$ using the HONO/NO$_x$ of 1.17% was from 0.085±0.038 to 0.34±0.15 ppbv h$^{-1}$, which was slightly higher than the daytime emission rate of HONO in Xi'an (Huang et al., 2017b)".

Line 386: the reported range for the upper limit is the same as reported for the lower limit, I'm guessing a typo?

**Response:** Thank you. From line 428 to 431 in the revised manuscript, "The lower limit of $E_{vehicle}$ was 0.013±0.006-0.053±0.023 ppbv h$^{-1}$, which was close to the estimated emission rate of HONO in Jinan (Li et al., 2018). The upper limit was in the range of 0.13±0.06-0.53±0.23 ppbv h$^{-1}$". The upper limit is one order of magnitude higher than the lower limit because the corresponding HONO/NOx is 0.18 % and 1.8%.

Line 389: Please give the reference for the emission flux you used, the value and also why grassland was the most appropriate.

**Response:** Thank you. Oswald et al. measured the emission flux of HONO from 17 soil samples, including eucalyptus forest, tropical rain forest, coniferous forest, pasture, woody savannah, grassland, stone desert, maize field, wheat field, jujube field an cotton field etc. (Oswald et al., 2013). Tropical rain forest, coniferous forest and grassland are the typical plants in downtown Beijing (Huang et al., 2017a). At the same time, their emission fluxes of HONO

are comparable (Oswald et al., 2013). Thus, we used the emission flux from grassland to calculate the emission rate of HONO from soil in Beijing because the temperature and water holding content dependent emission flux of grassland was available (Fig. R6). We added this paragraph "Oswald et al. (2013) measured the emission flux of HONO from 17 soil samples, including eucalyptus forest, tropical rain forest, coniferous forest, pasture, woody savannah, grassland, stone desert, maize field, wheat field, jujube field an cotton field etc. Tropical rain forest, coniferous forest and grassland are the typical plants in downtown Beijing (Huang et al., 2017a). At the same time, their emission fluxes of HONO are comparable (Oswald et al., 2013). Thus, we used the emission flux from grassland to calculate the emission rate of HONO from soil in Beijing because the temperature and water holding content dependent emission flux of HONO from grassland soil was available" in lines 433-441 in the revised manuscript.

[Figure]

Fig. R6. The emission flux of HONO from grassland at different temperature and water holding capacity (Oswald et al., 2013).

Line 397: why does only the middle values for $E_{soil}$ have uncertainty calculated? Also how did you estimate the uncertainty for $E_{soil}$? And why did you use a range of soil water content for lower, middle and upper, why not just use a single value?

**Response:** Thank you. We added the uncertainties of other values in lines 446-448 in the revised manuscript. "The lower limit, the middle value and the upper limit of the $E_{soil}$ are 0.0032±0.0027-0.013±0.014, 0.0046±0.0039-0.020±0.20 and 0.0057±0.0047-0.025±0.024 ppbv h$^{-1}$, respectively, calculated according to Eq. (2)". The uncertainty is the standard deviation when calculating the diurnal curve. We didn't measure the water content of the soil, while it should vary with seasons. Thus, we used a range of soil water content rather than a single value. For other sources, we also added all the uncertainties in the revised manuscript.

Line 416: Please provide more information on the night time temperature dependence of OH concentration, and the equations used in this calculation.

**Response:** Thank you. In the night, OH concentration usually varied from $1.0×10^5$ molecules cm$^{-3}$ (Li et al., 2012;Tan et al., 2018) in winter to $5×10^5$ molecules cm$^{-3}$ in summer (Tan et al., 2017). In the original manuscript, we linearly calculated the nighttime OH from $1×10^{-5}$ to $4×10^{-5}$ molecules m$^{-3}$ according to

$$c_{OH,night} = 1 \times 10^5 + 4 \times 10^5 \times \frac{T - T_{min,night}}{T_{max,night} - T_{min,night}} \quad (R1).$$

In the revised manuscript, we updated the calculation method as suggested by another reviewer. Because the nighttime OH is mainly generated from reaction of $O_3$ with alkenes measured with the SPIMS, we changed the estimation method using

$$c_{OH,night} = 1 \times 10^5 + 4 \times 10^5 \times \frac{(c_{O_3} \times c_{alkenes})_{night} - (c_{O_3} \times c_{alkenes})_{night,min}}{(c_{O_3} \times c_{alkenes})_{night,max} - (c_{O_3} \times c_{alkenes})_{night,min}} \quad (R2).$$

In lines 468-471 in the revised manuscript, we changed it as "The nighttime OH concentration was estimated linearly correlated with the product of nighttime $O_3$ concentration and alkenes concentration, namely,

$$c_{OH,night} = 1 \times 10^5 + 4 \times 10^5 \times \frac{(c_{O_3} \times c_{alkenes})_{night} - (c_{O_3} \times c_{alkenes})_{night,min}}{(c_{O_3} \times c_{alkenes})_{night,max} - (c_{O_3} \times c_{alkenes})_{night,min}} \quad (16)"$$

Line 419: Please give the reported OH concentrations by Li et al (2018) and Huang et al (2017) and if they were calculated or measured OH levels.

**Response:** Thank you. Here we compared the production rate of HONO from homogenous reaction between NO and OH among different researches but not OH concentration. In lines 475-478 in the revised manuscript, we added the corresponding $P_{NO-OH}$ as "The calculated middle value of $P_{\text{NO-OH}}$ (with mean daytime value of 0.49±0.35 ppb h$^{-1}$) was comparable with these estimated values by Li et al. (2018) (0.4 ppb h$^{-1}$) and Huang et al. (2017b) (0.28 ppb h$^{-1}$)".

Line 433: How was the HONO form nitrate photolysis calculated? Which equation (give number)? What do these ranges represent?

**Response:** Thank you. HONO formation from nitrate photolysis was calculated according to Eqs (3) and (8). Bao et al. reported the $J_{\text{nitrate}}$ at zenith angle of 0°. We normalized the $J_{\text{nitrate}}$ with the zenith angle at our observation station. The time series of the middle value was shown in Fig. R2 and was also added in Fig. S7. In lines 489-495 in the revised manuscript, we revised it. "A recent work reported the photolysis rate of nitrate ($J_{\text{nitrate}}$) in ambient PM$_{2.5}$ at a solar zenith angle of 0° (Bao et al., 2018). The $J_{\text{nitrate}}$ varied from $1.22 \times 10^{-5}$ to $4.84 \times 10^{-4}$ s$^{-1}$ with the mean value of $8.24 \times 10^{-5}$ s$^{-1}$. These values were further normalized according to the zenith angle and UV light at our observation station to calculate the low limit, the upper limit and the middle $J_{\text{nitrate}}$. The time series of the measured nitrate concentration and the middle value of $J_{\text{nitrate}}$ were shown in Fig. 1 and Fig. S7, respectively". In addition, the equation numbers was pointed out in line 498 in the revised manuscript. "The corresponding daytime lower limit, the middle value and the upper limit of HONO from photolysis of nitrate were 0.0011±0.0021-0.096±0.092, 0.0072±0.0021-0.66±0.092 and 0.042±0.082-3.86±0.008 ppbv h$^{-1}$, respectively, calculated in the light of Eqs. (3) and (8)".

Line 447: by the end of this paragraph, it was not at all clear to me which uptake co-efficient you actually used. Please clarify.

**Response:** Thank you. In the dark, the low limit, middle value and upper limit of $P_{\text{aerosol}}$ were calculated using the RH dependent $\gamma_{\text{NO2}}$ on kaolin ($4.47 \times 10^{39}/(1.75 \times 10^{46} + 1.93 \times 10^{45}\text{RH})$) (Liu et al., 2015), the fixed $\gamma_{\text{NO2}}$ ($1.2 \times 10^{-8}$) recommended by Crowley et al. (Crowley et al., 2010) and the RH dependent $\gamma_{\text{NO2}}$ on kaolin on hematite ($\gamma_{\text{NO2}} = 4.46 \times 10^{39}/(6.73 \times 10^{44} + 3.48 \times 10^{44}\text{RH})$) (Liu et al., 2015), respectively, along with the $\gamma_{\text{NO2}}$ on black carbon ($1.17 \times 10^{-5}$). In the daytime, the light enhanced uptake $\gamma$ of NO$_2$ ($1.9 \times 10^{-6}$) on mineral dust was parameterized (Ndour et al., 2008) after normalized to the solar radiation intensity in Beijing.

For $P_{ground}$, the low limit, middle value and upper limit of $P_{ground}$ were calculated using the same $\gamma_{NO2}$ as $P_{aerosol}$ in night, while $\gamma_{NO2}$ of $NO_2$ on urban regime ($\gamma_{NO2}$=7.4× $10^{-7}$+5.5×$10^{-8}$ RH) (Liu et al., 2019) was used after normalized to the light intensity at BUCT in the daytime. In the revised manuscript, we pointed out these equations as "($\gamma_{NO2}$=4.47 × $10^{39}$/(1.75 × $10^{46}$ + 1.93 ×$10^{45}$RH)", "($\gamma_{NO2}$=4.46 × $10^{39}$/(6.73 × $10^{44}$ + 3.48×$10^{44}$ RH)" and "($\gamma_{NO2}$=7.4× $10^{-7}$+5.5×$10^{-8}$ RH)" in lines 507-508, 509 and 521, respectively.

Line 477-9: the authors state that 'Heterogeneous reactions of $NO_2$ on aerosol surface and ground surfaces were unimportant compared with other sources because of the very low uptake coefficient'. What do you mean by the very low uptake coefficient, low compared to what? Is the issue more that you used the wrong co-efficient?

**Response:** Thank you. For heterogeneous reaction of $NO_2$ on aerosol surface, the production rate is determined by the uptake coefficient according to Eqs. (3)-(5). Modelling studies have found that a given chemical process should be important in the tropospheric chemistry if the uptake coefficient of a trace gas on particles is greater than $10^{-5}$ (Zhang and Carmichael, 1999;Zheng et al., 2015). As discussed in this work, the typical uptake coefficient of $NO_2$ on aerosol is on the order of $10^{-7}$-$10^{-8}$. It was it was recommended to be 1.2×$10^{-8}$ (Crowley et al., 2010). Furthermore, we also performed laboratory studies about uptake of $NO_2$ on kaolin, hematite and soot particles. The uptake coefficient on mineral dust is on the order of $10^{-7}$-$10^{-8}$ (Liu et al., 2015). In addition, we found that the $\gamma_{NO2,\,BET}$ at steady state (or after aged in air) was one order of magnitude lower than that of fresh sample. Therefore, we chose the $\gamma_{NO2,\,BET}$ $10^{-7}$-$10^{-8}$ in this work. It was lower than that ($10^{-6}$) used in modeling studies (Zhang et al., 2016;Aumont et al., 2003).

We double checked the parameters for budget calculation. We found a bug when calculating the heterogeneous reaction of $NO_2$ on black carbon. A conversion factor of time from second to hour was missed. So, the contribution of heterogeneous reaction to HONO source was underestimated. Now, the $P_{aerosol}$ was 0.038±0.030-0.088±0.072. It was on the same orders as soil emission. In the revised manuscript, we updated Figures 3-5 and the corresponding numbers in section 3.3.

In lines 544-554 the revised manuscript, we added a paragraph "It should be pointed out that HONO production from heterogeneous reaction of $NO_2$ on both aerosol and ground surface greatly depend on the $\gamma_{NO2, BET}$ and $A_s$. The $A_s$ of aerosols was comparable with the modeling input. However, the small nighttime $\gamma_{NO2, BET}$ ($10^{-8}$ - $10^{-7}$) on dust were used in this work rather than the $\gamma_{NO2, BET}$ ($1\times10^{-6}$) used in modelling studies (Zhang et al., 2016;Aumont et al., 2003;Gall et al., 2016). This leads to a lower production rate of HONO from heterogeneous reaction of $NO_2$ on aerosols. As for heterogeneous reaction of $NO_2$ on ground surface, besides the small $\gamma_{NO2, BET}$ used in this work, the $A_s$ of ground surface (0.0015 to 0.0385 $m^{-1}$) calculated using the surface roughness and PBL height was also significantly lower than the fixed value of 0.3 $m^{-1}$ in modeling studies that might overestimate the contribution of HONO production from heterogeneous reaction of $NO_2$ on ground surface". In lines 558-561 in the revised manuscript, we revised the sentences as "These results mean that heterogeneous reaction might not be a major HONO source. This is consistent with a recent work that heterogeneous reaction should be unimportant when compared with traffic emission during haze events in winter in Beijing (Zhang et al., 2019)". And in lines 605-607 in the revised manuscript, we also revised the sentence "Heterogeneous reactions of $NO_2$ on aerosol surface and ground surfaces were not the major HONO source during night unlike the modelled results (Zhang et al., 2016;Aumont et al., 2003).".

Fig 2D: if you take the bottom and top points in Feb/Mar (blue), I am not sure there this a correlation. It would be good to check if you get a similar slope and r2 without these 2 points. **Response:** Thank you. If we remove these two points as you suggested, the correlation coefficient will decrease from 0.74 to 0.31. A positive correlation can be still observable when the uncertainty is taken into consideration (Fig. R7). We think it is unreasonable to remove them because these data points are valid. We agree with you that it should be better if more data points are available. This will be further investigated in the future.

[revised manuscript text omitted]

---

## Author Comment (AC2) · 18 May 2020

**Reviewer 2#**

This paper reports the possible promotion effect of nitrous acid (HONO) on the formation of secondary organic aerosol (SOA) and nitrate in winter based on a five-month comprehensive observation in urban Beijing. Evidence for the relationship between secondary aerosol formation and the consumed HONO was obtained from the observations. The detailed budget of ambient HONO was explored, and vehicle emission was proposed as a significant source of HONO in Beijing. Overall, the manuscript is logically organized and well written, and the measurement data are much valuable. I would like to recommend that it can be considered for publication after the following major and specific comments being properly addressed.

**Response:** Thank you so much for your positive comments.

Section 2.2. HONO budget calculation: the description of the budget calculation is not clear enough. Firstly, the method used in this study was the budget analysis, other than the stationary state analysis. For a thorough budget analysis, physical terms (such as vertical and horizontal transport) should be considered for equations (1) and (12). At least, the authors need to evaluate if the physical processes were negligible for the analysis presented here. Secondly, a number of parameters (e.g., $F_{HONO}$, $c_{OH}$, gamma, $Y_{HONO}$, $J_{nitrate}$, $J_{HONO}$, etc.) are required for the analysis. It is not clear how these parameters were obtained or approximated in this study. Although they were described more or less in the following Section 3.3, the authors may need clearly state the data source and uncertainties of these key parameters at their first appearance. This may help the readers better understand the overall methodology of this study.

**Response:** Thank you for your good suggestions. We revised the "stationary state HONO concentration" to "HONO budget" in line 171 in the revised manuscript.

The vertical and horizontal transport have been added in Eq. (1) and (12) from line 171 to 179 in the revised manuscript. "The HONO budget could be calculated by,

$$\frac{dc_{HONO}}{dt} = E_{HONO} + P_{HONO} - L_{HONO} + T_{vertical} + T_{horizontal} \qquad (1)$$

where $\frac{dc_{HONO}}{dt}$ is the observed change rate of HONO mixing ratios (ppbv h$^{-1}$); $E_{HONO}$ represents the emission rate of HONO from different sources (ppbv h$^{-1}$); $P_{HONO}$ is the in-*situ* production rate of HONO in the troposphere (ppbv h$^{-1}$); $L_{HONO}$ is the loss rate of HONO (ppbv h$^{-1}$) (Li et al., 2018); $T_{\text{vertical}}$ and $T_{\text{horizontal}}$ are the vertical and horizontal transport (Soergel et al., 2011), which can mimic source or sink terms depending on the HONO mixing ratios of the advected air relative to that of the measurement site and height (Soergel et al., 2011)".

In addition, we added two paragraphs to discuss the transport in lines 222-232 "Vertical transport by advection ($T_{\text{vertical}}$), which is an important sink of HONO in the night (Gall et al., 2016;Meng et al., 2019), can be calculated according to equation (12).

$$T_{vertical} = -K_h(z,t)\frac{\partial c(z,t)}{\partial z}\frac{1}{h} \quad (12)$$

where $K_h(z,t)$ is the eddy diffusivity of heat (m$^2$ s$^{-1}$) at height $z$ (m) and time $t$, $h$ is the height of the second layer (18 m in this study) (Gall et al., 2016). On the other hand, both the vertical and horizontal transport can be estimate according to Eq. (13),

$$T_{\text{vertical}} = k_{\text{dilution}}(c_{\text{HONO}} - c_{\text{HONO,background}}) \quad (13)$$

where $k_{\text{dilution}}$ is a dilution rate (0.23 h$^{-1}$, including both vertical and horizontal transport) (Dillon et al., 2002), $c_{\text{HONO}}$ and $c_{\text{HONO,background}}$ is the HONO concentration at the observation site and background site, respectively (Dillon et al., 2002)" and lines 574-598 "As pointed in Section 2.2, vertical transport by advection is an important nocturnal sink of HONO (Gall et al., 2016). In this work, the vertical distribution of HONO concentration is unavailable. Recently, Meng et al. (2019) measured the vertical distribution of HONO in Beijing in December, 2016. The concentration of HONO showed nearly flat profiles from ground level to 240 m in pollution events after sunset, while negative profiles of HONO were observed in pollution events during night (Meng et al., 2019). The nighttime concentration gradient was 0.0047±0.0025 ppb m$^{-1}$ derived from the nighttime dataset (Meng et al., 2019). In the daytime, we assume a zero concentration gradient. On the other hand, the eddy diffusivity of heat in urban environment was measured in New Delhi, Indian (Yadav et al., 2003). Using their dataset with the wind speed lower than 2.0 m s$^{-1}$, we derived the relationship between the $K_h$ and the wind speed (WS) ($K_h$=0.9389×WS-0.3374 m$^2$ s$^{-1}$). The nighttime $T_{\text{vertical}}$ changed from 0.15 to 0.37 ppbv h$^{-1}$ in winter, while it was from 0.12 to 0.68 ppbv h$^{-1}$ according to Eq. (12) from April to June. Because the wind speed was usually lower than 1.0 m s$^{-1}$ in pollution events (Fig. S6), horizontal transport should have little influence on the daytime sources or sinks of HONO because of the short lifetime of HONO. In the night, 79 % of the wind speed was lower than

1.0 m s$^{-1}$ in winter, thus the air masses from suburban areas should have influence on the sources and sinks of HONO in Beijing. If the HONO concentration at background was zero, the vertical and horizontal transport rate of HONO varied from 0.17 to 0.61 ppbv h$^{-1}$ which is calculated in the light of Eq. (13) on haze days in winter and from 0.15 to 0.74 ppbv h$^{-1}$ in pollution events from April to June. These values were higher than that calculated according to Eq. (12). Because the background HONO concentration was unavailable, we only considered the nighttime transport calculated according to Eq. (12) in the following section". At the same time, the relative contribution of each source in Section 3.3 was updated based on the total sinks.

We added a sentence "The $E_{vehicle}$ was calculated using the hourly NO$_x$ emission inventory from vehicles in Beijing (Yang et al., 2019) after converted to emission flux of HONO ($F_{HONO}=F_{NOx}\times$ HONO/NO$_x$) and the PBL height as described in Section 2.2" to make it clearer about $F_{HONO}$ in lines 421-423 in the revised manuscript.

The OH concentration was estimated based on $J_{O1D}$ (Tan et al., 2019;Tan et al., 2018a) or $J_{O1D}$ and $J_{NO2}$ (Li et al., 2018). Photolysis rate constants of NO$_2$($J_{NO2}$), HONO($J_{HONO}$) and O$_3$($J_{O3}$) for clear sky conditions were calculated according to the solar zenith angle and the location using a box model (FACSIMILE 4). NO$_2$ photolysis sensor ($J_{NO2}$, Metcon) was unavailable during our observation study. However, it was available from Aug 17 to Sep 16, 2018. Calibration function between the measured UVB light intensity and $J_{NO2}$ from Aug 17 to Sep 16, 2008 was established to correct the influence the climatological O$_3$ column, aerosol optical depth and cloud cover on surface UV light intensity. As shown in Figure S10A, the model well predicted the $J_{NO2}$. Then the $J_{NO2}$ during this campaign study was predicted using the model. The details about photolysis rates are shown in the Supplement Information. In lines 458-460 in the revised manuscript, we added a sentence "The method for the photolysis rates calculation were shown in SI and the time series of the photolysis rates were shown in Fig. S7". In lines 472-473 in the revised manuscript, we added a sentence "The time series of OH concentration calculated using different methods was shown in Fig. S11".

Three kinds of OH concentration was used in this work. In lines 460-466 in the revised manuscript, we revised it as "On polluted days, high concentration of NO$_2$ resulted into lower OH concentrations estimated using the Eq. (13). Therefore, the corresponding $P_{NO-OH}$ was taken as the low limit for homogeneous reaction between NO and HONO because polluted events were discussed in this work, while $P_{NO-OH}$ calculated using the OH concentration ($J_{O1D} \times 4.33 \times 10^{11}$ molecules cm$^{-3}$) (Tan et al., 2018a) was taken as the upper limit and $P_{NO-OH}$ calculated using the OH concentration ($J_{O1D} \times 2 \times 10^{11}$ molecules cm$^{-3}$) (Tan et al., 2019) was the middle value".

As for $J_{nitrate}$, we added the number as "The $J_{nitrate}$ varied from $1.22 \times 10^{-5}$ to $4.84 \times 10^{-4}$ s$^{-1}$ with the mean value of $8.24 \times 10^{-5}$ s$^{-1}$" in line 490-491 in the revised manuscript. About $\gamma_{NO2}$ on aerosol and ground surface, we pointed out from line 502 to 510 in the revised manuscript "The uptake coefficient ($\gamma$) of NO$_2$ on different particles varied from $5 \times 10^{-9}$ to $9.6 \times 10^{-6}$ (Ndour et al., 2009;Underwood et al., 2001;Underwood et al., 1999), while it was recommended to be $1.2 \times 10^{-8}$ (Crowley et al., 2010), which was used to calculate the $P_{aerosol}$ in the base case. It has been found that the $\gamma$ highly depends on the relative humidity (RH). The low limit bound of $P_{aerosol}$ was calculated based on the RH dependent uptake coefficient of NO$_2$ on kaolinite ($\gamma_{NO2}=4.47 \times 10^{39}/(1.75 \times 10^{46} + 1.93 \times 10^{45} RH)$, while the upper limit of $P_{aerosol}$ was calculated according to the RH dependent $\gamma$ on hematite ($\gamma_{NO2}=4.46 \times 10^{39}/(6.73 \times 10^{44} + 3.48 \times 10^{44} RH)$ (Liu et al., 2015)". In the night, $Y_{HONO}$ was set to 0.5 because of the hydrolysis reaction of NO$_2$, while it was 0.8 for light enhanced reaction. It has been added "The $Y_{HONO}$ was set to 0.5 because of the hydrolysis reaction of NO$_2$ (Liu et al., 2015), while it was 0.8 for light enhanced reaction (Liu et al., 2019a;Ndour et al., 2008) and on BC (Han et al., 2013)" in lines 528-530 in the revised manuscript.

Section 3.3 and Figures 3&4: following the first comment, the description and discussion of the HONO budget are also not clear and need clarification. The contributions of heterogeneous reactions of NO$_2$ on aerosol and ground surfaces were too low, and they were even lower than the OH+NO reactions during the nighttime. This is unusual. Is it reasonable to approximate the nighttime OH concentrations linearly with the temperature? Furthermore, the heterogeneous reactions of NO$_2$ on aerosol and ground surfaces were highly dependent on the NO$_2$ concentrations, the uptake coefficients, and surface density, some of which are highly uncertain. It is not clear what values were actually adopted for the uptake coefficients of NO$_2$ on aerosol and ground surfaces, and how much were the ambient NO$_2$ levels and surface density? More details about the calculation of HONO budget are needed.

**Response:** Thank you for your good suggestion and comments. HONO production from heterogeneous reactions of $NO_2$ on both aerosol surface and ground surface greatly depend on the uptake coefficient ($\gamma_{NO2, BET}$) and the surface to volume ratio (S/V). The mean and median surface to volume ratio of aerosols are $1.33\times10^{-3}$ and $1.36\times10^{-3}$ $m^{-1}$ during pollution events. This is comparable with the input parameters in modeling studies (Zhang et al., 2016). However, as discussed from line 501 to 503 in the revised manuscript, the $\gamma_{NO2, BET}$ on aerosols varied from $5\times10^{-9}$ to $9.6\times10^{-6}$. The $\gamma_{NO2, BET}$ on pure oxides was usually higher that on composite oxides. The selection of a proper $\gamma_{NO2, BET}$ is quite tricky. For example, a fixed $\gamma_{NO2, BET}$ was set to $1\times10^{-6}$ in nighttime and $2\times10^{-5}$ in daytime due to photochemical reaction of $NO_2$ on soot surface in a modelling study (Zhang et al., 2016;Aumont et al., 2003). The nighttime $\gamma_{NO2, BET}$ is ~2 orders of magnitude higher than ours ($1.2\times10^{-8}$) as suggested by Crowley et al. (Crowley et al., 2010). In our previous work, we have measured the $\gamma_{NO2, BET}$ on kaolin and hematite (Liu et al., 2015) and soot (Han et al., 2013). The initial $\gamma_{NO2, BET}$ on kaolin is $4.85\pm0.39\times10^{-8}$ at 47 % RH. It should be pointed out that the $\gamma_{NO2, BET}$ decreases significantly with exposure time due to surface saturation. Fig. R1 shows the typical uptake curve of $NO_2$ and the $\gamma_{NO2, obs}$ which is not normalized to the specific surface area. The $\gamma_{NO2, BET}$ at steady state ($2.56\times10^{-9}$ to $4.56\times10^{-9}$ on kaolin and $1.23\times10^{-8}$ to $1.50\times10^{-8}$ on hematite) is around one order of magnitude lower than the initial $\gamma_{NO2, BET}$ as shown in Fig. R2. Therefore, we used the recommended value ($1.2\times10^{-8}$) (Crowley et al., 2010) in the base case. On the other hand, the $\gamma_{NO2, BET}$ decreases with RH due to competitive adsorption (Liu et al., 2015). High mass concentration of $PM_{2.5}$ usually accompanied with high RH in winter in Beijing. Thus, we calculated the RH dependent $\gamma_{NO2, BET}$ according to the equation we determined previously (Liu et al., 2015) and the measured ambient RH in this work. This should be more reasonable than a fixed uptake coefficient used in modeling studies.

[Figure]

Fig. R1. Uptake curves of NO₂ on (A) kaolin and (B) hematite; and evolution of the observed uptake coefficient on (C) kaolin and (D) hematite at 298 K and at 47 % of RH    (Liu et al., 2015).

[Figure]

Fig. R2. The dependence of $\gamma_{NO2,BET}$ on relative humidity (Liu et al., 2015).

In the original version of manuscript, we found a bug when calculating the heterogeneous reaction of $NO_2$ on black carbon. A conversion factor of time from second to hour was missed. So, the contribution of heterogeneous reaction to HONO source was underestimated in the original version. Now, the $P_{aerosol}$ was 0.038±0.030-0.088±0.072, which was dominated by the heterogeneous reaction of $NO_2$ on black carbon. It was on the same orders as soil emission. However, this was still significantly lower than the contribution of vehicle emission. In the revised manuscript, we updated Figures 3-5 and the corresponding numbers in section 3.3.

As for heterogeneous reaction of $NO_2$ on ground surface, we used the same $\gamma_{NO2,BET}$ on dust aerosol in the night, which is similar to the methodology used in modelling studies (Zhang et al., 2016;Aumont et al., 2003). As pointed out in Section 2.2 from line 197 to 209 in the revised manuscript, the S/V was estimated using the surface roughness calculated in the light of satellite image of Beijing. The calculated surface roughness is 3.85, which is slightly higher than the value (2.2) used by Li et al. (2017). Thus, the calculated S/V varied from 0.0015 to 0.0385 $m^{-1}$ (with a mean value of 0.0125 $m^{-1}$) because of the variation of the PBL height during pollution events. However, a fixed S/V of ground surface was set to 0.3 $m^{-1}$ in the modeling studies (Zhang et al., 2016). This corresponds to a surface roughness ~92, which means the surface area is ~92 times of the projected area of ground. It's too high. If both the $\gamma_{NO2,BET}$ ($1\times10^{-6}$) and surface roughness are increased to the values used in modeling studies, the nighttime production rate of HONO via heterogeneous reaction of $NO_2$ on ground surface will be 2.9 ppb $h^{-1}$. This means a large sink missed if this number is reasonable. In lines 558-561 in the revised manuscript, we also revised the sentence "These results mean that heterogeneous reaction might not be a major HONO source. This is consistent with a recent work that heterogeneous reaction should be unimportant when compared with traffic emission during haze events in winter in Beijing (Zhang et al., 2019c)" and in lines 605-607 in the revised manuscript, "Heterogeneous reactions of $NO_2$ on aerosol surface and ground surfaces were not the major HONO source during night unlike the modelled results (Zhang et al., 2016;Aumont et al., 2003) ".

During the Chinese New Year (CNY) and the COVID-19 event in 2020, traffic emission decreased significantly in Beijing. This provides us a unique opportunity to verify the relative importance of each HONO source. We only analyzed the nocturnal data from January 1 to

February 29 because the HONO sources related to photochemical reactions could be avoided. The CNY vacation was from January 23 to February 2. These results are in preparation for a separate paper. Fig. R3 shows the relative change of the concentrations of HONO, $NO_x$ and non-refractory $PM_{2.5}$ (NR-$PM_{2.5}$). The traffic index and the chemical age of the air masses which is defined as -log($NO_x/NO_y$) are also shown in Fig. R3. $NO_y$ was measured with a $NO_y$ analyzer (Thermo 42i-Y). The concentration of nighttime HONO decreased significantly during the CNY and after the CNY accompanied with the reduction in vehicle emission as supported by both the concentration of $NO_x$ and traffic index (Fig. R3B). Interestingly, the NR-$PM_{2.5}$ concentration during and after the CNY increased obviously when compared with that before the CNY. The effective conversion of $NO_2$ aerosol surface was almost constant because both the promotion effect of increased $PM_{2.5}$ concentration and the inhibition effect of reduced $NO_2$ concentration during and after the CNY as shown in Fig. R3C. On the other hand, we found that the nighttime chemical age of the air masses during and after the CNY was also obviously larger than that before the CNY (Fig. R3D). This means that heterogeneous reaction of $NO_2$ on both aerosol surface and ground surface should be more effective due to longer residence of the air masses during and after the CNY than that before the CNY. When the reduction of $NO_2$ concentration was taken into consideration, the product of $NO_2$ concentration and –log($NO_x/NO_y$) in COVID-19 epidemic periods increased slightly (Fig. R4). Therefore, the observed HONO concentration should decreased in COVID-19 epidemic or at least be constant in different periods and be independent on the reduction of vehicle emission if heterogeneous reaction of $NO_2$ on aerosol and ground surfaces dominates nighttime HONO source or if the vehicle emission is a minor HONO source. However, we observed the decrease of HONO along with reduction of vehicle emission (Fig.R3A). This well supports our (and other researchers' (Meng et al., 2019;Zhang et al., 2019c) conclusion that vehicle emission is an important source of HONO in Beijing.

[Figure]

Fig. R3. Relative change of nighttime (A) HONO concentration, (B) NOx concentration and the traffic index, (C) non-refractory PM$_{2.5}$ (NR-PM$_{2.5}$) concentration and (D) relative chemical age (-log(NO$_x$/NO$_y$)) of air masses before Chinese New Year (CNY) (2020.1.1-2020.1.22), during CNY (2020.1.23-2020.2.1) and after CNY (2020.2.2-2020.2.29).

[Figure]

Fig. R4. Relative change of nighttime the product of NO$_2$ and NR-PM$_{2.5}$ concentration and the product of $NO_2$ concentration and $-\log(NO_x/NO_y)$ in different periods.

In lines 523-528 in the revised manuscript, we pointed out "The $A_s$ of aerosols which was measured using a DMPS varied from $1\times10^{-4}$ to $4.8\times10^{-3}$ $m^{-1}$ with a mean value of $1.4\pm0.5\times10^{-3}$ $m^{-1}$ during pollution events. This value is comparable with that used in modeling studies (Zhang et al., 2016;Aumont et al., 2003). The $A_s$ of ground surface which was calculated according to Eq. (6) and (7) varied from $1.5\times10^{-3}$ to $3.85\times10^{-2}$ $m^{-1}$ with a mean value of $1.3\pm0.9\times10^{-2}$ $m^{-1}$ during pollution events. The surface roughness was 3.85 calculated according to Eq. (7)". And from line 544 to 554 in the revised manuscript, we added a paragraph to discuss the reason why we obtained the low production rate of HONO via heterogeneous reaction of $NO_2$ on ground and aerosol surfaces as "It should be pointed out that HONO production from heterogeneous reaction of $NO_2$ on both aerosol and ground surface greatly depend on the $\gamma_{NO2, BET}$ and $A_s$. The $A_s$ of aerosols is comparable with the modeling input. However, the small nighttime $\gamma_{NO2, BET}$ ($10^{-8} - 10^{-7}$) were used in this work rather than the $\gamma_{NO2, BET}$ ($1\times10^{-6}$) used in modelling studies (Zhang et al., 2016;Aumont et al., 2003). This leads to a lower production rate of HONO from heterogeneous reaction of $NO_2$ on aerosols. As for heterogeneous reaction of $NO_2$ on ground surface, besides the small $\gamma_{NO2, BET}$ used in this work, the $A_s$ of ground surface (0.0015 to 0.0385 $m^{-1}$) calculated using the surface roughness and PBL height was also significantly lower than the fixed value of 0.3 $m^{-1}$ used in modeling studies that might overestimate the contribution of HONO production from heterogeneous reaction of $NO_2$ on ground surface".

As for nighttime OH concentration, we revised it to using the proxy of alkene and $O_3$ concentration to normalize it from $1\times10^5$ to $4\times10^5$ molecules $cm^{-3}$. In lines 468-472 in the revised manuscript, we revised it as "The nighttime OH concentration was estimated linearly correlated with the product of nighttime $O_3$ concentration and alkenes concentration, namely,

$$c_{OH,night} = 1 \times 10^5 + 4 \times 10^5 \times \frac{(c_{O_3} \times c_{alkenes})_{night} - (c_{O_3} \times c_{alkenes})_{night,min}}{(c_{O_3} \times c_{alkenes})_{night,max} - (c_{O_3} \times c_{alkenes})_{night,min}} \quad (16)$$

The time series of OH concentration calculated using different methods was shown in Fig. S11" At the same time, we updated the data in Figs. 3-5 in the revised manuscript.

The authors attempted to quantify the contributions of vehicle-emitted NO to ambient HONO

via NO+OH reaction based on the source apportionment of NO emissions. This is not convincing because the homogeneous HONO formation is generally limited by OH other than NO. This means that the produced HONO should be not linearly dependent to the NO emissions. It can be concluded that vehicle emission should contribute significantly to not only direct HONO emission but also HONO formation through reactions of NO and $NO_2$. However, the current quantification analysis needs be more careful.

**Response:** Thank you so much for your comment. We agree with you that the production rate of HONO ($P_{NO-OH}$) from this reaction is greatly dependent on or determined by OH concentration as shown in Fig. 3 because the variation degree of OH concentration from nighttime to daytime is obviously larger than that of NO concentration. However, we think $P_{NO-OH}$ actually reflects both the variations of OH concentration and the emission of NO from vehicles. From the point view of HONO sources, it still represents the indirect source of HONO related to traffic emission although the diurnal variation of OH concentration is mainly determined by light intensity. In the revised manuscript, we added a sentence to discuss this point "As shown in Fig.3, although the diurnal curve of $P_{NO-OH}$ coincided well with that of OH concentration (Fig. S10), which means the $P_{NO-OH}$ should be mainly determined by OH concentration, the $P_{NO-OH}$ should still reflect the indirect contribution of traffic related emission to HONO source because the ambient NO concentration was used to calculate the $P_{NO-OH}$." in lines 637-641 in the revised manuscript.

Specific Comments:

Line 44: fine particulate matter with diameter less than or equal to 2.5…

**Response:** Thank you. It has been corrected in line 44 in the revised manuscript.

Line 78: on polluted days

**Response:** Thank you. It has been corrected throughout the paper.

Lines 84-94: to my knowledge, there have been a number of observational studies of HONO in recent years in China, and similar HONO budget analyses were performed. I suggest the authors to comprehensively review the existing results about the HONO sources in China and compare them against the source analysis results obtained in the present study.

**Response:** Thank you for you good suggestion. In the revised manuscript, we added two paragraph to review the previous results. "The HONO concentration has been measured with a wide rang from 0.18 to 9.71 ppbv at different locations, such as Beijing (Zhang et al., 2019c;Hu et al., 2002;Hendrick et al., 2014;Wang et al., 2017), Shanghai (Wang et al., 2013;Zhang et al., 2019a), Guangdong (Hu et al., 2002;Su et al., 2008a), Hongkong (Xu et al., 2015), Shandong (Li et al., 2018), Xi'an (Huang et al., 2017) and so on in China since 2000" in lines 85-89 and "At the present time, the study of the HONO budget is still far from closed, which would require a significant effort on both the accurate measurement of HONO and the determination of related kinetic parameters for its production pathways (Liu et al., 2019b). For example, photo-enhanced conversion of $NO_2$ (Su et al., 2008b) and photolysis of particulate nitrate were found to be the two major mechanisms with large potential of HONO formation during noontime, but the associated uncertainty may reduce their importance (Liu et al., 2019b). Some other researches proposed that heterogeneous reactions on ground/aerosol surfaces were important during nighttime (Wang et al., 2017;Zhang et al., 2019b) and daytime in Beijing-Tianjin-Hebei (BTH) (Zhang et al., 2019b). But the heterogeneous reaction was unimportant in Ji'an compared with the unknown sources and the homogeneous reaction between NO and OH (Li et al., 2018). In addition, the traffic emission was proposed to be an important HONO source during nighttime but not significant during daytime in BTH (Zhang et al., 2019b). However, it was proposed that direct emission of HONO from vehicles should contribute about 51.1 % (Meng et al., 2019) and 52 % of nighttime HONO in Beijing (Zhang et al., 2019c). These results mean that more studies is still required on the HONO budget" in lines 99-115.

Line 138: replace "nitrous acid" by "nitric acid"

**Response:** Thank you. It has been corrected in line 81 in the revised manuscript.

Line 146: Equation (1) describes the budget analysis other than the stationary state analysis. Both methods are different. Transport terms need be considered here.

**Response:** Thank you. It has been corrected in line 171 in the revised manuscript.

Lines 198-202 and Fig. 1: it would be much better if the authors could also plot the other related parameters, such as NOx and meteorological parameters, in Fig. 1. It is difficult for the readers to look at the same measurements separately from main text and supplement.

**Response:** Thank you for your suggestion. We have combined these parameters in Fig. 1 and Fig. R5.

[Figure]

Fig. R5. An overviewed measurement of non-refractory-PM$_{2.5}$ (NR-PM$_{2.5}$), HONO, NOx, PM$_{2.5}$ and meteorological parameters from Feb. 1 to July 1, 2018. (A) the mass concentration of different components of PM$_{2.5}$, (B) the mass fraction of individual component, (C) HONO and NOx concentration, (D) temperature and RH, (E) wind speed and wind direction, (F) UVB and PBL height and (G) visibility and PM$_{2.5}$ concentration during observation.

Line 228: the increase in temperature…

**Response:** Thank you. It has been corrected in line 268 in the revised manuscript.

Line 262: what does "in RO$_2$ chemistry" mean? Rephrase this sentence.

**Response:** Thank you. We revised it to "the reaction between NO and HO$_2$" in line 298 in the revised manuscript. It means the reaction: HO$_2$ + NO → OH + NO$_2$.

Line 266: replace "dominating" by "dominant"

**Response:** Thank you. It has been corrected in line 303 in the revised manuscript.

Lines 270-282: it is not clear how the $P_{OH-HONO}$ and $P_{OH-O3}$ were calculated from the current discussion. The calculated $P_{OH-HONO}$ and $P_{OH-O3}$ levels in winter and April-June seem to be too high. Detailed calculation methods should be given here. Usually, the OH+NO reactions should be subtracted from the photolysis of HONO to denote the real contribution of HONO to the OH source.

**Response:** Thank you for your good suggestions. The photolysis rates of $NO_2$($J_{NO2}$), HONO($J_{HONO}$) and $O_3$($J_{O3}$) under clear sky conditions were calculated according to the solar zenith angle and the location using a box model (FACSIMILE 4). $NO_2$ photolysis sensor ($J_{NO2}$, Metcon) was unavailable, while UVB is always available during our observation study. However, the $J_{NO2}$ sensor was available from Aug 17 to Sep 16, 2018. A calibration function between the measured UVB light intensity and the $J_{NO2}$ was established to correct the influence the climatological $O_3$ column, aerosol optical depth and cloud cover on surface UV light intensity from Aug 17 to Sep 16, 2008. As shown in Figure S10, the model well predicted the $J_{NO2}$. Then, the $J_{NO2}$ during this campaign study was predicted using the measured UVB light and the modelled photolysis rates. These information has been shown in the SI. The time series of the daytime photolysis rates were added in the SI (Fig. S7) and shown in Fig. R6. Overall, the $J$ values are comparable with literature data during the similar season in Beijing (Tan et al., 2018b;Zhang et al., 2019c). In addition, we recently compared the OH concentration using the $J_{O1D}$ with that derived from measured $H_2SO_4$ concentration using a box model. We found the OH concentration calculated with the two methods are comparable as shown in Fig. R7.

[Figure]

Fig. R6. The photolysis rate of $NO_2$, HONO, $O_3$ (O1D) and nitrate (middle value) from 8:00 am to 6:00 pm.

[Figure]

Fig. R7. Comparison of OH concentration between UV proxy method and $H_2SO_4$ proxy method.

In lines 313-314 in the revised manuscript, we added a sentence "The details about the $J_{HONO}$ and $J_{O1D}$ calculation were shown in the Supplement Information and their time series were shown in Fig. S7".

Line 291: photolysis of HCHO is actually the primary source of $HO_2$.

**Response:** Thank you. OH is formed from the reaction between NO and $HO_2$, which is related to HCHO photolysis (Alicke et al., 2003). We revised it as "...,while photolysis of $O_3$ and HCHO related reactions usually dominated primary OH production in summer (Alicke et al., 2003)" in lines 330-331 in the revised manuscript.

Line 330: delete "and".

**Response:** Thank you. It has been corrected in line 366 in the revised manuscript.

Lines 333-336: Several recent papers about the nitrate aerosol trend and formation mechanisms in China are highly relevant to this study, and should be acknowledged.

Wen et al., Summertime fine particulate nitrate pollution in the North China Plain: increasing trends, formation mechanisms, and implications for control policy, Atmospheric Chemistry and Physics, 18, 11261-11275, 2018.

Sun et al., Two years of online measurement of fine particulate nitrate in the western Yangtze River Delta: influences of thermodynamics and N2O5 hydrolysis, Atmospheric Chemistry and Physics, 18, 17177-17190, 2018.

**Response:** Thank you so much. These work have been cited in the revised manuscript (lines 369-370).

Line 363: high or low HONO concentration?

**Response:** Thank you. We think is should be low HONO concentration if the secondary formation is unimportant.

Line 408: Tan et al., (2019)

**Response:** Thank you. It has been corrected in line 453 in the revised manuscript.

Line 407-408: it would be better if the authors could provide the estimated OH levels here.

**Response:** Thank you. The calculated OH concentration was added in the revised SI (Fig. S11) and Fig. R8. And in lines 472-473 in the revised manuscript, we added a sentence "The time series of OH concentration calculated using different methods was shown in Fig. S11".

[Figure]

Fig. R8. Estimated OH concentration using different methods.

Lines 430-431: it is not clear how the photolysis frequency of nitrate was corrected? Details are needed here. What are the J values used finally?

**Response:** Thank you. In the literature (Bao et al., 2018), the authors measured the photolysis of $PM_{2.5}$ samples under irradiation by a Xenon lamp. The authors adjusted the *J* value to ambient sunlight condition at solar zenith angle of 0°. We additionally normalized this value according to the solar zenith angle during our observation according to the latitude, the date and the observed sunlight intensity. They reported a series of $J_{nitrate}$ which varied from $1.22 \times 10^{-5}$ to $4.84 \times 10^{-4}$ $s^{-1}$ with a men value of $8.24 \times 10^{-5}$ $s^{-1}$. Thus, we used these three values to calculate the lower limit, higher limit and middle value of the $J_{nitrate}$. The time series of the middle value has been added as an example in Fig. S7 in the revised SI and Fig. R6. We added a sentence as "The time series of the measured nitrate concentration and the middle value of $J_{nitrate}$ were shown in Fig. 1 and Fig. S7, respectively" in lines 493-495 in the revised manuscript.

Line 553-554: again, it is budget analysis other than stationary state calculation.

**Response:** Thank you. It has been corrected in line 683 in the revised manuscript.

Line 568: indirect production

**Response:** Thank you. It has been corrected in line 698 in the revised manuscript.

**References:**

[revised manuscript text omitted]

---

## Author Comment (AC3) · 18 May 2020

**Reviewer 3#**

The title of this paper is very intriguing that (1) wintertime HONO promotes aerosol formation and (2) >50% of observed HONO is traffic related in Beijing. After reviewing this paper, I think it will be a grave mistake if the editor decides to publish this paper with these two conclusions in any form. The conclusions are pure speculations. I find no evidence to support either of the two claims in this paper.

**Response**: Thank you for your comments. We will answer your questions in the following section point by point.

The discussion for conclusion (1) is in section 3.2. One of the many mistakes in this section is that the authors do not understand that the largest source of OH is from the reaction of $HO_2+NO$. Even when OH production from HONO photolysis is larger than from $O_3$ photolysis, the effect on OH is much smaller than the photolysis rate comparison. Line 301-304 is based on another paper; the data in this paper do not either support or dispute that oxidation by OH promotes aerosol formation. Figure 2D is used at the observation evidence supporting conclusion (1). There are many reasons that HONO/CO correlates with OA/CO. For example, CO is primary in winter in Beijing. If HONO and OA variations are from secondary sources, there will be high correlations as shown. Line 318 states ": : : it was reasonable to mainly ascribe the increase of OA concentration to local secondary formation initiated by OH radical from HONO photolysis." It is a pure speculation. The observation data in this paper do not support this statement. It is the same with Line 328. The vague statement cannot be supported by the data in this paper. Line 332 is again a speculation. Ammonia is mostly neutralized by sulfate in Beijing. Line 338-400 is another speculative and ambiguous statement. Line 345-345 cites other people's work but is not supported by the data in this work.

**Response**: Thank you for your instructive comment. The budget of $HO_x$ or $RO_x$ radical has been investigated at several locations in China based on field measurements and modelling studies (Tan et al., 2018;Tan et al., 2017;Tan et al., 2019;Tang et al., 2015). Using WRF-Chem model, Tang et al (2015) proposed that $HO_2+NO$ was the major OH source, followed by HONO photolysis and $O_3$ photolysis in Beijing, Shanghai and Guangzhou when both primary and secondary OH sources were taken into consideration (Figure R1). As shown in this figure, however, photolysis of HONO was still an important OH sources and the dominate primary OH source in Beijing. Other studies also confirmed that HONO photolysis is an important OH sources, in particular, dominated the primary OH source at various locations (Figure R2) (Tan et al., 2019;Liu et al., 2019;Tan et al., 2018;Tan et al., 2017).

[Figure]

Figure R1. Averaged production rate of OH in Beijing, Shanghai and Guangzhou (Tang et al., 2015).

[Figure]

Figure R2. Comparison of the OH–HO$_2$–RO$_2$ radical budget in four cities under daytime conditions (06:00 to 18:00 LT). The numbers are sorted from left to right in the order of Beijing, Shanghai, Guangzhou, and Chongqing. The blue, black, red, and yellow boxes denote the primary radical sources, radical termination, radical propagation, and equilibrium between radicals and reservoir species, respectively (Tan et al., 2019).

We agree with you that HO$_2$+NO is the major OH source when both the primary and secondary OH sources are taken into consideration. In section 3.2, we were not going to discuss the budget of OH or RO$_x$. We want to confirm that HONO should play an important role in the initiation of RO$_x$ chemistry during our observation. From line 285 to 289, in the original version of the manuscript, we may mislead you because of the improper statements ("This means that the photolysis of HONO dominates the daytime OH production in polluted days in winter, while photolysis of O$_3$ behaves as a bigger OH source from April to June. This is consistent with the previous findings that HONO photolysis is the dominant OH source in winter of BTH"). In lines 324-328 in the revised manuscript, we revised it "These results mean that the photolysis of HONO should play an important role in the initiation of the daytime HO$_x$ and RO$_x$ chemistry on polluted days in winter, while photolysis of O$_3$ becomes more important from April to June. This is consistent with the previous findings that HONO photolysis dominants the primary OH source in winter of BTH…". At the same time, a sentence has also been added in lines 305-307 in the revised manuscript "In addition, it has been confirmed that HONO dominates the primary OH source at various locations (Tan et al., 2018;Liu et al., 2019;Tan et al., 2017;Aumont et al., 2003)".

We agree that the relationship among different pollutants are very complicated in the atmosphere because many variables are entangled. So, it is difficult to isolate the cause and effect relationship between two variables. In both laboratory and modeling studies, one can change the experiment conditions or the input parameters to test the sensitivity of a target parameter to a given variable. For example, a modeler can change the HONO concentration to simulate the change of aerosol concentration and quantify the influence of HONO on secondary aerosol formation. However, this is impossible for field measurements. Thus, correlation analysis is a common method to reasonably deduce the possible mechanism occurring in the atmosphere based on existing knowledge and reasonable assumptions in field measurements (it does so even in modeling studies and laboratory studies). For example, based on correlation analysis, it has been proposed that amines play a crucial role in new particle formation (Kirkby et al., 2011;Almeida et al., 2013) and NH$_3$/NO$_2$ can promote sulfate formation in aqueous phase (Wang et al., 2016).

It has been well recognized that secondary organic aerosol is formed via multiple steps oxidation of VOCs (Kroll and Seinfeld, 2008). At the same time, HOx and ROx play very important role in VOCs oxidation (Atkinson et al., 2006). As discussed above, HONO is the important source of primary OH in the atmosphere. It also has been found that HONO is responsible for the initiation of photochemical reactions in chamber studies (Rohrer et al., 2005). In addition, modelling studies have confirmed that HONO can enhance secondary aerosols formation in Beijing-Tianjin-Hebei (BTH) region (Zhang et al., 2019b) and Pearl-River-Delta (PRD) region of China (Zhang et al., 2019a;Xing et al., 2019). Therefore, it is reasonable to deduce that the increase of OA concentration ($\Delta c_{OA}$) should be related to the OH from HONO photolysis ($\Delta c_{HONO}$) after normalized to CO as supported by the linear correlation between $\Delta c_{OA}/c_{CO}$ and -$\Delta c_{HONO}/c_{CO}$ in Figure 2D. CO is a primary pollutant and a stable species in the atmosphere like BC. Thus, it was used to partially alleviate the influence of PBL variation (Cheng et al., 2016). In addition, $\Delta c_{OA}/c_{BC}$ has also been used to characterize SOA formation during air mass transport (Liggio et al., 2016). Therefore, we correlated the $\Delta c_{OA}/c_{CO}$ with the -$\Delta c_{HONO}/c_{CO}$ rather than the $c_{OA}$ with the $c_{HONO}$. This has been pointed out as in lines 291-293 in the revised manuscript "After partially ruling out the possible influence of PBL variation by normalizing the concentrations of all pollutants to CO (Cheng et al., 2016) or BC (Liggio et al., 2016)…".

It should be noted that the daytime lifetime of HONO is very short due to photolysis. This means regional transport should has little influence on local HONO concentration. However, OA concentration is ready to be affected by regional transport. Thus, we chose these pollution events under stagnant metrological conditions. Therefore, we pointed out that "As the meteorological condition was stagnant during these cases as indicated by the low wind speed (< 1.0 m s$^{-1}$, Fig. S5D), it was reasonable to mainly ascribe the increase of OA concentration to local secondary formation initiated by OH radical from HONO photolysis" from line 317 to 319 in the original versions of the manuscript. It has been recognized that oxidation of $NO_2$ by OH dominates the daytime nitrate formation (Tian et al., 2019). Thus, we can deduce that OH from photolysis of HONO should promote nitrate formation because of the good correlation between the $\Delta c_{nitrate}/c_{CO}$ and -$\Delta c_{HONO}/c_{CO}$ in lines 375-377 in the revised manuscript.

In North China Plain, $NH_3$ is enough to neutralize both sulfate and nitrate the due to the intensive emission of $NH_3$. Figure R3A shows the correlation between the charge of $NH_4^+$ and anions (including sulfate, nitrate and chloride) in non-refractory $PM_{2.5}$ measured using the ACSM in this work. In general, ammonium can neutralize both nitrate and sulfate. In addition, as shown in Figure R3B, the cations can also neutralize the anions measured using a MARGA in Shijiazhuang from March, 2018 to April, 2019. Therefore, the increase of $\Delta c_{ammounium}/c_{CO}$ and $-\Delta c_{HONO}/c_{CO}$ can be ascribed to the fact that ammonium keeps the pace of nitrate through neutralization. In the revised SI, we added the correlation of the charge between the cations and anions in Fig. S8. In the revised manuscript (lines 368-371), we also pointed out that "We explained the increased ammonium as the result of enhanced neutralization of $HNO_3$ by $NH_3$ (Wang et al., 2018;Wen et al., 2018;Sun et al., 2018) because $NH_4^+$ was adequate to neutralize both sulfate and nitrate as shown in Fig.S8".

[Figure]

Figure R3. Correlation of the charge between inorganic anions and cations (A) in non-refractory $PM_{2.5}$ in Beijing and (B) in soluble $PM_{2.5}$ in Shijiazhuang.

From line 338 to 340 in the original manuscript, we concluded that promotion effect of HONO photolysis on nitrate formation could not be excluded. From line 240 to 347, we proposed that HONO photolysis has little influence on sulfate formation based on the correlation between $\Delta c_{sulfate}/c_{CO}$ and $-\Delta c_{HONO}/c_{CO}$ as well as the previous studies, and then made a conclusion that photolysis of HONO could promote aerosol formation during pollution events in winter. From line 348 to 400, we showed the calculation details about HONO emission from vehicle and soil after referring to literatures. From line 345 to 348, we cited the results of modeling studies. We pointed out that "this work well supported the recent modeling results that HONO could obviously promote the aerosol production in winter (Zhang et al., 2019a;Zhang et al., 2019b;Xing et al., 2019;An et al., 2013) from the point of view of observation". So, we need to cite these previous work.

Conclusion (2) is based on some calculation that was not described in the paper. Line 376-378 states that the mean emission factor is 1.17% with a lower limit of 0.18% and an upper limit of 1.8%. (Why is the mean so close to the upper limit and 6.6 times larger than the lower limit?) The mean value is similar to previous studies and is not the reason for conclusion (2). Line 381 gives a vehicle HONO emission rate of 0.085 to 0.34 ppbv/h. The unit implies some volume was used in the calculation. No discussion was given on what volume was used and how it varied in a day. Another important factor not considered in this study is the outflow of vehicle HONO and NOx by advection at night. It is the largest sink at night but is not included in the budget discussion. The nighttime source of $NO_2$ from the ground is 38 times less than vehicle emissions. However, no other paper I know of found that HONO concentrations at night cannot be explained mostly by a ground source. It led me to conclude that the vehicle HONO emission source in this paper is overestimated by 10-100 times. The authors should look at previous modeling papers that included vehicle HONO emissions. What they found is that the effect of vehicle HONO emissions is small.

**Response**: Thank you for your comments. As for the contribution of vehicle-related emission to HONO source, the details have been discussed in both Section 2.2 and 3.3. In this work, we used two different methods to estimate the emission ratio of HONO to NOx (HONO/NOx) from vehicle exhaust. 1.8 % was calculated based on empirical analysis of field data, while 1.17% was obtained by using the low limit correlation of field data. Table R1 and Table S3 summaries the emission ratio from vehicles estimated or measured in China. The lowest value is 0.18% based on chassis dynamometer tests. From line 376 to 378 in the original manuscript we described as "Thus, three levels of vehicle emission factor were considered. 1.17±0.05% was taken as the middle value, while 0.18% (Liu et al., 2017) and 1.8 % were the lower limit and the upper limit, respectively". Here, we used a "middle value" (using the low limit correlation method of filed data) but not a "mean value". Actually, this value is very close to the mean emission ratio (1.21) if we consider all these reported values in Table R1. In the revised manuscript, we revised this sentence "Thus, three levels of vehicle emission factor were considered. 1.17±0.05% was taken as the middle value which was very close to the mean emission ratio (1.21) for all of these reported values in China (Li et al., 2018;Xu et al., 2015;Yang et al., 2014;Liu et al., 2017;Gall et al., 2016;Meng et al., 2019), while 0.18% (Liu et al., 2017) and 1.8 % were the lower limit and the upper limit, respectively" from line 415 to 420 in the revised manuscript.

Table R1. Summary of the measured emission ratio of HONO to NOx from vehicles in China.

| Time | Place | Methods | ΔHONO/ΔNOx | | Reference |
|------|-------|---------|------------|------|-----------|
| | | | Range | Mean | |
| 2015/9/1-2016/8/31 | Ji'nan, Shandong | Empirical analysis of field data | 0.19%-0.87% | 0.53±0.20% | (Li et al., 2018) |
| 2011/8/3-2012/5/31 | Hongkong | Empirical analysis of field data | 0.5%-1.6% | 1.2±0.4% | (Xu et al., 2015) |
| 2015/3/11-2015/3/21 | Hongkong | Tunnel experiment | - | 1.24±0.35% | (Liang et al., 2017) |
| 2014 | Beijing | Tunnel experiment | - | 2.1% | (Yang et al., 2014) |
| 2017 | Beijing | Chassis dynamometer test | 0.03%-0.42% | 0.18% | (Liu et al., 2017) |
| 2016/12/16 - 2016/12/24 | Beijing | Empirical analysis of field data | - | 1.3% | (Zhang et al., 2019c) |
| 2016/12/7-2016/12/13 | Beijing | Low limit correlation of field data | - | 1.41% | (Meng et al., 2019) |
| 2018/2/1-2018/6/30 | Beijing | Empirical analysis of field data | 1.3-2.4% | 1.8±0.5% | This study |
| 2018/2/1-2018/6/30 | Beijing | Low limit correlation of field data | - | 1.17±0.05% | This study |

In Section 2.2, we defined the calculation method for the emission rate of HONO from vehicles "The emission rate ($E_{HONO}$, ppbv h$^{-1}$) was calculated based on the emission flux ($F_{HONO}$, g m$^{-2}$ s$^{-1}$) and PBL height ($H$, m) according to the following equation,

$$E_{HONO} = \frac{a \cdot F_{HONO}}{H} \quad (2)$$

where, α is the conversion factor ($\alpha = \frac{1 \times 10^9 \cdot 3600 \cdot R \cdot T}{M \cdot P} = \frac{2.99 \times 10^{13} \cdot T}{M \cdot P}$), $M$ is the molecular weight (g mol$^{-1}$), $T$ is the temperature (K) and $P$ is the atmospheric pressure (Pa)" from line 152 to 156

in the original manuscript. The emission flux of HONO was calculated according to the the hourly emission flux of NOx from vehicle sector (Yang et al., 2019) and the relative emission ratio of HONO to $NO_x$ as discussed above. It was pointed out as "The hourly NOx emission inventory from vehicles in Beijing, with an annual emission rate of 109.9 Gg $yr^{-1}$ (Yang et al., 2019), was used when calculating the $E_{vehicle}$ in this work." from line 379 to 381 in the original manuscript. To make it clearer, we revised this sentence "The $E_{vehicle}$ was calculated using the hourly $NO_x$ emission inventory from vehicles in Beijing (Yang et al., 2019) after converted to emission flux of HONO ($F_{HONO}=F_{NOx}\times HONO/NO_x$) and the PBL height as described in Section 2.2. Thus, the calculated emission rate reflected the diurnal variation of both the emission inventory and the PBL height" in lines 421-425 in the revised manuscript.

As you suggested, the outflow HONO by advection is an important sink of HONO in the night (Gall et al., 2016;Meng et al., 2019). The the loss of HONO ($T_{advenction}$) via vertical advection can be calculated according to the following equation,

$$T_{vertical} = -K_h(z,t)\frac{\partial c(z,t)}{\partial z}\frac{1}{h} \quad (R1)$$

where $K_h$(z,t) is the eddy diffusivity of heat ($m^2$ $s^{-1}$) at height $z$ (m) and time $t$, $h$ is the height of the second layer (Gall et al., 2016). We added two paragraphs to discuss the transport in lines 222-232 "Vertical transport by advection ($T_{vertical}$), which is an important sink of HONO in the night (Gall et al., 2016;Meng et al., 2019), can be calculated according to equation (12).

$$T_{vertical} = -K_h(z,t)\frac{\partial c(z,t)}{\partial z}\frac{1}{h} \quad (12)$$

where $K_h$(z,t) is the eddy diffusivity of heat ($m^2$ $s^{-1}$) at height $z$ (m) and time $t$, $h$ is the height of the second layer (18 m in this study) (Gall et al., 2016). On the other hand, both the vertical and horizontal transport can be estimate according to Eq. (13),

$$T_{vertical} = k_{dilution}(c_{HONO} - c_{HONO,background}) \quad (13)$$

where $k_{dilution}$ is a dilution rate (0.23 $h^{-1}$, including both vertical and horizontal transport) (Dillon et al., 2002), $c_{HONO}$ and $c_{HONO,background}$ is the HONO concentration at the observation site and background site, respectively (Dillon et al., 2002)" and lines 574-598 "As pointed in Section 2.2, vertical transport by advection is an important nocturnal sink of HONO (Gall et al., 2016). In this work, the vertical distribution of HONO concentration is unavailable. Recently, Meng et al. (2019) measured the vertical distribution of HONO in Beijing in

December, 2016. The concentration of HONO showed nearly flat profiles from ground level to 240 m in pollution events after sunset, while negative profiles of HONO were observed in pollution events during night (Meng et al., 2019). The nighttime concentration gradient was $0.0047\pm0.0025$ ppb m$^{-1}$ derived from the nighttime dataset (Meng et al., 2019). In the daytime, we assume a zero concentration gradient. On the other hand, the eddy diffusivity of heat in urban environment was measured in New Delhi, Indian (Yadav et al., 2003). Using their dataset with the wind speed lower than 2.0 m s$^{-1}$, we derived the relationship between the $K_h$ and the wind speed (WS) ($K_h=0.9389\times$WS$-0.3374$ m$^2$ s$^{-1}$). The nighttime $T_{vertical}$ changed from 0.15 to 0.37 ppbv h$^{-1}$ in winter, while it was from 0.12 to 0.68 ppbv h$^{-1}$ according to Eq. (12) from April to June. Because the wind speed was usually lower than 1.0 m s$^{-1}$ in pollution events (Fig. S6), horizontal transport should have little influence on the daytime sources or sinks of HONO because of the short lifetime of HONO. In the night, 79 % of the wind speed was lower than 1.0 m s$^{-1}$ in winter, thus the air masses from suburban areas should have influence on the sources and sinks of HONO in Beijing. If the HONO concentration at background was zero, the vertical and horizontal transport rate of HONO varied from 0.17 to 0.61 ppbv h$^{-1}$ calculated according to Eq. (13) on haze days in winter and from 0.15 to 0.74 ppbv h$^{-1}$ in pollution events from April to June. These values are higher than that calculated according to Eq. (12). Because the background HONO concentration was unavailable, we only considered the nighttime transport calculated according to Eq. (12) in the following section". At the same time, the relative contribution of each source in Section 3.3 was updated based on the total sinks.

HONO production from heterogeneous reactions of $NO_2$ on both aerosol surface and ground surface greatly depend on the uptake coefficient ($\gamma_{NO2,\ BET}$) and the surface to volume ratio (S/V). The mean and median surface to volume ratio of aerosols are $1.33\times10^{-3}$ and $1.36\times10^{-3}$ m$^{-1}$ during pollution events. This is comparable with the input parameters in modeling studies (Zhang et al., 2016). However, as discussed from line 501 to 503 in the revised manuscript, the $\gamma_{NO2,\ BET}$ on aerosols varied from $5\times10^{-9}$ to $9.6\times10^{-6}$. The $\gamma_{NO2,\ BET}$ on pure oxides was usually higher that on composite oxides. The selection of a proper $\gamma_{NO2,\ BET}$ is quite tricky. For example, a fixed $\gamma_{NO2,\ BET}$ was set to $1\times10^{-6}$ in nighttime and $2\times10^{-5}$ in daytime due to photochemical reaction of $NO_2$ on soot surface in a modelling study (Zhang et al., 2016;Aumont et al., 2003). The nighttime $\gamma_{NO2,\ BET}$ is ~2 orders of magnitude higher than ours

$(1.2 \times 10^{-8})$ as suggested by Crowley et al. (Crowley et al., 2010). In our previous work, we have measured the $\gamma_{NO2, BET}$ on kaolin and hematite (Liu et al., 2015) and soot (Han et al., 2013). The initial $\gamma_{NO2, BET}$ on kaolin is $4.85 \pm 0.39 \times 10^{-8}$ at 47 % RH. It should be pointed out that the $\gamma_{NO2, BET}$ decreases significantly with exposure time due to surface saturation. Fig. R4 shows the typical uptake curve of $NO_2$ and the $\gamma_{NO2, obs}$ which is not normalized to the specific surface area. The $\gamma_{NO2, BET}$ at steady state ($2.56 \times 10^{-9}$ to $4.56 \times 10^{-9}$ on kaolin and $1.23 \times 10^{-8}$ to $1.50 \times 10^{-8}$ on hematite) is around one order of magnitude lower than the initial $\gamma_{NO2, BET}$ as shown in Fig. R5. Therefore, we used the recommended value $(1.2 \times 10^{-8})$ (Crowley et al., 2010) as the base case. On the other hand, the $\gamma_{NO2, BET}$ decreases with RH due to competitive adsorption (Liu et al., 2015). High mass concentration of $PM_{2.5}$ usually accompanied with high RH in winter in Beijing. Thus, we calculated the RH dependent $\gamma_{NO2, BET}$ according to the equation we determined previously (Liu et al., 2015) and the measured ambient RH in this work. This should be more reasonable than a fixed uptake coefficient used in modeling studies.

[Figure]

Fig. R4. Uptake curves of $NO_2$ on (A) kaolin and (B) hematite; and evolution of the observed uptake coefficient on (C) kaolin and (D) hematite at 298 K and at 47 % of RH (Liu et al., 2015).

[Figure]

Fig. R2. The dependence of $\gamma_{NO2,BET}$ on relative humidity (Liu et al., 2015).

In the original version of manuscript, we found a bug when calculating the heterogeneous reaction of $NO_2$ on black carbon. A conversion factor of time from second to hour was missed. So, the contribution of heterogeneous reaction to HONO source was underestimated in the original version. Now, the $P_{aerosol}$ was 0.038±0.030-0.088±0.072, which was dominated by the heterogeneous reaction of $NO_2$ on black carbon. It was on the same orders as soil emission. However, this was still significantly lower than the contribution of vehicle emission. In the revised manuscript, we updated Figures 3-5 and the corresponding numbers in section 3.3.

As for heterogeneous reaction of $NO_2$ on ground surface, we used the same $\gamma_{NO2,BET}$ on dust aerosol in the night, which is similar to the methodology used in modelling studies (Zhang et al., 2016;Aumont et al., 2003). As pointed out in Section 2.2 from line 197 to 209 in the revised manuscript, the S/V was estimated using the surface roughness calculated in the light of satellite image of Beijing. The calculated surface roughness is 3.85, which is slightly higher than the value (2.2) used by Li et al. (2017). Thus, the calculated S/V varied from 0.0015 to 0.0385 $m^{-1}$ (with a mean value of 0.0125 $m^{-1}$) because of the variation of the PBL height during pollution events. However, a fixed S/V of ground surface was set to 0.3 $m^{-1}$ in the modeling studies (Zhang et al., 2016). This corresponds to a surface roughness ~92, which means the surface area is ~92 times of the projected area of ground. It's too high. If both the $\gamma_{NO2,BET}$ ($1\times10^{-6}$) and surface roughness are increased to the values used in modeling studies, the nighttime production rate of HONO via heterogeneous reaction of $NO_2$ on ground surface will be 2.9 ppb $h^{-1}$. This means a large sink missed if this number is reasonable. In lines 558-561 in the revised manuscript, we also revised the sentence "These results mean that heterogeneous reaction might not be a major HONO source. This is consistent with a recent work that found heterogeneous reaction being unimportant when compared with traffic emission during haze events in winter in Beijing (Zhang et al., 2019c)" and in lines 605-607 in the revised manuscript, "Heterogeneous reactions of $NO_2$ on aerosol surface and ground surfaces were not the major HONO source during night unlike the modelled results (Zhang et al., 2016;Aumont et al., 2003)".

During the Chinese New Year (CNY) and the COVID-19 epidemic in 2020, traffic emission decreased significantly in Beijing. This provides us a unique opportunity to verify the relative importance of each HONO source. We only analyzed the nocturnal data from January 1 to February 29 because the HONO sources related to photochemical reactions could be avoided. The CNY vacation was from January 23 to February 2. These results are in preparation for a separate paper. Fig. R4 shows the relative change of the concentrations of HONO, $NO_x$ and non-refractory $PM_{2.5}$ (NR-$PM_{2.5}$). The traffic index and the chemical age of the air masses which is defined as -log($NO_x/NO_y$) are also shown in Fig. R4. $NO_y$ was measured with a $NO_y$ analyzer (Thermo 42i-Y). The concentration of nighttime HONO decreased significantly during the CNY and after the CNY accompanied with the reduction in vehicle emission as supported by both the concentration of $NO_x$ and traffic index (Fig. R4B). Interestingly, the NR-$PM_{2.5}$ concentration during and after the CNY increased obviously when compared with that before the CNY. The effective conversion of $NO_2$ aerosol surface was almost constant because both the promotion effect of increased $PM_{2.5}$ concentration and the inhibition effect of reduced $NO_2$ concentration during and after the CNY as shown in Fig. R4C. On the other hand, we found that the nighttime chemical age of the air masses during and after the CNY was also obviously larger than that before the CNY (Fig. R4D). This means that heterogeneous reaction of $NO_2$ on both aerosol surface and ground surface should be more effective due to longer residence of the air masses during and after the CNY than that before the CNY. When the reduction of $NO_2$ concentration was taken into consideration, the product of $NO_2$ concentration and $-\log(NO_x/NO_y)$ in different periods increased slightly (Fig. R5). Therefore, the observed HONO concentration should be at least constant in different periods and independent on the reduction of vehicle emission if heterogeneous reaction of $NO_2$ on aerosol and ground surfaces dominates nighttime HONO source or if the vehicle emission is a minor HONO source. However, we observed the decrease of HONO along with reduction of vehicle emission (Fig.R4A). This well supports our (and other researchers' (Meng et al., 2019;Zhang et al., 2019c) conclusion that vehicle emission is an important source of HONO in Beijing.

[Figure]

Fig. R4. Relative change of nighttime (A) HONO concentration, (B) NOx concentration and the traffic index, (C) non-refractory $PM_{2.5}$ (NR-$PM_{2.5}$) concentration and (D) relative chemical age ($-\log(NO_x/NO_y)$) of air masses before Chinese New Year (CNY) (2020.1.1-2020.1.22), during CNY (2020.1.23-2020.2.1) and after CNY (2020.2.2-2020.2.29).

[Figure]

Fig. R5. Relative change of nighttime the product of $NO_2$ and NR-PM$_{2.5}$ concentration and the product of $NO_2$ concentration and $-\log(NO_x/NO_y)$ in different periods.

In lines 523-528 in the revised manuscript, we pointed out "The $A_s$ of aerosols which was measured using a DMPS varied from $1\times10^{-4}$ to $4.8\times10^{-3}$ m$^{-1}$ with a mean value of $1.4\pm0.5\times10^{-3}$ m$^{-1}$ during pollution events. This value is comparable with that used in modeling studies (Zhang et al., 2016;Aumont et al., 2003). The $A_s$ of ground surface which was calculated according to Eq. (6) and (7) varied from $1.5\times10^{-3}$ to $3.85\times10^{-2}$ m$^{-1}$ with a mean value of $1.3\pm0.9\times10^{-2}$ m$^{-1}$ during pollution events. The surface roughness was 3.85 calculated according to Eq. (7)". And from line 544 to 554 in the revised manuscript, we added a paragraph to discuss the reason why we obtained the low production rate of HONO via heterogeneous reaction of $NO_2$ on ground and aerosol surfaces as "It should be pointed out that HONO production from heterogeneous reaction of $NO_2$ on both aerosol and ground surface greatly depend on the $\gamma_{NO2, BET}$ and $A_s$. The $A_s$ of aerosols is comparable with the modeling input. However, the small nighttime $\gamma_{NO2, BET}$ ($10^{-8} - 10^{-7}$) were used in this work rather than the $\gamma_{NO2, BET}$ ($1\times10^{-6}$) used in modelling studies (Zhang et al., 2016;Aumont et al., 2003). This leads to a lower production rate of HONO from heterogeneous reaction of $NO_2$ on aerosols. As for heterogeneous reaction of $NO_2$ on ground surface, besides the small $\gamma_{NO2, BET}$ used in this work, the $A_s$ of ground surface (0.0015 to 0.0385 m$^{-1}$) calculated using the surface roughness and PBL height was also significantly lower than the fixed value of 0.3 m$^{-1}$ used in modeling studies that might overestimate the contribution of HONO production from heterogeneous reaction of NO$_2$ on ground surface".

In summary, I think that the calculation and reasoning of this paper are either incorrect or ambiguous. It does seem likely that any revision can correct the flaws in the two main conclusions. I suggest that the authors scratch the conclusions and redo the analysis of their observation data.

**Response:** Thank you for your comments again. We have answered your questions above.

[revised manuscript text omitted]

---

## Author Response (AR2)

Dear Editor,

We appreciate the careful consideration of our manuscript by the reviewer and you. We have carefully responded to all of the point-by-point comments and issues raised by the reviewers and have revised the manuscript accordingly. We also have edited the language. These revisions are described in detail below.

**Reviewer 2#**

A main problem with this paper is that the authors make use of many qualitative arguments based on previously published papers instead of using their observation data to quantitatively support their speculations. Despite my strong objections to these speculations in the first-round review, the long-winded response mostly re-stated what was in the original paper. Reviewing the original paper and reading the responses are painful exercises because the authors' arguments are mostly based on some "beliefs" seemingly garnered from incorrectly reading other papers. Their observations were often either ignored or misused. My concerns are not properly addressed and I cannot find good reasons to change my original recommendation. (1) In response to my first comment, they cited several previously published papers and corrected a mistake I pointed out in the first-round review. A larger primary radical source from HONO photolysis does not mean that OH increases in the same proportion as the primary source increase. The question, which they did not answer in the response, is how much OH increase can be sustained by the photolysis of HONO. The response ran around in circles of this question.

**Response:** Thank you for your patience to review the response file. We hope to well reply each point of your comments. So, the response file was somewhat long.

We agree with you that it is better to quantitatively discuss the contribution of HONO to OH concentration if it is possible. However, it is impossible to get the quantitative information at the present time. This will be done using an air quality model in the future. In the revised manuscript (lines 720-722), we have pointed out it as

"Finally, it is necessary to quantify the contribution of traffic-related source of HONO on OH production and secondary aerosol formation based on modelling studies in the future". And in lines 372-374, we pointed it as "Overall, this work qualitatively supported the recent modelling results that HONO could promote the aerosol production in winter (Zhang et al., 2019a;Zhang et al., 2019b;Xing et al., 2019;An et al., 2013) from the point of view of observation".

The logic of this work is: 1) HONO could promote wintertime OA formation based on qualitative analysis; 2) traffic related emission should be an important contributor to ambient HONO in Beijing. We think the significance of this work is to provide a new perspective to Haze Beijing although more quantitative work is required in the future.

The newly added statement, "These results mean that the photolysis of HONO should play an important role in the initiation of the daytime HOx and ROx chemistry on polluted days in winter, while photolysis of $O_3$ becomes more important from April to June.", did not answer the question on the quantitative effect of HONO photolysis on OH. The statement itself is meaningless. What does "more important" mean? Is "more" for comparison between April-June and winter or between HONO and O3 photolysis? The understanding of chemistry is also flawed. The primary radical source from O3 photolysis is not JO1D*CO3. It is much smaller because most O1D reacts with N2 and O2. JO1D*CO3 cannot be compared to the photolysis rate of HONO directly (which was done in this paper).

**Response:** Thank you. As for the contribution of HONO photolysis on OH concentration or production, we have replied in the first question. The quantitative analysis will be carried out using air quality model in the future.

$J_{O1D}*c_{O3}$ reflects the production rate of O1D instead of OH production rate. The production rate of OH from photolysis of $O_3$ is directly proportional to $J_{O1D}*c_{O3}$. However, the increase of $J_{O1D}*c_{O3}$/JHONO in summer indicates the enhanced role of $O_3$ in OH production compared with that in winter. In the revised manuscript in lines 298-301, we pointed it as "We simply compared the OH production via photolysis of

HONO ($P_{\text{OH-HONO}}=J_{\text{HONO}}\times c_{\text{HONO}}$) and O1D production rate from O$_3$ ($P_{\text{O1D}}=J_{\text{O1D}}\times c_{\text{O3}}$) in Fig. 2 when the PM$_{2.5}$ concentration was larger than 50 μg m$^{-3}$ and the RH was less than 90 % to understand the chemistry in pollution events". In lines 315-319, we revised the sentence as "Because the production of OH from photolysis of O$_3$ should be directly proportion to P$_{\text{O1D}}$, these results imply that the relative importance of the photolysis of HONO compared with that of O$_3$ for initiating the daytime HO$_x$ and RO$_x$ chemistry on polluted days should be more important in winter than that from April to June".

The response did not address my comment of "There are many reasons that HONO/CO correlates with OA/CO. For example, CO is primary in winter in Beijing. If HONO and OA variations are from secondary sources, there will be high correlations as shown."

**Response:** Thank you. If two pollutants are from secondary sources and driven by a third factor, their concentration or normalized concentration to CO will be highly correlated. However, the absolute value of HONO/CO was actually anti-correlated with OA/CO in the selected pollution events although secondary reaction was the important contributor to both OA and HONO. In this work, we correlated the increased value of OA/CO from the early morning to noon with the corresponding consumed HONO/CO in different pollution events. Because HONO is a well-recognized precursor of OH in the morning and SOA formation is greatly affected by oxidants, we proposed that HONO should promote SOA formation. We think the correlation is reasonable based on the well-known knowledge. In the revised manuscript (lines 324-330), we made it clearer as "Oxidation of trace gas pollutants, in particular VOCs, by OH is their main removal pathway in the troposphere (Atkinson and Arey, 2003), subsequently contribute to secondary aerosol formation (Kroll and Seinfeld, 2008). A very recent work has found that oxidation of VOCs from local traffic emission is still efficient even under pollution conditions (Guo et al., 2020). This means high HONO concentration might promote SOA formation after sunrise because HONO is an important primary OH source in the early morning".

The statement, "It should be noted that the daytime lifetime of HONO is very short due to photolysis. This means regional transport should has little influence on local HONO concentration.", is also incorrect. The lifetime of HONO at night is long and transport affects HONO concentrations. In daytime, if a substantial fraction of HONO is from $NO_2$, transport certainly affects NOx and therefore HONO. In the statement, "As the meteorological condition was stagnant during these cases as indicated by the low wind speed (< 1.0 m s-1, Fig. S5D), it was reasonable to mainly ascribe the increase of OA concentration to local secondary formation initiated by OH radical from HONO photolysis", how could the authors know that the increase of OA is from "local secondary formation initiated by OH radical from HONO photolysis"?

**Response:** Thank you. To make our statement stricter, we revised the statement "As the meteorological condition was stagnant during these cases as indicated by the low wind speed (< 1.0 m s$^{-1}$, Fig. S5D), it was reasonable to ascribe the increase of OA concentration to local secondary formation initiated by OH radical and photolysis of HONO should play an important role in initiation the HOx and ROx chemistry" to "The dataset under stagnant meteorological conditions as indicated by the low wind speed (< 1.0 m s$^{-1}$, Fig. S5D) was analyzed to decrease the contribution of transport to the observed HONO and OA" in lines 331-333.

The new statement, "We explained the increased ammonium as the result of enhanced neutralization of $HNO_3$ by $NH_3$ (Wang et al., 2018;Wen et al., 2018;Sun et al., 2018) because $NH_4^+$ was adequate to neutralize both sulfate and nitrate as shown in Fig.S8" is incorrect. In Beijing winter, higher $HNO_3$ does not necessarily convert more $NH_3$ to ammonia. The authors appear to have a limited understanding of aerosol chemistry.

**Response:** Thank you for your comment. Ammonium is a secondary pollutant. Thus, the increase of particulate ammonium should be correlated to the conversion from ammonia to ammonium. As shown in Figure R1, the concentration of particulate ammonium linearly correlated with that of nitrate and sulfate. In particular, the correlation between nitrate and ammonium is stronger that that between sulfate and

ammonium. Therefore, the daytime increase of ammonium should be highly related to neutralization of sulfate and nitrate by ammonia. We revised the statement (lines 358-360) as "We explained the increased ammonium as the result of enhanced neutralization of sulfate and nitrate by $NH_3$ (Wang et al., 2018; Wen et al., 2018; Sun et al., 2018) because $NH_4^+$ was adequate to neutralize both sulfate and nitrate as shown in Fig. S8".

[Figure]

(2) In response to my comments on the HONO budget, Table S3 is useful. However, looking at Fig. S9, could some of the HONO measured with ~100 ppb NOx be a result of inlet conversion of $NO_2$ to HONO? Some dirty vehicles may have high HONO/$NO_2$ emission ratios, but I find it puzzling why no measurements seem to have HONO/$NO_2$ ratio < 1% for the selected "fresh emission" data points.

**Response:** Thank you for your comment. For HONO measurement, the interference from the sampling inlet is inevitable for any kind of instrument. To decrease the interference, we sampled ambient air from the window and used a sample line as short as possible (~1.0 m). In addition, we did the control experiment by sampling 100 ppb of $NO_2$ balanced with zero air at 50% RH into the instrument. The measured HONO was 0.4 ppb, while the ambient HONO concentration was 5.98±1.00 ppb when the $NO_2$ concentration was higher than 80 ppb on pollution days. This means sampling

interference would overestimate ~6.7 % of HONO concentration on pollution days. If the conversion ratio of NO₂ to HONO in the sampling line is constant, the emission ratio of HONO to NOx should be decrease from 1.17 % to 1.09 %. In the revised manuscript, we updated the related data. In the revised manuscript (lines 397-401), we added a paragraph "It should be pointed out that the interference from the sampling inlet overestimated 6.7 % of HONO concentration based on control experiments. Thus, the ratio of ΔHONO/ΔNOx should be 1.09±0.05% when the interference from the sampling inlet was taken into consideration". The data in Figs.3 and 4 and in the text have been updated correspondingly.

The response did not answer my comment "No discussion was given on what volume was used and how it varied in a day." The response states "The hourly NOx emission inventory from vehicles in Beijing, with an annual emission rate of 109.9 Gg yr-1 (Yang et al., 2019), was used when calculating the E$_{vehicle}$ in this work." The authors need to state what area is used for Beijing. The Beijing metropolitan area is very large. Vehicle emission rates can be extremely low in the rural regions of Beijing, the area of which is much larger than the urban core. A further statement "…the PBL height as described in Section 2.2. Thus, the calculated emission rate reflected the diurnal variation of both the emission inventory and the PBL height" does not provide useful information. I cannot find where they got the PBL data and their calculation on the effects of diurnally varying PBL on the budget of HONO. For example, the nighttime PBLH is usually 10-50 times less than the daytime. Therefore, the nighttime vehicle HONO source is 10 times or more than daytime (after accounting for lower emissions at night). The diurnal variation for the vehicle source in Figure 3 is too small.

**Response:** Thank you. In the first-round of response and in Section 2.2, we pointed out that "The emission rate ($E_{HONO}$, ppbv h⁻¹) was calculated based on the emission flux ($F_{HONO}$, g m⁻² s⁻¹) and PBL height ($H$, m) according to the following equation,

$$E_{HONO} = \frac{\alpha \cdot F_{HONO}}{H} \quad (2)$$

where, α is the conversion factor ($\alpha = \frac{1 \times 10^9 \cdot 3600 \cdot R \cdot T}{M \cdot P} = \frac{2.99 \times 10^{13} \cdot T}{M \cdot P}$), $M$ is the molecular

weight (g mol$^{-1}$), $T$ is the temperature (K) and $P$ is the atmospheric pressure (Pa)". Here, $F_{HONO}$ is the emission intensity of HONO over a unit area.

When calculating the $F_{HONO}$ according to $F_{HONO}=EI_{HONO}/A$, the core urban area of Beijing was used (with 20 km of diameter based on google map and the distribution of annual-average vehicular NOx concentration in Beijing (Yang et al., 2019). This was added in the revised manuscript (lines 173-174) as "where, $EI_{HONO}$, is the emission inventory of HONO (g s$^{-1}$), $A$ is the core urban area of Beijing (m$^2$, with diameter of 20 km),…"

After divided by the PBL height, the emission rate means the emission intensity of HONO in a column. Therefore, the variation of $E_{HONO}$ depends on both the $F_{HONO}$ (or the emission inventory of NOx) and the PBL height. Fig. R2 shows the diurnal variation of the PBL height and the emission inventory of NOx from vehicles in Beijing (Yang et al., 2019). Because both the emission of NOx from vehicles and the PBL height were significantly higher in the day than that in the night, the difference in the daytime and nighttime $E_{HONO}$ in Fig. 3 should be smaller than that of PBL height because the PBL height is the denominator in Eq. (2).

[Figure]

Figure R2. Diurnal curve of PBL height in pollution days and the hourly emission inventory of NOx from vehicles in Beijing (Yang et al., 2019)

As for the PBL height, we measured it using a ceilometer. In lines 146-158, we pointed it "Visibility and planetary boundary layer (PBL) height were measured using a visibility sensor (PWD22, Vaisala) and a ceilometer (CL51,Vaisala), respectively". In addition, the time series of PBL height was shown in Fig. 1F.

The only information of PBLH data used in this study I can find is Fig. S4A. The figure shows that PBLH varies from 20 to 3500 m. The distribution does not seem to show that PBLH increases from winter to summer. It provides no useful information on answering my question of diurnal HONO budget variation.

**Response:** Thank you for your comment. We showed the time series of the PBL height in Fig.1F and Fig. R3. It increased from winter to summer as you pointed out.

[Figure]

Fig. R3. The time series of the PBL height.

The newly added statement, "In the daytime, we assume a zero concentration gradient", is wrong. If the PBLH is 3 km, how can HONO be constant from the surface to 3 km? At night, PBLH is usually low. Vertical mixing is not even a sink of HONO for a budget analysis that extends from the surface to the PBL top. (Eqs. (12) and (13) cannot even be used to estimate the vertical loss of HONO when the vehicle emission source is estimated as emission rate/PBLH).

**Response:** Thank you for your comment. The measurement on the vertical distribution of HONO was scarce at the present time. According to a recent field measurement, the concentration of HONO showed nearly flat profiles from ground level to 240 m in pollution events after sunset, while negative profiles of HONO were observed in pollution events during night (Meng et al., 2019). Unfortunately, they did not measure the vertical profiles in the daytime. In the revised manuscript (lines 571-573), we revised this sentence as "Because the daytime vertical gradient of HONO concentration

is unavailable in Beijing, we do not calculate the daytime vertical transport".

The maximal PBL height was around 3 km in clean days. However, we calculated the HONO budget in pollution days. The mean PBL height was 487±460 m in pollution days. As pointed in the manuscript, we calculated the nighttime $T_{vertical}$ using the reported concentration gradient but not the PBL height. We agree with you that the height of the vertical mixing layer might be lower than the PBL height. This will underestimate the emission rates from vehicle, soil and heterogeneous reaction on ground surface using the PBL height. Actually, this is a common problem for HONO budget calculation even for modelling studies. In the revised manuscript, we added a sentence to discuss the uncertainty in lines 710-713 as "In addition, the exact height of vertical mixing of HONO was assumed to be the same as the PBL height, this might underestimate the contribution of vehicle, soil and heterogeneous reaction on ground surface".

The statement "In the night, 79 % of the wind speed was lower than 1.0 m s-1 in winter" is likely based on surface wind measurements at a site where wind is blocked by buildings in the city. Looking at any meteorological data, wind speed is stronger in winter than summer in Beijing and the average wind speed in Beijing in winter is much higher than 1 m/s.

**Response:** Thank you for your comment. The wind speed was measured on the ground surface (18 m above the ground surface). This was pointed out in Section 2.1. The wind speed is usually stronger in winter than summer in Beijing. However, it varied monthly as shown in Fig. S4. The mean wind speed in Beijing in winter is much higher than 1 m s$^{-1}$. Here, we meant that 79 % of the wind speed was lower than 1.0 m s$^{-1}$ on polluted days when the PM$_{2.5}$ concentration was larger than 50 μg m$^{-3}$ and the RH was less than 90 %. In the revised manuscript (lines 580-584), we corrected it as "In the night, 79 % of the surface wind speed was lower than 1.0 m s$^{-1}$ on pollution days when the PM$_{2.5}$ concentration was larger than 50 μg m$^{-3}$ and the RH was less than 90 % in winter, thus the air masses from suburban areas should have influence on the sources and sinks of

HONO in Beijing".

The authors assume that ground level gamma values are the same as dust aerosols and calculated a low surface HONO source. What is the justification? The assumption is arbitrary.

**Response:** At the present time, the uptake coefficient of $NO_2$ on ground surface was usually assumed to be the same as that on particle surface in modelling studies (Zhang et al., 2016;Aumont et al., 2003). In the revised manuscript (line 511), the references have been added. The urban ground surfaces include plant leaves, building surface, and rock and soil surfaces and so on. To our best knowledge, the uptake coefficient of $NO_2$ on ground surfaces is unavailable at the present time. For example, there is no publication on reaction kinetics of $NO_2$ on plant leaves and rocks. In our previous work, we measured the kinetics of $NO_2$ uptake and HONO formation on kaolinite, which is an important kind of soil. The uptake coefficient of $NO_2$ is on $10^{-8}$ order of magnitude. It is close to that on aerosols recommended by Crowley et al. (2010).

I do not follow the reasoning from "If both the $\gamma_{NO2,BET}$ ($1 \times 10^{-6}$) and surface roughness are increased to the values used in modelling studies, the nighttime production rate of HONO via heterogeneous reaction of $NO_2$ on ground surface will be 2.9 ppb $h^{-1}$. This means a large sink missed if this number is reasonable" to the conclusion statement "These results mean that heterogeneous reaction might not be a major HONO source. This is consistent with a recent work that found heterogeneous reaction being unimportant when compared with traffic emission during haze events in winter in Beijing (Zhang et al., 2019c)". The authors found that the surface source can be much larger (a factor of 10) than the vehicle source. However, because they or previous publications believe that the vehicle HONO source is most important, the authors concluded that the vehicle HONO source is the most important in their dataset too. The argument is circular and meaningless.

**Response:** Thank you for your comment. In the first-round of response file, you pointed

out that we got a too small ground source of HONO, but a large source from vehicle. We calculated the ground source according to these parameters used in previous modelling studies and obtained a very large HONO source from ground surface (2.9 ppb h$^{-1}$). Even if the contribution of other sources is omitted, the nighttime HONO source is much higher than these reported values. On the other hand, according to the reported kinetics of NO$_2$ on different aerosols, we don't think these parameters are reasonable. Thus, we chose a small $\gamma_{NO2,BET}$. This value was recommended by Crowley et al. (2010) and was close to that measured in our laboratory. That is the reason why we get a small ground source compared with vehicular emission. As discussed in the first-round of response file, heterogenous reactions on ground surface and aerosol cannot explain the decrease of HONO concentration during the Chinese New Year, 2020 because the concentrations of both PM$_{2.5}$ and NO$_2$ did not decrease obviously, while HONO concentration decrease obviously due to reduction of vehicle emissions compared with that before Chinese New Year. These results will be discussed in a separate work. This further supported our conclusion that vehicular emission should be more important for HONO source in Beijing when compared with heterogeneous reaction. In the revised manuscript (lines 718-720), we added a sentence "The importance of vehicle emission to HONO source also needs to be further confirmed during special periods such as Chinese New Year when vehicle emission reduces obviously in the future".

The HONO budget analysis is flawed for several reasons. The methodology has errors. There is no closure on the (hourly) budget. Each source and sink terms have very large uncertainties and some arbitrary decisions were made on the parameter values to justify that vehicle emissions are the largest HONO source. The analysis results in this paper are not scientifically credible.

**Response:** Thank you for your comment. Actually, the method for HONO budget calculation is a simple model in this work. Some methods such as emission sources from vehicle and soil, homogeneous reaction between OH and NO, were also reported

in literatures. Because each source was calculated based on several parameters, the uncertainties was inevitable like other modelling studies. The uncertainties of each source have been discussed in the manuscript. Although the sources and sinks are still not closed as shown in Fig. 4, the unknown source has been taken into consideration when we discuss the relative contribution of these sources. Therefore, we think the importance of vehicle emission in HONO source in Beijing should be credible and has been well confirmed by the vehicle emission reduction in Chinese New Year, 2020 as replied in the first-round response file. We think the results of this work will help for understanding the complex cause of haze in Beijing. In addition, this work provided the details about the parameterization for HONO budget calculation. Some parameters are more reasonable compared with previous modelling studies. For example, the surface to volume ratio of ground was calculated based on a surface roughness calculated based on the building surface and PBL height in this work rather than a fixed value of 0.3 $m^{-1}$ (Zhang et al., 2016;Aumont et al., 2003); the uptake coefficients of $NO_2$ on aerosol was chosen based on laboratory results; the emission ratio of HONO/NOx was calculated based on measured data and the newest emission inventory of traffic NOx was used in this work.

References:

An, J., Li, Y., Chen, Y., Li, J., Qu, Y., and Tang, Y.: Enhancements of major aerosol components due to additional HONO sources in the North China Plain and implications for visibility and haze, Adv. Atmos. Sci., 30, 57-66, 10.1007/s00376-012-2016-9, 2013.

Aumont, B., Chervier, F., and Laval, S.: Contribution of HONO sources to the NOx/HOx/O3 chemistry in the polluted boundary layer, Atmos. Environ., 37, 487-498, https://doi.org/10.1016/S1352-2310(02)00920-2, 2003.

Xing, L., Wu, J., Elser, M., Tong, S., Liu, S., Li, X., Liu, L., Cao, J., Zhou, J., El-Haddad, I., Huang, R., Ge, M., Tie, X., Prévôt, A. S. H., and Li, G.: Wintertime secondary organic aerosol formation in Beijing–Tianjin–Hebei (BTH): contributions of HONO sources and heterogeneous reactions, Atmos. Chem. Phys., 19, 2343-2359, 10.5194/acp-19-2343-2019, 2019.

Yang, D., Zhang, S., Niu, T., Wang, Y., Xu, H., Zhang, K. M., and Wu, Y.: High-resolution mapping of vehicle emissions of atmospheric pollutants based on large-scale, real-world traffic datasets, Atmos. Chem. Phys., 2019, 8831–8843, 10.5194/acp-2019-32, 2019.

Zhang, J., An, J., Qu, Y., Liu, X., and Chen, Y.: Impacts of potential HONO sources on the concentrations of oxidants and secondary organic aerosols in the Beijing-Tianjin-Hebei region of China, Sci. Total Environ., 647, 836-852, https://doi.org/10.1016/j.scitotenv.2018.08.030, 2019a.

Zhang, J. W., Chen, J. M., Xue, C. Y., Chen, H., Zhang, Q., Liu, X. G., Mu, Y. J., Guo, Y. T., Wang, D. Y., Chen, Y., Li, J. L., Qu, Y., and An, J. L.: Impacts of six potential HONO sources on HOx budgets and SOA formation during a wintertime heavy haze period in the North China Plain, Sci. Total Environ., 681, 110-123, 10.1016/j.scitotenv.2019.05.100, 2019b.

Zhang, L., Wang, T., Zhang, Q., Zheng, J., Xu, Z., and Lv, M.: Potential sources of nitrous acid (HONO) and their impacts on ozone: A WRF-Chem study in a polluted subtropical region, Journal of Geophysical Research-Atmospheres, 121, 3645-3662, 10.1002/2015jd024468, 2016.

---

## Author Response (AR3)

Dear Editor,

We greatly appreciate your positive decision and constructive suggestions. As you suggested, the language/grammatical errors have been edited by a native speaker. These revisions are described in detail below.

line 97: "some other researches" is incorrect

**Response:** Thank you. It has been replaced with "Other studies".

line 100: not clear what "Ji'an" refers to

**Response:** Thank you. I'm sorry for the mistake. It's Ji'nan. It has been corrected.

line 101: "In addition, the traffic emission was proposed to be...." should be "....traffic emissions were proposed..."

**Response:** Thank you. It has been corrected.

line 106: "more studies is still required..." should be "are required.."

**Response:** Thank you. It has been corrected.

line 157: "they were believed as minor sources" should be "...to be minor sources.."

**Response:** Thank you. It has been corrected.

line 236: "....pollution events showed...."; "exhibited" would be better.

**Response:** Thank you. It has been corrected.

line 286: "The similar trends...." should be "...similar trends..."

**Response:** Thank you. It has been corrected.

line 315: do not start sentence with "Because"

**Response:** Thank you. This sentence has been corrected as "These results imply that the relative importance of the photolysis of HONO compared with that of $O_3$ for

initiating the daytime $HO_x$ and $RO_x$ chemistry on polluted days is more important in winter than from April to June because the production of OH from the photolysis of $O_3$ should be directly proportional to $P_{O1D}$".

line 325: "....2003), subsequently contribute to..." should be "...contributing..."
**Response:** Thank you. It has been corrected.

line 337: "...HONO exhibited quick reduction" should be "... a quick reduction..."
**Response:** Thank you. It has been corrected.

line 360: I think you mean ammonia not ammonium.
**Response:** Thank you. Yes, I mean ammonia. It has been corrected.

line 396: ".... the lowest marge with..."; not clear what this is
**Response:** Thank you. It has been revised as "In the 2D space of HONO $vs$. $NO_x$ (Fig. S8), the data below the $2^{nd}$ percentile of HONO/NOx and with $\Delta NO/\Delta NOx$ values of greater than 0.8 were chosen for the linear correlation".

line 398: should be: "...an interference from the sampling inlet overestimated HONO concentration by 6.7%..."
**Response:** Thank you. It has been corrected.

line 455: should be - "...taken as the lower limit for..."
**Response:** Thank you. It has been corrected.

line 509: should be - "...on ground surfaces..."
**Response:** Thank you. It has been corrected.

**Additional corrections:**
The edits from a native speaker has been marked in blue.